# Oriented arrangement of simple monomers enabled by confinement: towards living supramolecular polymerization

Yingtong Zong [1], Si-Min Xu [1], Wenying Shi [1✉] & Chao Lu [1✉]

The living supramolecular polymerization technique provides an exciting research avenue. However, in comparison with the thermodynamic spontaneous nucleation, using simple monomers to realize living supramolecular polymerization is hardly possible from an energy principle. This is because the activation barrier of kinetically trapped simple monomer (nucleation step) is insufficiently high to control the kinetics of subsequent elongation. Here, with the benefit of the confinement from the layered double hydroxide (LDH) nanomaterial, various simple monomers, (such as benzene, naphthalene and pyrene derivatives) successfully form living supramolecular polymer (LSP) with length control and narrow dispersity. The degree of polymerization can reach ~6000. Kinetics studies reveal LDH overcomes a huge energy barrier to inhibit undesired spontaneous nucleation of monomers and disassembly of metastable states. The universality of this strategy will usher exploration into other multifunctional molecules and promote the development of functional LSP.

[1] State Key Laboratory of Chemical Resource Engineering, Beijing University of Chemical Technology, Beijing, P. R. China. ✉email: shiwy@mail.buct.edu.cn; luchao@mail.buct.edu.cn

Supramolecular polymers present us with a broad blueprint to struggle for the new generation of functional materials such as soft matter electronics, drug delivery, and catalysts in a relatively simple way[1–7]. Especially, the emergence of the living supramolecular polymerization (LSP) method provides an exciting and novel research avenue, which has been applied to establish supramolecular polymers with efficient modulating and uniformity of chain growth and dispersity[8,9]. To call a "living", in the strict sense, the growth of supramolecular polymers from the active ends should be demonstrated by multicycle experiments[10–14]. In 2014, Sugiyasu and Takeuchi's[15] group established the first seeded LSP via off-pathway aggregates. Inspired by this, several clever strategies were developed to encode the molecular information to achieve programmable LSP[16–18]. However, the further requirement for LSP systems is precise control over the degree of polymerization, chain stereochemistry, and lifetime[19]. Thus, in 2015, Aida and Miyajima's group first introduced the chain-growth mechanism to realize the initiator-controlled LSP[20]. This unique LSP provoked a paradigm shift in precision macromolecular engineering[21,22]. From the development milestones, LSP is still in its infant stage, and its ultimate goal is to acquire controlled life-like active materials[23–25]. In 2018, Balasubramanian and George's group demonstrated the first biomimetic LSP by consuming the chemical fuel ATP[26]. With the benefit of this strategy, various analogs of biological self-assembly motifs have been developed and bring us closer to complex biological entities. Even at this stage, scientists are still confronted with a severe challenge on the monomer design, because the aforementioned works are directed to specific systems, which require painstaking regulation and multi-step modification of the monomer structure[27–29]. This problem inevitably increases the synthetic difficulty and the material cost, greatly limiting their universality and application. To breakthrough the current bottleneck, it is an effective way to fabricate LSPs using simple and commercially available monomers. Nevertheless, this design strategy is hardly possible to succeed from an energy principle, because the activation barrier of simple monomers in the nucleation step is insufficiently high in comparison with that of the spontaneous nucleation to control the kinetics of subsequent elongation[15]. Therefore, the key to solving this problem lies in finding suitable synthesis methodologies to rationally select assembly pathways, so as to bring the supramolecular polymer with simple monomers to life.

The confinement space that can impact all chemical events taking place in a small cavity has been well documented in the field of nanoreactors, biosensors, and drug delivery vehicles, leading to a contrasting outcome than in the bulk[30,31]. In terms of self-assembly, there is evidence that confinement space can promote the formation and stability of self-assembled complexes held together by intermolecular interactions[32–37]. For example, in the cell-confined environment, the assembly and folding speed of the polypeptide chain is significantly accelerated[38]. In the molecular chaperone confined nano-cage, abnormal folding, and aggregation of protein can be inhibited, which in turn promotes the folding rate of normal proteins[39,40]. In the membrane-compartmentalized confinement environment, actin filaments, and cell microtubules are regulated by a finely tuned, highly complex molecular machinery[41]. These biological confinement phenomena have provided important clues to guiding the ordered assembly of simple monomers through the confinement approaches to strengthen the intermolecular interactions, so as to achieve the LSP dominated by kinetics control.

In this work, to validate our hypothesis, the confinement effect of the layered double hydroxide (LDH) nanomaterial is tried to guide the assembly of simple monomers by intercalating method[42–44]. Theoretically, after the removal of LDHs template,

the ordered assembly of guests will be destroyed because it has been reported that the activation barrier (30 kcal mol$^{-1}$) for a similar process, spontaneous initiation of styrene polymerization[40], is far beyond the energies associated with non-covalent bond formation. However, here, an unexpected phenomenon is discovered that the long-range anisotropic structure (metastable state) assembled by simple monomers cannot only be preserved after dissolving of LDH, but also elongate successfully upon addition of suitable solution. Kinetics and mechanistic studies reveal LDH overcomes a huge energy barrier to inhibit spontaneous nucleation of simple monomers and disassembly of metastable states, which provided the prerequisites of living polymerization. Owing to the living ends, two types of chiral monomers are added in growth steps to form block copolymer, by which the naked eye visualization of chiral recognition can be realized.

## Results and discussion

**Monomer selection.** Here, 8-hydroxypyrene-1,3,6-trisulfonate (solvent green 7, SG7) is chosen as a model to study the formation of LSP because it possesses a pyrene molecule bearing hydrogen-bonding moieties and negative charges, which is expected to self-assemble via π–π stacking of the pyrene planes and hydrogen bond (H-bond) of sulfonyl group. Importantly, there exist remarkable different fluorescence (FL) properties between monomer and various stacking states (Supplementary Figs. 1–2 for a detailed analysis), providing the prerequisite for naked eye visualization of assembly events.

**Growth process of supramolecular polymers.** Figure 1a depicts the formation process of LSP and seed-induced supramolecular polymer (SSP) from SG7 monomer. Upon intercalating SG7 monomer into the confinement space of LDH, named as SG7-LDH, which is confirmed by powder X-ray diffraction (XRD) measurements (step 1, Supplementary Fig. 3). The oriented arrangement of SG7 in the interlayer of LDH was achieved, as confirmed by FL anisotropy, where SG7-LDH showed a high anisotropic value ($r = 0.565$) compared with untreated SG7 powder ($r = 0.0201$) (Supplementary Figs. 4–5). The ab initio molecular dynamics (AIMD) simulations further testified the ordered SG7 with an orientation angle of $\theta = 9°$ ($\theta$ is the inclined angle between the principal axis of molecule and the elongation axis of oriented arrangement, Supplementary Fig. 6). After removal of LDH by using methanol/trifluoroacetic acid (CH$_3$OH/TFA) mixture solvent (5:3 v/v), the pre-assembly as the seed is obtained, which is a metastable and living supramonomer (LSM) (step 2). As confirmed by FL spectra and photos under UV light (Fig. 1b−c, Supplementary Figs. 7–8), compared with SG7 monomer ($\lambda_{em} = 430$ nm), the pre-assembly LSM ($\lambda_{em} = 460$ nm) showed red-shifted wavelength, indicating emergence of J-aggregate among LSM; compared with SG7-LDHs ($\lambda_{em} = 590$ nm), the pre-assembly LSM ($\lambda_{em} = 460$ nm) showed a blue shift attributed to shorter J-aggregate. On the other hand, the metastable properties of LSM can be proved by that LSM can maintain for ~10 min and spontaneously convert into the equilibrium state over time (detailed in Supplementary Fig. 9); living properties can be proved by that LSM can continue to grow to metastable LSP initiated by increasing poor solvents (TFA) until CH$_3$OH/TFA (1:3 v/v), that is, the step 3. This growth shows self-replication characteristic and formation of longer J-aggregate, confirmed by the nonlinear sigmoidal increase in temperature-dependent absorption (Supplementary Fig. 10) and emission red-shift from 460 to 535 nm (Fig. 1c, Supplementary Figs. 8 and 11). As shown in the FL intensity time scan (FITS), the metastable properties of LSP can be proved by its constant FL intensity and wavelength in the first 5 min (Supplementary Figs. 12–13).

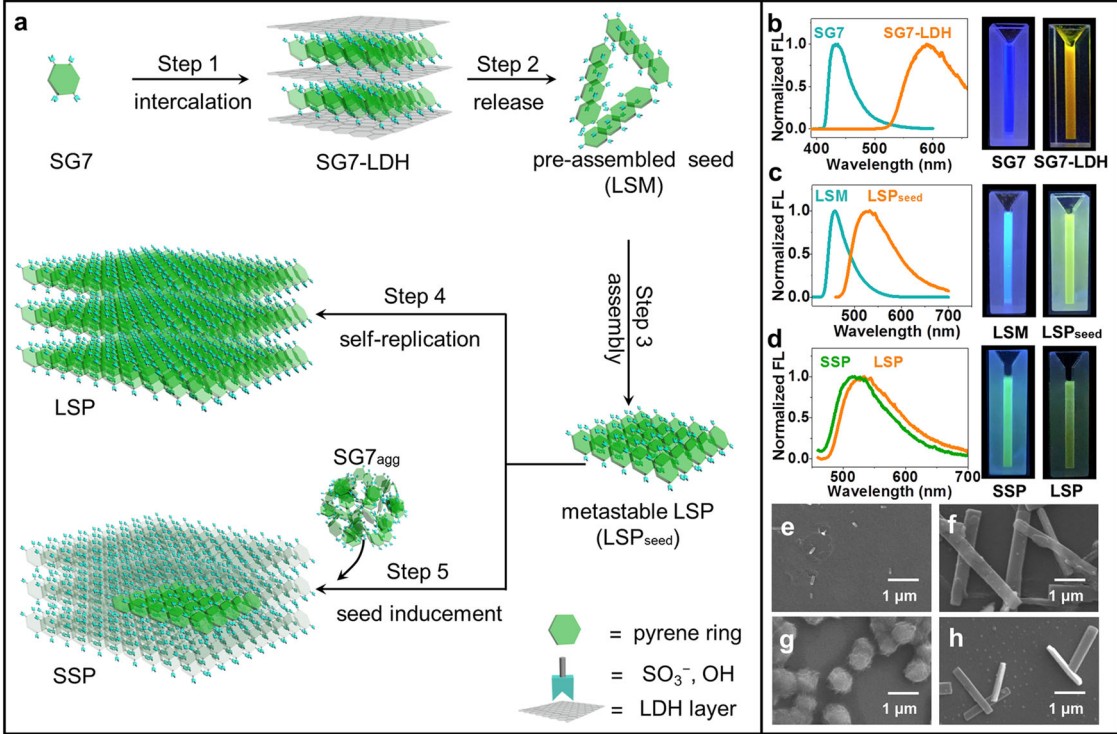

**Fig. 1 The growth process and characters of LSP and SSP. a** Schematic presentation for the formation of the LSP. Normalized FL spectra and photos under UV light of **b** SG7 and SG7-LDH in $CH_3OH$; **c** fresh LSM and metastable LSP ($LSP_{seed}$); **d** fresh SSP and LSP. SEM images of **e** metastable LSP **f** LSP after aging for 12 h **g** $SG7_{agg}$, and **h** fresh SSP.

The wide excitation range of LSP is also in line with the long-range π–π stacking rather than a single molecule (Supplementary Fig. 14). Step 4 is an energetically favored elongation step of metastable LSP. As confirmed by scanning electron microscopy (SEM) images and FITS (Fig. 1e−f and Supplementary Figs. 12 and 15), elongated LSP showed increased size and decreased FL intensity over time. Compared with metastable LSP, decreased FL intensity of elongated LSP in FL emission spectra (Supplementary Fig. 12) is attributed to the FL quenching by elongation; unchanged wavelength ($\lambda_{em} = 535$ nm) is attributed to the maintenance of J-aggregate. Step 5 is a seed-induced living polymerization by adding inactive $SG7_{agg}$ (SG7 dissolved in $CH_3OH$/TFA (1:3 v/v), 2.5 mM) into metastable LSP ($LSP_{seed}$) to form SSP. In FL emission spectra (Supplementary Figs. 16–18), the formed SSP ($\lambda_{em} = 525$ nm) shows redshift compared with $SG7_{agg}$ ($\lambda_{em} = 450$ nm), indicating the rearrangement of $SG7_{agg}$ induced by $LSP_{seed}$. In addition, SSP shows an emission color change and blue shift of 10 nm compared with LSP (Fig. 1d), attributing to the tiny difference in newly formed J-aggregate compared with original $LSP_{seed}$. In SEM images (Fig. 1g–h and Supplementary Fig. 19), the transformation of morphology from spherical ($SG7_{agg}$) to rectangular (SSP) and increased size of SSP in cycle experiments also confirm the living polymerization induced by LSP. In addition, this system is reversible and can be prepared in low concentration (detailed in Supplementary Figs. 20–24). None of the products is crystal (Supplementary Fig. 25). The above results fully prove that this assembly process belongs to supramolecular polymerization.

It is worth mentioning that the SG7 is not only intercalated in the cavity of LDH, but also adsorbed on the outer surface of the LDH. To quantitatively describe the effect of SG7 adsorption on assembly, we employed the $CO_3$-$LDH_{20}$ (the interlayer anions of LDH is $CO_3^{2-}$) to adsorb SG7 (marked as $SG7$-$LDH_{20}$-surface, detailed in supplementary information), where the intercalation does not occur because the affinity between $CO_3^{2-}$ and LDH layer is much greater than that between organic simple molecules and LDH layer[45,46]. ICP-MS and elemental analysis results show that the adsorption capacity is very low and only accounts for 13.41% of the total SG7 (Supplementary Table 1–2). The polarized FL profiles in Supplementary Figs. 4 and 26, SG7-$LDH_{20}$-surface ($r = 0.171$) shows much lower anisotropic value than that of SG7-$LDH_{20}$ ($r = 0.565$). In addition, the adsorbed SG7 did not assemble into LSP, as confirmed by SEM images and FL spectrum. SEM showed that a SP with rectangular morphology can be obtained, but it fails to induce following cycle products (Supplementary Fig. 27), indicating that $SP_{20}$ is not a kind of LSP. Instead, the appearance of depolymerization of $SP_{20}$ in cycle process, as certified by decreased FL intensity of cycle products of $SP_{20}$ (Supplementary Fig. 28). To sum up, the adsorbed SG7 did not assemble into LSP owing to lack of ordered arrangement. Importantly, the appearance of adsorbed SG7 did not affect the formation of LSP, as proved by physically mixing SG7-$LDH_{20}$-surface and SG7-LDH to perform cycle experiments. The cycle products showed rectangular morphology with increased size area in SEM and FL intensity, indicating the cycle products have the ability for living assembly (Supplementary Figs. 27–28).

Generally, the presence of inorganic ions during the assembly process will affect the result of the assembly[47,48]. In our system, in the process of dissolving LDH, metal ions ($Mg^{2+}$ and $Al^{3+}$) are introduced. In order to eliminate their influence, we designed a series of comparative samples treated by the same process as an assembly of SG7-LDH, including SG7 powder, Cl-LDH, and SG7+LDH (physically mixed SG7 and Cl-LDH). Both SG7 and SG7+LDH can only form isotropic spherical structures as shown in SEM image (Supplementary Fig. 29a, c). For Cl-LDH, SEM image shows irregular morphology of metal salt (Supplementary Fig. 29b). These results indicate that the ion of dissolved LDH is not the key to facilitate rectangular morphology of LSP.

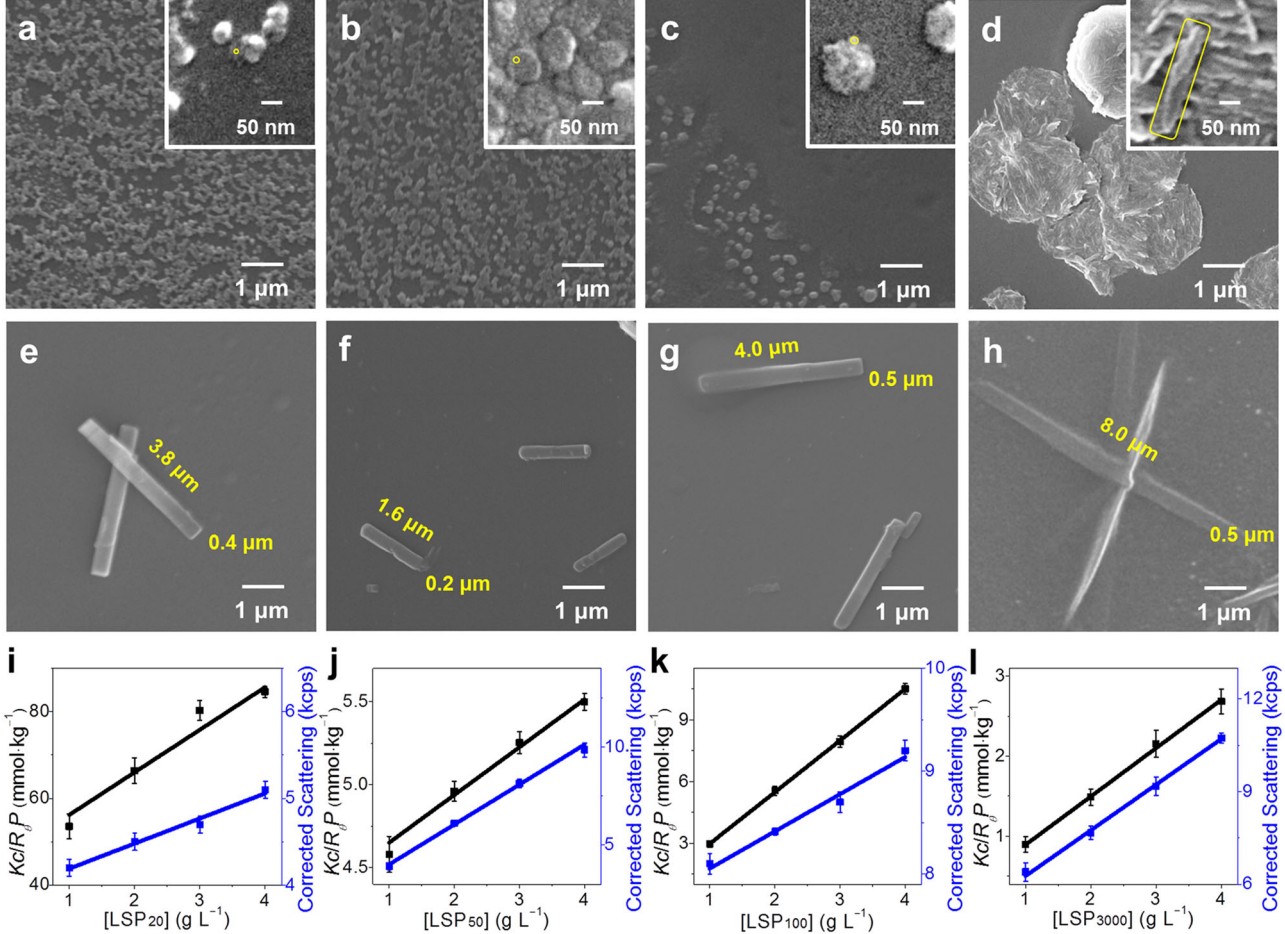

**Fig. 2 Tuning the size of LSP via LDH confinement. a–d** SEM images of metastable LSM prepared from SG7-LDHs with different size: **a** 20 nm, **b** 50 nm, **c** 100 nm, and **d** 3 μm (insets are corresponding magnifications and the sizes of metastable LSM in the insets are marked with a yellow line). **e–h** SEM images of elongated LSP from the corresponding metastable LSM in **a–d**, respectively. **i–l** Debye plot of metastable LSP made of corresponding metastable LSM in **a–d**, respectively. Error bars are calculated using the standard error formalism for the data of three replicate experiments.

**Relationship between outcome and LDH confinement effect.**
According to the properties of the kinetic and non-equilibrium system, the formation pathway and outcome of metastable LSP should be controlled by different preparation protocols[11]. Thus, we tried to change the confinement size by tuning LDH size. As the size of LDH has great controllability and will not be significantly affected by the intercalation process of SG7 (Supplementary Figs. 30–31), it provides the prerequisite for the size regulation of SG7 LSM[49,50]. It should be pointed out that for nanoscale structures, aggregation cannot be completely avoided[51,52]. Thus, in order to observe accurately, we etch part of the LDH to expose the metastable LSM before aggregation, that is, the metastable LSM is still on the etched fragments of LDH. As confirmed by SEM images (Fig. 2a–d and their insets), the sizes of metastable LSMs gradually increase with the raising of LDH sizes from 20 nm to 3 μm (as-prepared LSMs from different size LDHs are named as metastable $LSM_{20}$, $LSM_{50}$, $LSM_{100}$, and $LSM_{3000}$, respectively). Note that, metastable LSMs easily aggregate to its stable state confirmed by SEM, preventing following transformation to LSP (Supplementary Figs. 9 and 32). Thus, the increase of poor solvent TFA can initiate metastable LSMs to continue to grow into metastable LSPs (named as metastable $LSP_{20}$, $LSP_{50}$, $LSP_{100}$, and $LSP_{3000}$, respectively). The highly ordered structure of metastable $LSP_{20}$–$LSP_{3000}$ can be confirmed by high FL anisotropic values ($r > 0.45$, Supplementary Fig. 33). As shown in SEM images, the sizes from metastable $LSP_{20}$ to $LSP_{3000}$ increase

sequentially (Supplementary Fig. 34). During their formation, the decline rate of FL intensity and redshift are different from each other, indicating that the kinetics of transformation depends on confinement space owing to pathway complexity (Supplementary Fig. 35). The pathway not only influenced the rate of primary nucleation of metastable LSP but also in energy, confirmed by their different emission wavelength at 525, 510, 516, and 523 nm, respectively) ($\lambda_{ex} = 370$ nm, Supplementary Figs. 35–36). In addition, metastable $LSP_{20}$–$LSP_{3000}$ had different activity in elongation during the relaxation to the energetically favored state, confirmed by the different area of elongated $LSP_{20}$–$LSP_{3000}$ (1.50, 0.32, 2.0, and 4.0 μm$^2$ in Fig. 2e–h). The activity of metastable LSP will be discussed in the following cycle experiments.

To evaluate the polydispersity index (PDI) and degree of polymerization (DP), we, respectively, studied SEM images and static light scattering of metastable LSP in three replicate experiments (Fig. 2i–l and Supplementary Fig. 34). The number-average area ($A_n$), weight-average area ($A_w$), and PDI ($A_w/A_n$) of $LSP_{20}$–$LSP_{3000}$ were obtained by evaluating over 50 objects in SEM images (detailed in Supplementary Table 3). According to Rayleigh equation[53], the DP of the metastable $LSP_{20}$, $LSP_{50}$, $LSP_{100}$, and $LSP_{3000}$ is ~40, 400, 4000, and 6000 (Supplementary Table 4–5), respectively. Considering the size extension over time (Supplementary Fig. 15), in fact, the DP of ultimately obtained LSP is much higher than these values. Compared with previous reports, the DP of LSP proposed in our

work has made an exciting breakthrough (Supplementary Table 6).

**Relationship between outcome and formation pathway.** In addition, the conventional preparation protocol have also been employed to testify the pathway selection of the metastable LSPs (taking the $LSP_{20}$ an example) by changing experimental parameters, such as timing of addition of the bad solvent and mechanical agitation.

The size of metastable $LSP_{20}$ is dependent on the timing of addition of TFA to SG7-LDH (4 mg in 200 μL $CH_3OH$) during self-assembly. In detail, Condition (1) 0 min with 600 μL; Condition (2) 0 min with 200 μL and 30 min with 400 μL; Condition (3) 0 min with 120 μL and 30 min 480 μL; Condition (4) 0 min with 120 μL and 24 h with 480 μL (Fig. 3a). As seen in the time-dependent FL spectra (detailed in Supplementary Figs. 9, 35–37), metastable $LSP_{20}$ in Condition (1–3) appeared at 20 min, 45 min, and 48 min, respectively, indicating the decreased rate of nucleation, and consequently its rate. SEM images showed the decreased size of metastable $LSP_{20}$ (0.04 $μm^2$, 0.006 $μm^2$, and 0.0036 $μm^2$, Supplementary Fig. 38a–c)). In condition (4), metastable $LSM_{20}$ failed to transform to $LSP_{20}$ because it has access to its energetically favored state before being kept in $CH_3OH$/TFA (1:3 v/v) (Supplementary Figs. 9 and 38d). Thus, structurally different metastable assemblies under the same final conditions can be obtained by changing the timing of the addition of TFA.

The size of metastable $LSP_{20}$ is also influenced by mechanical agitation. As seen in confocal laser scanning microscope image, metastable $LSP_{20}$ without ultrasound move vigorously before 10 min, but severe stack can be observed as time increase (Supplementary Figs. 39–40 and Movie 1). In contrast, mechanical agitation can create smaller structures with fully exposed active sites as initiators for living polymerization. When $LSM_{20}$ is sonicated for 1 h, the as-obtained product (metastable $LSP_{20}$) shows uniform distribution of area (Fig. 3b). After being kept for 12 h, kinetics capture extends the area from 0.03 $μm^2$ to 1.50 $μm^2$ (Supplementary Fig. 15), while retaining uniform characteristics (Fig. 3c). Importantly, a competitive growth of elongated $LSP_{20}$ (12 h) at anisotropic active sites can be observed. The average width increased by 200%, whereas the length only increased by 48% by prolonging ultrasound time (Fig. 3d and Supplementary Fig. 41).

**The kinetics behavior of SSP.** The characteristic kinetics evolution of metastable $LSP_{20}$ offers the possibility of acting as reactive seeds (defined as $LSP_{seed}$) to provoke a rapid growth or transformation of $SG7_{agg}$ to be SSP. Figure 4a−h and Supplementary Figs. 42−44 strongly suggest that the transformation from spherical inactivated $SG7_{agg}$ to rectangular SSP can be initiated by the addition of "seeds" ($LSP_{seed}$, step 5 in Fig. 1). Previous seed-induced polymerization acquired mixing diluted monomer stock solution with a seed suspension to prevent spontaneous aggregation of the monomer[54–56]. In contrast to these protocols, $SG7_{agg}$ in our system is inactivated spherical aggregates (Supplementary Fig. 29), which can join in the formation of SSP owing to the "infection" of $LSP_{seed}$. Compared with the metastable $LSP_{50}−LSP_{3000}$, metastable $LSP_{20}$ is the most superior in seeds experiment because $SSP_{20}$ shows regular increased size in cycle experiments (Supplementary Fig. 19), whereas $SSP_{50}−SSP_{3000}$ split into small fragments (Supplementary Fig. 45).

During the seed-induced polymerization, the area of $SSP_{20}$ can be controlled by regulating $SG7_{agg}$ concentration (Fig. 4a) or the ratio of $SG7_{agg}$ to $LSP_{seed}$ (Fig. 4b). The maximum area of $SSP_{20}$

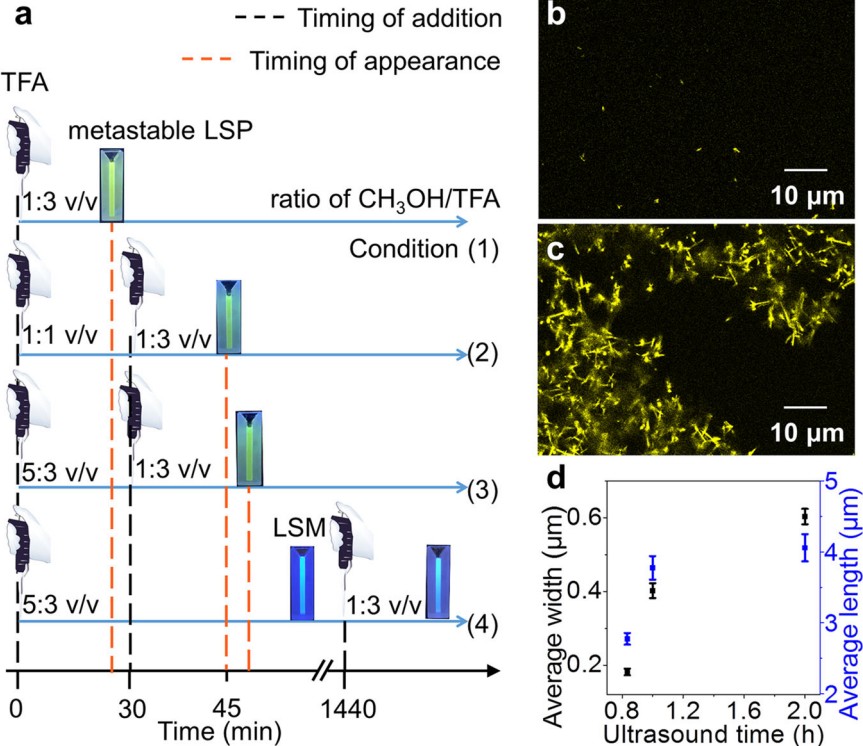

**Fig. 3 Tuning the size of LSP via solvent processing and mechanical agitation. a** Schematic illustration for condition (1–4) to study different kinetic assemblies via changing the timing of addition of TFA (0, 30, and 1440 min). The picture of micropipette in **a** is adapted from bio instruments template in ChemDraw. **b–c** CLSM images of: **b** metastable $LSP_{20}$ and **c** elongated $LSP_{20}$ for 12 h from **b**. **d** Average size of elongated $LSP_{20}$ prepared by different ultrasound time. All error bars are calculated using the standard error formalism for the data of three replicate experiments.

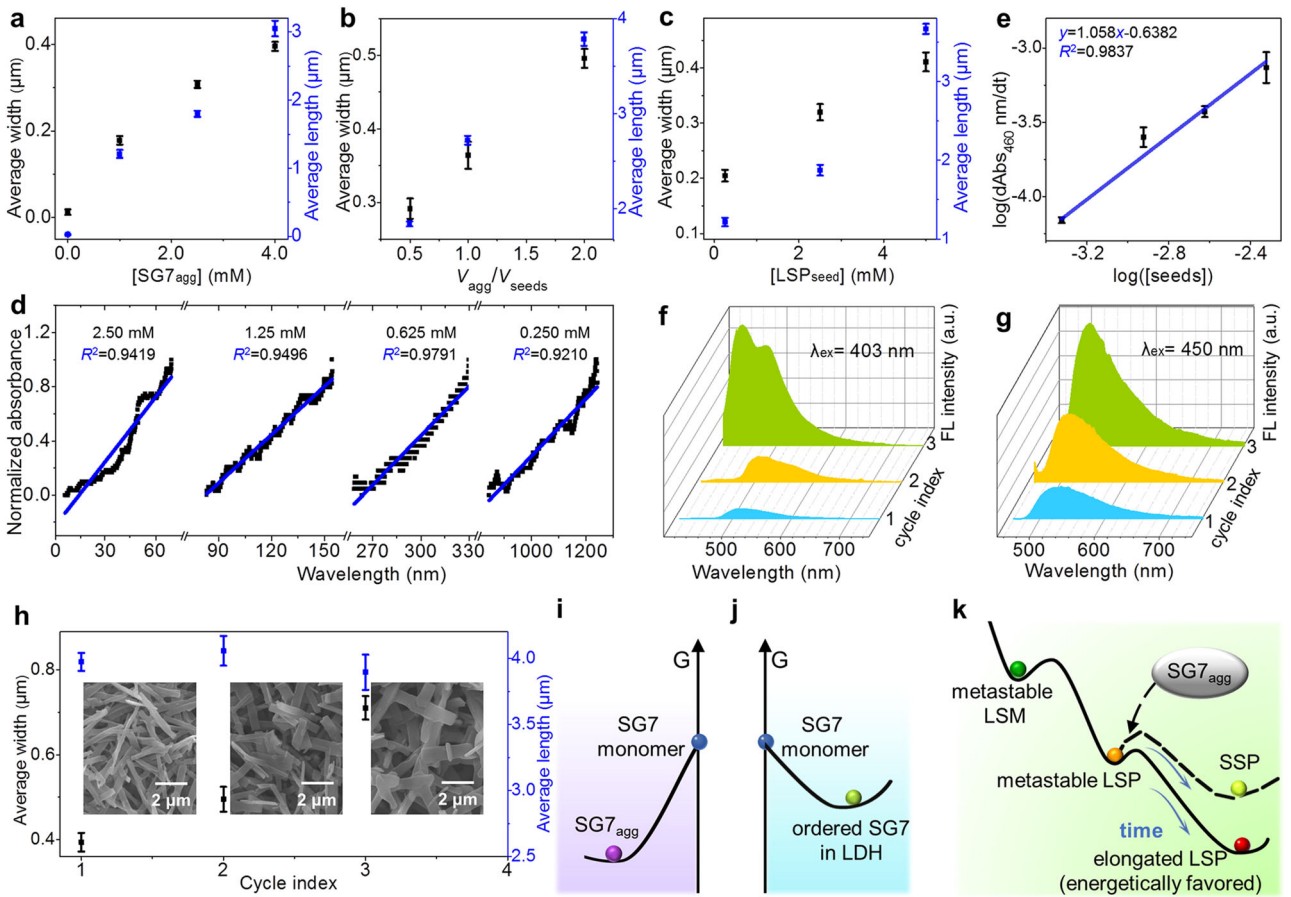

**Fig. 4 Kinetics behavior and living supramolecular polymerization of SSP. a–c** The size of SSP as a function of **a** the concentrations of $SG7_{agg}$, **b** the volume ratio of added $SG7_{agg}$ to the $LSP_{seed}$, **c** the concentrations of $LSP_{seed}$. **d** Log-log plot of the rate of increased absorbance at 460 nm as a function of the $LSP_{seed}$ concentration. **e** Time scan of absorbance of $SSP_{20}$ prepared by $LSP_{seed}$ with different concentrations. **f–g** FL spectra of $SSP_{20}$ obtained in Cycle 1–3. **h** The size of $SSP_{20}$ as a function of cycle number $x$ ($x = 1, 2,$ and 3) (insets: corresponding SEM images of $SSP_{20}$ obtained in cycle 1–3, respectively). Schematic illustration for the energy diagrams of **j** SG7 monomer and $SG7_{agg}$, **k** SG7 monomer, and SG7-LDH, together with **l** the products during the formation of LSP and SSP. All error bars are calculated using the standard error formalism for the data of three replicate experiments.

can reach ~1.95 $\mu m^2$ (Supplementary Figs. 42–43). Similarly, the $SSP_{20}$ growth can also be controlled by changing the concentration of $LSP_{seed}$, which is realized by adjusting the initial concentration of SG7-LDH (Fig. 4c). The area of fresh $SSP_{20}$ can reach ~2.22 $\mu m^2$ (Supplementary Fig. 44). This kind of growth characteristics related to initial feed parameters has been observed experimentally by Kazunori Sugiyasu et al. in their study of regulating SSP through self-assembly of porphyrin derivative[57].

During the living elongation process, the replicator $r$ can be calculated by Eq. (1). Where $F$ is a food molecule, $R$ is a replicator, $k_R$ is the rate constant, $f$ and $r$ are the order of the replication process in food and replicator, respectively.

$$\log \frac{d[R]}{dt} = \log k_R + f \log[F] + r \log[R] \qquad (1)$$

As the concentration of $LSP_{seed}$ increases from 0.250 to 2.50 mM, the time of $LSP_{seed}$ beginning to grow decreases from 800 to 5 s (Fig. 4d). Following the addition of $SG7_{agg}$ to different amounts of $LSP_{seed}$, polymerization proceeded linearly with respect to the reaction time, confirming the living elongation nature (Fig. 4e). The logarithm of the apparent polymerization rate of SSP, $\log(d(Abs_{460})/dt)$, is proportional to the logarithm of $LSP_{seed}$ concentration, $\log([seeds])$, with a slope of 1.058 (Fig. 4e). Thus, within the experimental error, the polymerization reaction is close to first order and appears to be capable of exponential

replication, which is one of the characteristics of living polymerization[58].

**Living supramolecular polymerization.** The FL and SEM spectra were also used to monitor the living nature of seed-induced polymerization (Fig. 4f–h). The metastable $LSP_{20}$ and fresh $SG7_{agg}$ were freshly prepared and mixed in equal volumes, followed with ultrasound for 1 h. In Cycle 1, after two kinds of solutions were mixed, the blue light attributed to $SG7_{agg}$ disappeared immediately, and the yellow light attributed to $SSP_{20}$ appeared (Fig. 4g). Further addition of the $SG7_{agg}$ stock solution to the resulting solution repeats the polymerization growth (1:1 v/v, Cycle 2 and 3), but with slower rates. This is consistent with the fact that the initial concentration of $LSP_{seed}$ is diluted by half in every cycle. In Cycle 2, it took 30 min for blue light to disappear. In cycle 3, the blue light did not completely disappear until it is extended for 60 min. In general, as living polymerization proceeds, concentrations of $LSP_{seed}$ and $SG7_{agg}$ decrease, and energy barrier of $SSP_{20}$ increases, leading to no longer negligible depolymerization (Fig. 4f). Therefore, the area of freshly prepared $SSP_{20}$ in Cycle 1 reaches 0.54 $\mu m^2$, while the length of final elongated $SSP_{20}$ shows unobvious change after Cycle 2 and 3 (Fig. 4h). Unexpectedly, this growth is directional-selective with a large increase in width from 300 to 600 nm (Fig. 4h), indicating highly ordered $LSP_{seed}$ shows fantastic directional rearrangement effect on $SG7_{agg}$.

This can be further testified by FL intensity increase and blue shift of SSP (Fig. 4g) with the increase of cycle index owing to the change of stacking direction from along the long axis to the short axis. Oversized metastable LSP with highly ordered structure has an indispensable role in the living polymerization, which solves the problem of the insufficient energetic barrier by jumping across the spontaneous nucleation. This unusual growth mode gives LSP a great advantage in terms of area compared with previous SSP reports (Supplementary Table 6).

**The energy diagram of the growth process.** The above experiments have fully proved the mechanism of the transformation from metastable LSM to LSP to be energetically favored. This beneficial elongation behavior derives from the huge energy barrier overcome by LDH, confirmed by theoretical calculation and optical information. As proved by theoretical calculation (Supplementary Fig. 46, detailed information for model construction is listed in the methods section), the locations of global minima and local minima are independent of the number of SG7 molecules, the energies of SG7 monomer, $SG7_{agg}$ and ordered SG7 in LDH can be calculated and compared by using the representative models composed of eight SG7 molecules, as displayed in Fig. 4i–j. The energy of SG7 monomer is calculated to be $167.02 \, kJ \, mol^{-1}$ higher than that of $SG7_{agg}$ and $126.92 \, kJ \, mol^{-1}$ higher than that of ordered SG7 in LDH (Supplementary Fig. 46). The driving force of incorporating SG7 into LDH is mainly Coulomb force and H-bond. Thus, the energy level of SG7 solution should be higher than that of thermodynamic equilibrium $SG7_{agg}$ and ordered SG7 in LDH. In addition, the conversion from $SG7_{agg}$ ($\theta = 35°$) to ordered SG7 in LDH ($\theta = 9°$) needs to overcome an energy barrier of $70.55 \, kJ \, mol^{-1}$ at $\theta = 18°$ (the maximum in potential energy surface from 35° to 9°). Thus, ordered SG7 in LDH ($\theta = 9°$) is kept in an energy well to be a metastable state.

With the increased number of SG7 molecules, calculating the energy level of the following transformation needs unaffordable computational cost, which is beyond the state of art. Thus, the schematic illustration for the energy level of metastable LSM, metastable LSP, elongated LSP, and SSP is presented according to the optical information (Fig. 4k). During the transformation, metastable LSP ($\lambda_{em} = 525 \, nm$) has lower energy than metastable LSM ($\lambda_{em} = 470 \, nm$), confirmed by the redshift in Supplementary Fig. 35. After then, the energy of elongated LSP and SSP are lower than that of metastable LSP owing to the energetically favored elongation step. The decreased emission intensity and redshift of LSP absorbance demonstrate the augment of J-agg owing to the formation of huger H-bond net (Supplementary Figs. 12 and 47–49). The H-bond net can stabilize LSP structure confirmed by the Fourier-transform infrared spectra (Supplementary Figs. 50–53) and solvent-related depolymerization behavior (Supplementary Fig. 54). The energy of the SSP is slightly higher than elongated LSP (step 5), confirmed by a blue shift ($\Delta\lambda = 10 \, nm$) of FL spectra (Supplementary Figs. 11 and 16).

**Application in chiral recognition.** The different affinity of chiral molecules during the supramolecular polymerization process endows metastable LSM the possibility of chiral recognition[59–61]. Based on the relationship between structure and luminescence property, the naked eye visual recognition can be realized by the lag time of the chiral recognition products (named as LSM+L or LSM+D). The addition of L- or D-arginine (L- or D-Arg) during the transformation from metastable $LSM_{3000}$ to LSP leads to the formation of corresponding polymers (defined as polymer-$L_{3000}$ and polymer-$D_{3000}$), which show a significant difference in the lag time (Fig. 5a–b). The lag time required for the formation of polymer-$L_{3000}$ is 50 min, accompanied by the solution

color change from blue to yellow, whereas the formation of polymer-$D_{3000}$ needs longer time up to 72 h (Fig. 5a, b and Supplementary Figs. 55–57). The mechanism of the chiral recognition is that the presence of different competitiveness units exerts a strong and different retardation during the assembly kinetics, where –NH$_2$, –OH, and –COOH on L- and D-Arg have different stereo conformation and binding energy (Supplementary Fig. 58). The theoretical calculation shows that the binding energy between metastable $LSM_{3000}$ with SG7 ($-52.64 \, kJ \, mol^{-1}$) is weaker than that of $LSM_{3000}$ with L-Arg ($-191.50 \, kJ \, mol^{-1}$) and $LSM_{3000}$ with D-Arg ($-471.64 \, kJ \, mol^{-1}$), providing basic conditions for co-assembly (Supplementary Figs. 46 and 58). Further, the difference in the binding energy between L-Arg and D-Arg gives $LSM_{3000}$ the ability to recognize the Arg type (Fig. 5c, d)[62]. To verify the reproducibility of chiral recognition, three batches of metastable $LSM_{3000}$ samples were prepared, and the results showed that the lag time for the formation of corresponding polymer-$L_{3000}$ was 50 min (Supplementary Fig. 59). Importantly, the chiral recognition effectiveness from metastable $LSM_{20}$, $LSM_{50}$, and $LSM_{100}$ and $LSM_{3000}$ is quite distinct (Supplementary Figs. 60–62 and the detailed analysis). For the smaller metastable $LSM_{20}$ and $LSM_{50}$ with higher activity, chiral recognition is difficult to realize due to indistinguishable lag time. For metastable $LSM_{100}$, the identification can only be seen within 20 min. Thus, the higher recognition ability of metastable $LSM_{3000}$ maybe origin in relatively maximized kinetics effects and minimal activity.

**Universality.** As proof of universality of our proposed preparation method for LSP, various simple molecules with aromatic rings and negative charge (Fig. 6) were chosen to orderly array in confinement space of LDH (Supplementary Figs. 63–77) and further performed the same assembly process as SG7 (Supplementary Figs. 78–83). By the assembly results, the basic principles are summarized as follows: (1) the formation of H-bond net among –NH$_2$, –HSO$_3$, and –OH. For example, the benzene-sulfonic acid (BSA) with insufficient active H-bond cannot form H-bond net (Supplementary Figs. 64, 67, and 70), leading to the failure in formation of LSP (Supplementary Fig. 79). In contrast, the appearance of multiple active H-bonds easily forms the LSPs (e.g., 3-aminobenzene sulfonic acid; 2,5-diaminobenzenesulfonic acid; Congo red, CR); (2) oriented long-range π-π stacking between neighboring molecules. For example, the 6,7-dihydroxynaphthalene-2-sulfonate (DHNS) shows the failed formation of LSP (Supplementary Fig. 82). The reasons originate from that although the appearance of π-π stacking of DHNS molecules in the interlayer of LDH as confirmed by UV spectrum (Supplementary Fig. 65), the π-π stacking is the negligible long-range ordered arrangement as confirmed by a low FL anisotropic value ($r = 0.0214$) (Supplementary Fig. 68). AIMD simulations, which are referred to XRD, ICP-MS, and elemental analysis results (Supplementary Fig. 63, Table 7), also further proved the adjacent DHNS molecules in the interlayer of LDH do have π-π stacking in short range, attributing to three functional groups and the large aromatic ring of DHNS. However, they show a disordered arrangement in the long range (Supplementary Fig. 73). Therefore, our approach provides a good alternative to form oversized LSP by simple molecules.

In conclusion, we propose a universal method for establishing LSP with simple molecules. The molecules equipped with functional groups to access cooperative intermolecular interactions (π-π stacking and H-bond, etc.) are monomer candidates for LSP. In our proposed protocol, the simple molecules in LDH confinement undergo newly orientated long-range π-π stacking and was kept in the energy well, which overcomes a huge energy

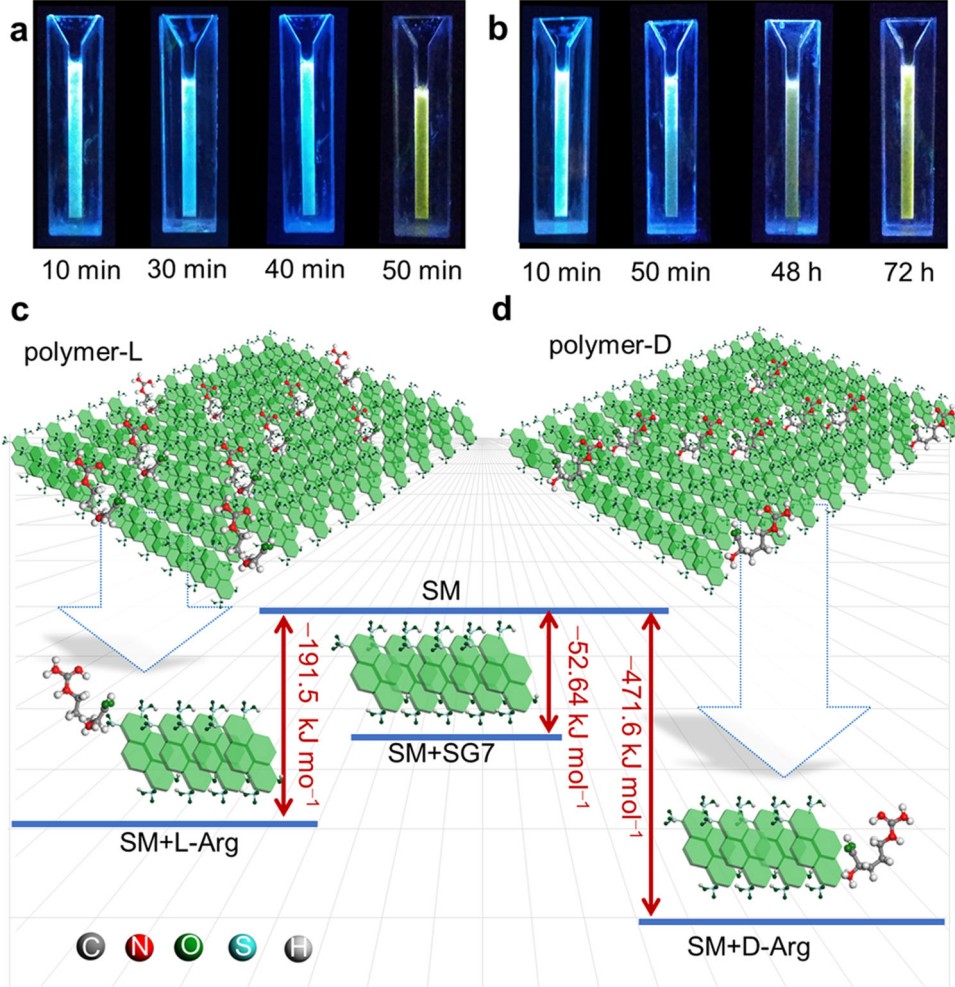

**Fig. 5 Application in the chiral recognition.** Photos under UV light of chiral recognition products of metastable LSM$_{3000}$ to **a** L-Arg and **b** D-Arg during co-assembly process, noted that only ultrasound for 50 min. **c–d** Optimized geometries of metastable LSM$_{3000}$ to **c** L-Arg and **d** D-Arg.

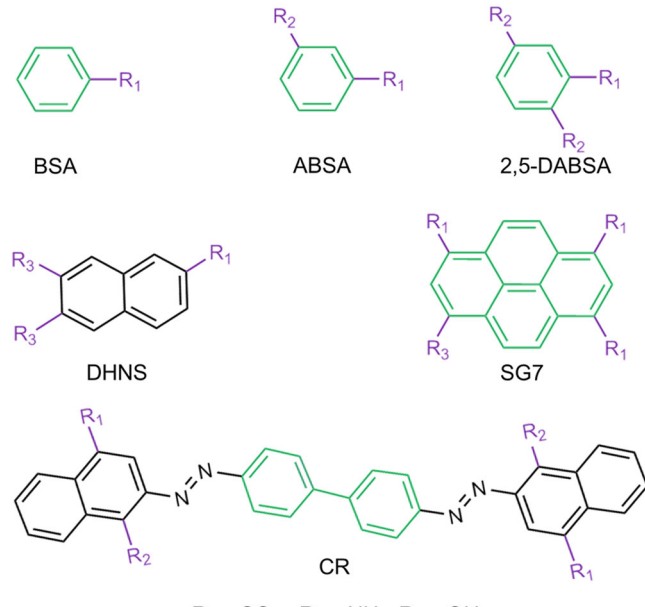

**Fig. 6 Universality.** Various molecules were chosen in our proposed method. All chemical structures are obtained in ChemDraw.

barrier to inhibit spontaneous aggregation of monomer and depolymerization of metastable state. The living assembly and seed-induced growth provide activity and length control, as well as narrow dispersity of LSP. Owing to the competitive growth at anisotropic active sites, the largest DP reported so far (~6000) has been obtained. The distinct living abilities from LSP$_{20}$−LSP$_{3000}$ provide the possibility of control according to actual requirements. We affirm that our system definitely is not a special case. A similar LSP can be established by other molecules, which can also induce the transformation from inactivated aggregates to ordered rearrangements. The living character endows LSP with explicit temporal control in complex dynamic programming, which should be a significant step toward the further development of chiral amplification, copolymer formation, and artificial biomaterials. It is anticipated that this strategy will broaden the spread of monomer candidates, leading to more possibilities for LSP in various domains, such as autonomous materials and regenerative medicine.

## Data availability

All data supporting the findings are available in the article as well as the supplementary information files and from the authors on reasonable request.

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

## Acknowledgements

This work was supported by the National Key Research and Development Program (2019YFC1906100), National Natural Science Foundation of China (21974008, 21571014, 21656001, and 21521005), and the Fundamental Research Funds for the Central Universities (12060093063).

## Author contributions

W.S. and C.L. conceived and directed the project. Y.Z. carried out the experimental work. S.X. carried out the theoretical calculation. All authors co-wrote the manuscript. Y.Z. and S.X. contributed equally to this work.

## Competing interests

The authors declare no competing interests.
