## [Peer Review File · Nature Communications]

REVIEWER COMMENTS

Reviewer #1 (Remarks to the Author):

Comments on NCOMMS-20-26877

Title: Biomimetic Confinement-Driven Supramolecular Polymerization with Simple-Structured Monomers: Living It up

Authors: Wenying Shi and Chao Lu et al

Summary:

This manuscript describes a unique strategy to modulate the pathway of supramolecular polymerisation, taking advantage of the confined space of layered double hydroxide (LDH) to stabilize the metastable state of simple structure monomers. The metastable state of aggregated monomers could remain even after the dissolution of LDH, rendering this system with the ability for a living supramolecular polymerization (LSP). Although the premise of the manuscript is novel and fascinating, the authors unfold their story in a confusing way. Results were presented in the main text and the supporting information but often detailed explanations are missing. This will make readers very confused. Also, the quality of the data is not up to the standard at this stage. In summary, this manuscript appears to be an excellent candidate for publication in Nature Communications but much improvement is needed.

Comments

1. The title of the manuscript seems not appropriate. Layered double hydroxide (LDH) of $Mg(NO_3)_2$ and $Al(NO_3)_3$ is a confined nanospace for sure, while the self-assembly in a confined space does not definitely correlate with the biomimetic term. The author should reconsider the title of the manuscript.
2. The full forms of several abbreviated scientific terms, eg. FITS, CLSM, TFA, SSP etc, should be clearly mentioned in the main text where the abbreviation first appears.
3. The authors should provide some evidence to confirm that the only the MSS intercalates and self-assembles in the cavity the LDH, but not adsorb on the outer surface of the LDH. Alternatively, some explanation is need of the reasons why the adsorbed MSS did not assemble or did not affect the formation of metastable SM to a significant extent.
4. The stability of the metastable SM in step 2, right after the removal of LDH by using CH_3OH/TFA , should be examined, for example by ageing the solution for at least one week.
5. In many places, the authors commented without giving any explanation or sufficient experimental data support. For example, on page 5, line 105: "The effect of ion on this result during the dissolution of LDH can be ruled out by control experiment (Supplementary Fig. 2)". The author should provide a comparison of SEM images, i.e. before and after mixing the SG7 to the LDH in CH_3OH/TFA (1:3 v/v). Again, in Page 6, line 111: "6,7-dihydroxynaphthalene-2-sulfonate (DHNS), the formation of π - π stacking is hardly possible owing to its orientation angle (Supplementary Fig. 8)". Solely based on these SEM images, how could the authors conclude that a molecule has negligible/no π - π stacking interactions? More information and explanation is needed.
6. Page 6, line 119: " SG7 is chosen as a model because its favorable optical properties allow naked eye visualizing of assembly events". To support this author only provides the spectroscopic data in the SI without any explanation. More information is needed.
7. The scale bar of both sets of SEM images in Figure 2 is different, which makes it difficult to appreciate the increased length and width of the living supramolecular polymeric structures. Can the authors provide the SEM images for both sets with an identical scale bar?
8. Throughout the manuscript, in all the SEM images the fibers appear in agglomerate and interconnected state. Because it is hard to clearly see the fibers discretely, the corresponding statics on their length and width profiles will be less reliable or accurate. Can the authors show examples of more diluted fibers to a better display of their length and width?
9. The LSP seed was prepared by sonication of metastable LSP. In addition to the duration, the temperature of sonication should be also reported in the supporting information.
10. In the last part of the result and discussions, the authors demonstrated that 'The different affinity of chiral molecules during supramolecular polymerization process endows metastable SM the possibility of chiral recognition'. However, as both the monomer and the confined space are achiral, what is the origin of the different affinity of chiral molecules with metastable SM? Is it consistent that Polymer-L3000 always has a shorter lag time than Polymer-D3000, or does the

order of lag time varies with different batches? Results of repetitive tests should be presented.
11. The referencing is generally OK but a very early paper on LSP that should be considered for citation as it predates many of the papers referenced is Robinson et al Chem. Comm. 2015, 51, 15921-15924.

Reviewer #2 (Remarks to the Author):

This manuscript reports living supramolecular polymerization (LSP) by arresting the metastable state of the monomer using confinement in a commercially available LDH. They demonstrate well controlled supramolecular polymerization of simple molecules to produce supramolecular polymers with controlled chain length and low dispersity. The confinement approach is indeed interesting and should make lasting impact in the field. I recommend publication after addressing following issues:

The paper is written in rather complex manner which is difficult to read at times. For example, in Fig 1, authors show several molecules which may not be required. Author could the main molecule and discuss the results while at the end to demonstrate general applicability, they may show other structures and results in the subsequent discussion.

I am not at all convinced whether confinement in LDH should be so much compared with biomimicking! This is a well-known thing that molecules intercalate in such inorganic layered structures or even clays. Those should not be compared with biomimicking.

Referencing should be balanced, recent examples on LSP and particular review articles should be included.

Reviewer #3 (Remarks to the Author):

Summary of my review. This manuscript reports the controlled self-assembly of small charged molecules. I agree with the latter part of this manuscript on the seeded growth of the assemblies. However, most of the main claims are not supported. Furthermore, it is difficult to understand the contents of this manuscript because the authors mixed up their assumptions (or imaginations) and experimental results. Hence, I recommend the authors to fully revise the manuscript and submit to other journals such as scientific report.

Regarding on the definition of supramolecular polymerization;

(1) In this manuscript, the authors controlled the self-assemblies of small organic molecules with their developed method using LDH. They considered these assembling processes as supramolecular polymerization. However, it is just crystallization rather than supramolecular polymerization. In particular, the authors named their monomers as MSS (monomer with simple structure). I don't understand why they need to emphasize their monomers by using such special abbreviation. I guess the authors thought that their monomer designs were different from monomers used for the conventional supramolecular polymers. Simply, the assemblies, reported in this manuscript, are not supramolecular polymers and, as a result, the monomer designs are different. I recommend not to describe the reported self-assembling processes as supramolecular polymerization.

(2) Even if the reported self-assemblies were considered as supramolecular polymerization, LDH or biomimetic confinement did not induce nor promote supramolecular polymerization. The authors should reconsider the title "Biomimetic confinement driven supramolecular polymerization".

Regarding on the scientific discussion:

(3) I am an expert of supramolecular polymerization have read so many related papers. This manuscript is one of the worst papers in terms of writing skill. This manuscript contains some interesting results and is grammatically fine. However, it is very difficult to understand the contents scientifically because I could not easily understand with which data the authors support their claims. For example, from the beginning of the "Results and discussion", the authors claim

that MSS are orderly arrayed in LDH. I think most of the readers wonder why and how the authors probed it. As far as I found, their claims are based on their previous work in 2009 (ref 36). However, this previous work just discusses the orientation of the other small molecule. Hence, it is difficult to claim that MSSs used in this manuscript were also arrayed orderly in LDH. This point may not be critically important to discuss the possible mechanism. However, such scientifically inappropriate discussion makes us difficult to trust the results and discussions. There are so many similar inappropriate discussions (or scientifically not supported claims) in this manuscript. I don't have enough time to specially mention one by one but here I mention another example. The authors claimed that DHNS monomer could not realize desired supramolecular polymerization due to insufficient π - π stackings owing to its orientational angle with Supplementary figure 8. However, it is impossible to discuss the orientational angle with these figures in FigS.8. I could not understand why the authors could get such conclusion. In particular, DHNS has two hydroxy groups while none of the other monomers possess hydroxy group. These catechol moieties are known to strongly coordinate to transition metal ions. Hence, although I don't know the true reason, the authors have to show the solid evidence to support their claim.

(4) The authors have not quantified the amount of the incorporated MSSs in LDH. However, the authors must confirm that how much monomers were included in LDH, otherwise it is difficult to discuss the mechanisms with these materials.

(5) This is the most important comment from me. The authors claimed that they could control the size of metastable LSP with the size of LDH. In their proposed mechanism, they imagined that the sizes of SM were controlled by the size of LDH. However, there is no direct evidence to support this claim, because SMs were aggregated into big objects in Fig. 2a-d whose sizes are way much bigger than those of the corresponding LDHs. Even if the sizes of SMs were controlled by the size of LDHs, why the size of metastable LSPs could be controlled? In theory, the size of supramolecular polymers were controlled by the ratio of the monomers and nuclei (or seeds). Given that all the added monomers were used for the growth of the seeds (or short supramolecular polymers), the sizes can be controlled. To the best of my knowledge, none of the systems could control the size of the assemblies by changing the size of monomers. How could the authors explain? Furthermore, there is no scientific evidence that SMs are further assembled into metastable LSP or SMs are once disassembled into free MSS and re-assembled into metastable LSP. In my opinion, the authors did not explain why they could control the size with LDH, which is the key of this manuscript.

(6) Line 168 in p9; the authors described that "the activity of metastable LSP20 not only recovered, but also increased exponentially by gently shaking, confirmed by FL intensity increases to four times. It is reasonable that the activity of metastable LSP20 was recovered by shaking. In this case, the aggregated LSP20s are supposed to be disassembled into free LSP20s and, as a result, could initiate the seeded polymerization. On the other hand, the FL intensity change implies more than that. Basically, FL intensity changes due to the changes of the packing of SG7 accompanied by supramolecular polymerization. Generally, FL intensity does not change so much even if supramolecular polymers, like SMs, were aggregated, because their packings of monomers in supramolecular polymers were not changed. Hence, the observed increment in FL intensity may suggest that the aggregation of LSP20s were localized at the bottom of the optical cell and their FL intensity was underestimated. In that case, Figure 3D would be not be trustable. Hence, authors should monitor FL intensity under the gentle stirring and reconsider this part.

Regarding on experiments:

(7) I can understand that it is not easy to discuss the size of the assemblies in particular if they were aggregated. With the current data set such as figure S19 and Figure 3a,b, it is difficult to provide the trustable data. If the authors would like to discuss the sizes, they should find a condition where the assemblies are not aggregated and characterize their individual sizes, otherwise it's too risky.

CONTENTS

Responses to Reviewer 1.....S3

overview..... S3

comment 1..... S3

comment 2..... S4

comment 3..... S6

comment 4..... S12

comment 5..... S15

comment 6..... S28

comment 7..... S33

comment 8..... S37

comment 9..... S44

comment 10..... S45

comment 11..... S48

Responses to Reviewer 2.....S49

overview..... S49

comment 1..... S49

comment 2..... S54

comment 3..... S55

Responses to Reviewer 3.....S57

overview..... S57

comment 1..... S57

comment 2.....	S69
comment 3.....	S81
comment 4.....	S97
comment 5.....	S102
comment 6.....	S109
comment 7.....	S117

**Responses to Reviewer 1**

***Overview:** This manuscript describes a unique strategy to modulate the pathway of*
*supramolecular polymerisation, taking advantage of the confined space of layered double*
*hydroxide (LDH) to stabilize the metastable state of simple structure monomers. The*
*metastable state of aggregated monomers could remain even after the dissolution of LDH,*
*rendering this system with the ability for a living supramolecular polymerization (LSP).*
*Although the premise of the manuscript is novel and fascinating, the authors unfold their*
*story in a confusing way. Results were presented in the main text and the supporting*
*information but often detailed explanations are missing. This will make readers very confused.*
*Also, the quality of the data is not up to the standard at this stage. In summary, this*
*manuscript appears to be an excellent candidate for publication in Nature Communications*
*but much improvement is needed.*

Response: Thank you very much for your valuable comments. We appreciate your positive
evaluation on our work. Your comments are greatly helpful for improving the quality of our
manuscript. Therefore, we have sufficiently revised our manuscript according to your
comments, and all revisions are highlighted in red in the revised manuscript.

*1. The title of the manuscript seems not appropriate. Layered double hydroxide (LDH) of*
*Mg(NO₃)₂ and Al(NO₃)₃ is a confined nanospace for sure, while the self-assembly in a*
*confined space does not definitely correlate with the biomimetic term. The author should*
*reconsider the title of the manuscript.*

Response: Thank you very much for your suggestion. After our cautious consideration, we

decide to follow your advice and change the original title to “Supramolecular Polymerization
with Simple-Structured Monomers: Confinement Bring It to Life”. In addition, the
corresponding “Introduction” section has also been revised.

**p. 4 Line 9:** The confinement space that can impact all chemical events taking place in a
small cavity have been well documented in the field of nanoreactors, biosensors and drug
delivery vehicles, leading to a contrasting outcome than in the bulk.^{29,30} In terms of
self-assembly, there is evidence that confinement space can promote the formation and
stability of self-assembled complexes held together by intermolecular interactions.³¹⁻³⁶

**p. 4 Line 22:** To validate our hypothesis, the confinement effect of the layered double
hydroxide (LDH) nanomaterial was tried to guide the assembly of simple-structured
monomers by intercalating method.^{41,42} Theoretically, after removal of LDHs template, the
ordered assembly of guests will be destroyed because it has been reported that activation
barrier (30 kcal mol⁻¹) for similar process, spontaneous initiation of styrene polymerization⁴³,
is far beyond the energies associated with non-covalent bond formation.

*2. The full forms of several abbreviated scientific terms, eg. FITS, CLSM, TFA, SSP etc,*
*should be clearly mentioned in the main text where the abbreviation first appears.*

Response: Thank you for your valuable suggestions! As you suggested, all full forms of the
abbreviated scientific terms are shown clearly when first mentioned in the revised manuscript.

In addition, we renamed original supramonomer (SM) as living supramonomer (LSM) (Fig.

1). Because of the confinement space, the interlayer of LDH endows intercalated molecules

with oriented arrangement. After removal of LDH, the released ordered molecule array forms

LSM, acting as pre-assembled seeds, leading to the formation of thermodynamically
 favorable LSP.

 **Figure 1.** (a) Schematic presentation for the formation of the LSP. Normalized FL spectra
 and photos under UV light of (b) SG7 and SG7-LDH in CH₃OH; (c) fresh LSM and
 metastable LSP (LSP_{seed}); (d) fresh SSP and LSP. SEM images of (e) metastable LSP (f) LSP
 after aging for 12 h (g) SG7_{agg} and (h) fresh SSP.

The original supramonomer (SM) is renamed as living supramonomer (LSM) in the
 revised manuscript.

**p. 6 Line 7:** After removal of LDH by using methanol/trifluoroacetic acid (CH₃OH/TFA)
 mixture solvent (5:3 v/v), the pre-assembly as seed are obtained, which is a metastable and
 living supramonomer (LSM) (step 2). As confirmed by FL spectra and photos under UV light
 (Fig. 1b–c, Supplementary Fig. 7–8), compared with SG7 monomer ($\lambda_{em} = 430$ nm), the

pre-assembly LSM ($\lambda_{em} = 460$ nm) showed red-shifted wavelength, indicating emergence of
long-range π - π stacking among LSM; Compared with *SG7-LDHs* ($\lambda_{em} = 590$ nm), the
pre-assembly LSM ($\lambda_{em} = 460$ nm) showed a blue-shift, indicating the changes in stacking
structure. On the other hand, the metastable and living properties of LSM can be proved by
that LSM can continue to grow to metastable LSP driven by crystallization with increase poor
solvents TFA to CH₃OH/TFA (1:3 v/v), that is, the step 3.

*3. The authors should provide some evidence to confirm that the only the MSS intercalates*
*and self-assembles in the cavity of LDH, but not adsorb on the outer surface of the LDH.*
*Alternatively, some explanation is need of the reasons why the adsorbed MSS did not*
*assemble or did not affect the formation of metastable SM to a significant extent.*

Response: Thank you for this good suggestion. We have supplemented some experiments to
prove that simple-structured molecules are adsorbed on the surface of LDH and these
adsorbed molecules do not assemble into LSM and LSP. The failed formation of LSP
originates from the lower active SM. In addition, the appearance of adsorbed molecules
shows no significant effect on LSP formation.

It is worth mentioning that the simple-structured molecules in this work are not only
intercalated in the interlayer of LDH, but also adsorbed on the outer surface of LDH.
Fortunately, the adsorption capacity is very low. To quantitatively describe the effect of
adsorption of simple-structured monomers on assembly, we performed the following
experiments. CO₃-LDH (the interlayer anions of LDH is CO₃²⁻) was employed to adsorb
simple-structured molecules with negative charges to ensure that the intercalation could not

occur, because the affinity between CO_3^{2-} and LDH layer is much greater than that between
organic simple-structured molecules and LDH layer (*Catal. Lett.* **23**, 361–368 (1994); *Dalton*
*Trans.* **32**, 3499–3506 (2007)). Therefore, the amount of adsorbed and intercalated
simple-structured molecules in LDH can be quantified (Supplementary Table 1–2). For
example, in SG7-LDH₂₀ system, SG7 accounts for 26.32 wt.%. Among them, the adsorption
capacity of SG7 on the outer surface of CO_3 -LDH (named as SG7-LDH₂₀-surface) is 3.11
115 wt.%; the intercalation capacity of SG7 is 23.21 wt.%. That is, the adsorption capacity only
116 accounts for 13.41% of the total SG7.

In addition, the adsorbed simple-structured molecules not only did not assemble into
LSM and LSP, but its appearance would not affect the formation of LSP. This can be
confirmed by the successful formation of SSP in cycle experiments from the mixture of
SG7-LDH₂₀-surface and SG7-LDH₂₀.

Firstly, the adsorbed simple-structured molecules did not assemble into LSM and LSP,
which is confirmed by following experiment. We performed the adsorption of SG7 on the
surface of CO_3 -LDH₂₀, followed by same treatment process as SG7-LDH₂₀. The LDH layer
of SG7-LDH₂₀-surface (100 μL , 20.0 g L^{-1} , in CH_3OH) was removed by adding TFA (60 μL).
The fresh contrasting supramolecular polymer (SP_{20}) from adsorbed SG7 was obtained by
further adding TFA to $\text{CH}_3\text{OH}/\text{TFA} = 1:3$ v/v and applying ultrasound for 1 h. As seen from
the SEM images in Supplementary Fig. 18, rectangular SP_{20} split into small fragments in
cycle experiments by the addition of inactive SG7_{agg} due to the occurrence of disassembly,
indicating that SP_{20} failed to induce living assembly. In addition, the FL data can also confirm
above results. The cycle products of SP_{20} show decreased FL intensity in cycle 1–3,

attributing to the disassembly of SP₂₀ (Supplementary Fig. 19). Therefore, although adsorbed
SG7 forms SP₂₀ with rectangular morphology, it fails to induce following SSP in cycle
experiments, indicating that SP₂₀ is a kind of inactive SP. From the polarized FL profiles in
Supplementary Fig. 4 and 17, SG7-LDH₂₀-surface ($r = 0.171$) shows much lower anisotropic
value than that of SG7-LDH₂₀ ($r = 0.565$), indicating that the outer surface of LDH is lack of
the ability to make molecules to orderly arrange compared with the confinement space of
LDH. In all, the adsorbed SG7 did not assemble into LSP due to the absence of ordered
arrangement.

Secondly, the adsorbed small molecules did not affect the formation of LSP, which can
be confirmed by following experiment. The SG7-LDH₂₀-surface (50 μ L, 20.0 g L⁻¹, in
CH₃OH) and SG7-LDH₂₀ (50 μ L, 20.0 g L⁻¹, in CH₃OH) were physically mixed in equal
volumes, and then LDH layers were removed by adding TFA (60 μ L). Further adding TFA to
CH₃OH/TFA = 1:3 v/v and applying ultrasound for 1 h, the fresh mixed SP₂₀ were formed. In
SEM images (Supplementary Fig. 18), the cycle products of mixed SP₂₀ shows rectangular
morphology with increased size area, indicating successful formation of SSP. Therefore, the
mixed SP₂₀ has the ability to living assemble in cycle experiment. In Supplementary Fig. 19,
the FL intensity of the cycle products from mixed SP₂₀ shows regular increase in each cycle,
indicating the growth of cycle products (SSP). Therefore, we can conclude that adsorbed SG7
did not affect the formation of LSP.

**Supplementary Table 1.** The ICP-MS results of pure Cl-LDH precursors with different sizes
 in our work, the calculated ratio of Mg/Al and corresponding chemical formula.

Various LDHs	Mg (ppm)	Al (ppm)	Mg/Al	$[\text{Mg}_a\text{Al}_b(\text{OH})_x](\text{A}^{n-})_y \cdot z\text{H}_2\text{O}$	
				a	b
LDH ₂₀	3.89	1.41	3.06	0.75	0.25
LDH ₅₀	5.02	1.77	3.15	0.76	0.24
LDH ₁₀₀	6.49	2.44	2.95	0.75	0.25
LDH ₃₀₀₀	4.19	1.65	2.82	0.74	0.26

 **Supplementary Table 2.** The elemental analysis results of all intercalated LDHs in our work
 and the calculated weight fraction of intercalated molecules and corresponding chemical
 formula.

Various LDHs	S (wt.%)	SG7(wt.%)	average of SG7 (wt.%)
SG7-LDH ₂₀ -1	5.47	25.93	
SG7-LDH ₂₀ -2	5.74	27.21	26.32
SG7-LDH ₂₀ -3	5.45	25.83	
SG7-LDH ₂₀ -surface-1	0.66	3.13	
SG7-LDH ₂₀ -surface-2	0.66	3.13	3.11
SG7-LDH ₂₀ -surface-3	0.65	3.08	

 **Supplementary Figure 18.** SEM images of (a–d) fresh SP₂₀ made from SG7-LDH₂₀-surface
 and corresponding fresh SSP in Cycle 1–3, (e–h) fresh mixed SP₂₀ made from physically
 mixed SG7-LDH₂₀-surface and SG7-LDH₂₀ in equal volumes and corresponding fresh SSP in
 Cycle 1–3.

 **Supplementary Figure 19.** FL spectra of (a) SP₂₀ made from SG7-LDH-surface and
 corresponding fresh SSP in Cycle 1–3 (b) mixed SP₂₀ made from physically mixed
 SG7-LDH₂₀-surface and SG7-LDH₂₀ in equal volumes and corresponding fresh SSP in Cycle
 1–3.

**Supplementary Figure 4.** Polarized FL profiles and anisotropic value (r) for (a) SG7-LDH₂₀

($r = 0.565$), (b) SG7-LDH₅₀ ($r = 0.589$), (c) SG7-LDH₁₀₀ ($r = 0.531$) and (d) SG7-LDH₃₀₀₀ (r

$= 0.398$) in solid state on the quartz plate, respectively.

**Supplementary Figure 17.** Polarized FL profiles and anisotropic value (r) for

SG7-LDH₂₀-surface ($r = 0.171$) in solid state on the quartz plate.

As you suggest, the corresponding demonstration has been supplemented in the

manuscript to rule out the effect of adsorbed SG7.

**p. 7 Line 17:** It is worth mentioning that the SG7 is not only intercalated in the cavity of

LDH, but also adsorbed on the outer surface of the LDH. To quantitatively describe the effect
of SG7 adsorption on assembly, we employed the $\text{CO}_3\text{-LDH}_{20}$ (the interlayer anions of LDH
is CO_3^{2-}) to adsorb SG7 (marked as SG7-LDH₂₀-surface, detailed in Supporting Information),
where the intercalation does not occur because the affinity between CO_3^{2-} and LDH layer is
much greater than that between organic simple-structured molecules and LDH layer.^{44,45}
ICP-MS and element analysis results show the adsorption capacity is very low and only
accounts for 13.41% of the total SG7 (Supplementary Table 1–2). The polarized FL profiles
in Supplementary Fig. 4 and 17, SG7-LDH₂₀-surface ($r = 0.171$) shows much lower
anisotropic value than that of SG7-LDH₂₀ ($r = 0.565$). In addition, the adsorbed SG7 did not
assemble into LSP, as confirmed by SEM images and FL spectrum. SEM showed a
rectangular morphology SP can be obtained, but it fails to induce following cycle products
(Supplementary Fig. 18), indicating that SP₂₀ is not a kind of LSP. Instead, the appearance of
depolymerization of SP₂₀ in cycle process, as certified by decreased FL intensity of cycle
products of SP₂₀ (Supplementary Fig. 19). To sum up, the adsorbed SG7 did not assemble
into LSP due to lack of ordered arrangement. Importantly, the appearance of adsorbed SG7
did not affect the formation of LSP, as proved by physically mixing SG7-LDH₂₀-surface and
SG7-LDH to perform cycle experiments. The cycle products showed rectangular morphology
with increased size area in SEM and FL intensity, indicating the cycle products have the
ability to living assembly (Supplementary Fig. 18–19).

4. *The stability of the metastable SM in step 2, right after the removal of LDH by using*
*CH₃OH/TFA, should be examined, for example by ageing the solution for at least one week.*

Response: Thank you for your guiding comments. We have supplemented the corresponding
experiment according to the reviewer's suggestion, and the illustration has been added in the
revised manuscript.

In fact, the LSM in step 2 is unstable because CH₃OH is a good solvent for SG7. In
CH₃OH/TFA (5:3 v/v), it is easy for LSM to disassemble into SG7 monomer because the
H-bond net is replaced by the H-bond between solvent CH₃OH and SG7. As seen from the
Supplementary Fig. 23, LSM cannot keep its ordered structure for a long time (>20 min) in
CH₃OH/TFA (5:3 v/v). It is necessary to point out that precisely because of this instability,
when we continue to increase poor solvents TFA to CH₃OH/TFA (1:3 v/v), LSM is
crystallization-driven to form LSP. In addition, the theoretical calculation results also testify
that LSM prefers to continuously elongate to thermodynamically favorable LSP product due
to the high binding energy of $-52.64 \text{ kJ mol}^{-1}$ per molecule (Supplementary Fig. 11h). Thus,
in experimental process, we all use freshly prepared LSM instead of aged LSM.

**Supplementary Figure 23.** SEM images of aged SG7 LSM₂₀ for 20 min after removal of
LDHs in CH₃OH/TFA (5:3 v/v).

 **Supplementary Figure 11. (a)** Schematic illustration for the definition of orientation angle, θ .
 **(b)** Energy diagram of SG7 LSM with different orientation. Optimized geometries of **(c)**
 SG7-LDH, **(d)** inactive SG7 aggregates ($\theta = 35^\circ$), **(e)** SG7 LSM+SG7 monomer ($\theta = 35^\circ$) and
 **(f)** SG7 LSM+SG7 monomer ($\theta = 9^\circ$), together with the highlighted binding areas in e and f:
 **(g)** $\theta = 35^\circ$ and **(h)** $\theta = 9^\circ$. The color of each element is labeled in c.

As you suggested, we emphasize the instability of LSM.

**p. 10 Line 3:** Note that, metastable LSMs are easily destroyed by new H-bond between
 CH_3OH and SG7 molecule, preventing continued growth confirmed by SEM (Supplementary
 Fig. 23). Thus, by increasing the amount of poor solvent TFA, combined with the
 crystallization driving principle,⁵²⁻⁵⁴ LSMs can continue to grow into LSP (as-prepared LSPs
 made from different size LDHs are named as LSP₂₀, LSP₅₀, LSP₁₀₀ and LSP₃₀₀₀,
 respectively).

5. In many places, the authors commented without giving any explanation or sufficient
experimental data support. For example, on page 5, line 105: “The effect of ion on this result
during the dissolution of LDH can be ruled out by control experiment (Supplementary Fig.
2)”. The author should provide a comparison of SEM images, i.e. before and after mixing the
SG7 to the LDH in CH₃OH/TFA (1:3 v/v). Again, in Page 6, line 111:
“6,7-dihydroxynaphthalene-2-sulfonate (DHNS), the formation of π - π stacking is hardly
possible owing to its orientation angle (Supplementary Fig. 8)”. Solely based on these SEM
images, how could the authors conclude that a molecule has negligible/no π - π stacking
interactions? More information and explanation is needed.

Response: Thank you very much for your comments. We have supplemented the
corresponding experiments (SEM, spectroscopy, XRD and *ab initio* molecular dynamics
(AIMD) simulations) according to the reviewer’s suggestion, and the illustration has been
added in the revised manuscript.

● As you suggested, corresponding SEM images and statement have been presented to
provide a clear comparison of morphology between the LSP and control samples
(Supplementary Fig. 20). There are three control samples, including 1) pure SG7 powder, 2)
pure Cl-LDH and 3) SG7+LDH (physically mixed pure SG7 powder and pure Cl-LDH). The
three control samples were respectively treated by same process as the formation of SG7 LSP.
For pure SG7 powder, it can only form isotropic spherical structures in SEM image
(Supplementary Fig. 20a) and shows low anisotropic FL of $r = 0.0201$ (Supplementary Fig. 5).
For pure Cl-LDH, SEM image shows irregular morphology of metal salt because Cl-LDH
dissolves in CH₃OH/TFA (1:3 v/v) in the form of Mg²⁺, Al³⁺ and Cl⁻ (Supplementary Fig.

20b). For the SG7+LDH, it can also only form small isotropic spherical structures in SEM
image (Supplementary Fig. 20c). The results of three control experiments indicate that the ion
of dissolved LDH is not the key to facilitate rectangular morphology of LSP.

● Thank you very much for your comments. It is indeed hard to explain the failed
formation of DHNS LSP solely based on SEM images. Thus, we have supplemented the data
about the array of DHNS in LDH, including XRD, UV-vis spectra, polarized FL profiles and
anisotropy, ICP-MS, element analysis and AIMD simulations.

XRD results show the successful intercalation of DHNS into LDHs layers, named as
DHNS-LDH (Supplementary Fig. 52). Before intercalation, Cl-LDH shows diffraction peaks
of (003), (006), (009) and (110) at ca. 11.3°, 22.4°, 35.0° and 60.9°, respectively
(Supplementary Fig. 3). According to the Bragg equation of $2d\sin\theta = n\lambda$, where d is the
spacing between crystal plane, θ is the angle between the incident X-ray and the
corresponding crystal plane, n is diffraction series ($n = 1$) and λ is wavelength of X-ray ($\lambda =$
1.54 Å). Therefore, the d -spacing of Cl-LDH was calculated to be 0.782 nm. The d -spacing
was increased to 1.22 nm after the DHNS intercalation. The increased d -spacing indicates the
successful intercalation of DHNS into LDH layers. UV spectrum of the DHNS-LDH (20 g/L
in CH₃OH) shows red shift of 10 nm than DHNS (5 mM in CH₃OH), demonstrating the
appearance of π - π stacking of DHNS in the interlayer of LDH (Supplementary Fig. 54).
Nevertheless, the low FL anisotropic value ($r = 0.0214$) displays that the π - π stacking
structure is negligible long-range ordered arrangement (Supplementary Fig. 57).

To further confirm the above experimental results, the AIMD simulations of
DHNS-LDH were performed. The chemical formula of DHNS-LDH was

$\text{Mg}_{48}\text{Al}_{16}(\text{OH})_{128}(\text{C}_{10}\text{H}_5\text{SO}_5)_5\text{Cl}$ according to the results from ICP, element analysis
(Supplementary Table 1 and 6). The lattice parameters ($a = b = 2d_{(110)} = 3.07 \text{ \AA}$, $c = d_{(003)} =$
7.82 \AA) were referred to the corresponding XRD results. From the AIMD simulations results
in Supplementary Fig. 62, the adjacent DHNS molecules in the interlayer of LDH do have
π - π stacking in short-range, attributing to three functional groups and the large aromatics ring
of DHNS. However, these π - π stacking shows disordered arrangement in the long range.

On the basis of above four kinds of tests, DHNS molecules fail to orderly arrange in the
confinement space of LDH. Therefore, it is impossible for disordered DHNS molecules to
overcome a huge energy barrier from spontaneous nucleation to living elongation, leading to
the failure of formation of LSP.

● Furthermore, the data about the array of other molecules (SG7, BSA, ABSA, DABSA
and CR) used in our work are also supplemented through XRD, UV-vis spectra, polarized FL
profiles and anisotropy, ICP-MS, element analysis, and AIMD simulations.

Similarly, XRD results showed the successful intercalation of these molecules into
LDHs interlayers, named as SG7-LDH, BSA-LDH, ABSA-LDH, DABSA-LDH and
CR-LDH (Supplementary Fig. 52). The SG7-LDH, BSA-LDH, ABSA-LDH, DABSA-LDH
and CR-LDH show the d -spacing increased to 0.896 nm, 1.48 nm, 1.57 nm, 1.45 nm and
0.877 nm, respectively, indicating the successful intercalation.

UV spectra of SG7-LDH ($\Delta\lambda = \sim 100 \text{ nm}$), ABSA-LDH ($\Delta\lambda = \sim 5 \text{ nm}$), DABSA-LDH
($\Delta\lambda = \sim 75 \text{ nm}$) and CR-LDH ($\Delta\lambda = \sim 5 \text{ nm}$) showed varying degrees of red shift relative to
their respective monomers (Supplementary Fig. 1, 53 and 55). The redshifts are attributed to
stronger π - π stacking interactions between intercalated molecules than isotropic monomers in

CH₃OH. For intercalated BSA in LDH, there is no enhanced π - π stacking due to lack of
functional group, confirmed by the absence of red shift in UV spectra.

From the polarized FL profiles, SG7-LDH ($r = 0.565$) shows higher anisotropic value
than SG7 powder ($r = 0.0201$, Supplementary Fig. 4–5), indicating the ordered arrangement
of SG7 molecules in the interlayer of LDH. Similarly, ABSA-LDH ($r = 0.586$),
DABSA-LDH ($r = 0.760$) and CR-LDH ($r = 0.691$) show high anisotropic value, confirming
their ordered arrangement in the interlayer of LDHs (Supplementary Fig. 56 and 58). The low
anisotropic value of BSA-LDH ($r = 0.104$) demonstrates the lack of ordered arrangement of
intercalated BSA molecules, which is in line with UV spectra results.

To further confirm the above experimental results, the AIMD simulations were
performed on these intercalated LDH. The chemical formula were
$\text{Mg}_{48}\text{Al}_{16}(\text{OH})_{128}(\text{C}_{16}\text{H}_6\text{O}_{10}\text{S}_3)_4$, $\text{Mg}_{48}\text{Al}_{16}(\text{OH})_{128}(\text{C}_6\text{H}_5\text{SO}_3)_{16}$,
$\text{Mg}_{48}\text{Al}_{16}(\text{OH})_{128}(\text{C}_6\text{H}_6\text{NSO}_3)_{16}$, $\text{Mg}_{48}\text{Al}_{16}(\text{OH})_{128}(\text{C}_6\text{H}_7\text{N}_2\text{SO}_3)_{16}$, and
$\text{Mg}_{48}\text{Al}_{16}(\text{OH})_{128}(\text{C}_{32}\text{H}_{22}\text{N}_6\text{S}_2\text{O}_6)_2(\text{Cl})_{12}$ according to the results from ICP-MS, element
analysis (Supplementary Table 1 and 6), respectively. The lattice parameters ($a = b = 2d_{(110)}$, c
$= d_{(003)}$) were referred to the corresponding XRD results. From the AIMD simulations in
Supplementary Fig. 6, 60, 61 and 63, intercalated SG7, ABSA, DABSA and CR show
ordered arrangement in the interlayer of LDH, respectively. Furthermore, we also give the
precise angle θ (Supplementary Fig. 64) of the arrangement of these intercalated molecules.
The θ for SG7, ABSA, DABSA and CR can be calculated to be 9°, 43°, 86° and 0°,
respectively. The AIMD simulations of BSA-LDH show the disordered arrangement of BSA,
which is in consistent with the low FL anisotropic value of BSA-LDH ($r = 0.104$)

(Supplementary Fig. 56).

On the basis of above results, the successful formation of LSP is based on the ordered
arrangement of intercalated molecules in the confinement space of LDH. In contrast,
disordered intercalated molecules fail to promote the formation of LSP.

**Supplementary Figure 20.** SEM images of (a) pure SG7 (b) pure Cl-LDH and (c)
SG7+LDH dissolved in CH₃OH/TFA (1:3 v/v), all of them with the same dosage as SG7 LSP.

**Supplementary Figure 5.** Polarized FL profiles and anisotropic value (r) for untreated SG7
powder ($r = 0.0201$).

.

**Supplementary Figure 52.** XRD patterns of BSA-LDH, ABSA-LDH, DABSA-LDH,

DHNS-LDH and CR-LDH.

**Supplementary Figure 3.** XRD patterns of (a) pure LDH precursor powder and (b)

SG7-LDH with different sizes.

**Supplementary Figure 54.** UV-vis spectra of DHNS-LDH (10 g/L) and DHNS (5 mM) in

CH₃OH (optical path = 0.1 mm).

**Supplementary Figure 57.** Polarized FL profiles and anisotropic value (*r*) for DHNS-LDH

(*r* = 0.0214) in solid state on the quartz plate.

**Supplementary Table 1.** The ICP-MS results of pure Cl-LDH precursors with different sizes

in our work, the calculated ratio of Mg/Al and corresponding chemical formula.

Various LDHs	Mg (ppm)	Al (ppm)	Mg/Al	$[\text{Mg}_a\text{Al}_b(\text{OH})_x](\text{A}^{n-})_y \cdot z\text{H}_2\text{O}$	
				a	b
LDH ₂₀	3.89	1.41	3.06	0.75	0.25
LDH ₅₀	5.02	1.77	3.15	0.76	0.24
LDH ₁₀₀	6.49	2.44	2.95	0.75	0.25
LDH ₃₀₀₀	4.19	1.65	2.82	0.74	0.26

**Supplementary Table 2.** The elemental analysis results of all intercalated LDHs in our work
 and the calculated weight fraction of intercalated molecules and corresponding chemical
 formula.

Various LDHs	S	O	H	A ⁿ⁻	[Mg _a Al _b (OH) _x](A ⁿ⁻) _y ·zH ₂ O		
	(wt.%)	(wt.%)	(wt.%)	(wt.%)	x	y	z
SG7-LDH ₂₀	5.47	33.40	4.73	25.93	1.93	0.08	2.2
SG7-LDH ₅₀	5.72	33.90	4.40	27.11	1.80	0.11	2.9
SG7-LDH ₁₀₀	5.69	33.61	4.54	26.97	1.85	0.10	2.5
SG7-LDH ₃₀₀₀	5.70	33.63	4.60	27.02	1.90	0.09	2.4
BSA-LDH ₂₀	5.07	32.01	5.45	25.03	2.03	0.22	2.1
ABSA-LDH ₂₀	5.21	29.94	5.11	28.17	1.98	0.27	2.3
DABSA-LDH ₂₀	4.90	29.34	4.98	28.79	1.96	0.29	2.6
DHNS-LDH ₂₀	3.32	24.05	3.94	24.80	2.04	0.08	2.5
CR-LDH ₂₀	3.54	29.19	5.22	35.95	1.80	0.10	2.7

**Supplementary Figure 62.** The snapshot of DHNS-LDH after AIMD simulation of 100 ps.

**Supplementary Figure 1.** UV-vis spectra of (a) SG7-LDH aqueous dispersion (20.0 g L^{-1})

with different sizes (optical path = 0.2 mm), (b) SG7 solution in CH_3OH with a set of

concentrations (optical path = 10 mm) and (c) SG7 solution in CH_3OH with different

concentrations (optical path = 0.2 mm).

**Supplementary Figure 53.** UV-vis spectra of intercalated LDH (20 g/L in CH_3OH) and

corresponding contrast samples (5 mM in CH_3OH) (optical path = 0.1 mm): (a) BSA-LDH,

(b) ABSA-LDH and (c) DABSA-LDH.

**Supplementary Figure 55.** UV-vis spectra of CR-LDH (2 g/L in CH_3OH) and corresponding

contrast samples (1 mM in CH₃OH) (optical path = 0.1 mm).

**Supplementary Figure 4.** Polarized FL profiles and anisotropic value (r) for **(a)** SG7-LDH₂₀

($r = 0.565$), **(b)** SG7-LDH₅₀ ($r = 0.589$), **(c)** SG7-LDH₁₀₀ ($r = 0.531$) and **(d)** SG7-LDH₃₀₀₀ (r

= 0.398) in solid state on the quartz plate, respectively.

**Supplementary Figure 56.** Polarized FL profiles and anisotropic value (r) for **(a)** BSA-LDH

($r = 0.104$), **(b)** ABSA-LDH ($r = 0.586$) and **(c)** DABSA-LDH ($r = 0.760$) in solid state on

the quartz plate, respectively.

**Supplementary Figure 58.** Polarized FL profiles and anisotropic value (r) for CR-LDH ($r =$

0.691) in solid state on the quartz plate.

**Supplementary Figure 6.** (a) Schematic illustration for the definition of orientation angle, θ .

(b) The snapshot of SG7-LDH after AIMD simulation of 100 ps.

**Supplementary Figure 60.** The snapshot of ABSA-LDH after AIMD simulation of 100 ps.

**Supplementary Figure 61.** The snapshot of DABSA-LDH after AIMD simulation of 100 ps.

**Supplementary Figure 63.** The snapshot of CR-LDH after AIMD simulation of 100 ps.

**Supplementary Figure 64.** Schematic illustration for the definition of θ for (a) BSA, (b)

ABSA, (c) DABSA, (d) DHNS and (e) CR.

The demonstration about the supplemented control experiments and universality data
can be seen in the revised manuscript.

**p. 8 Line 14:** Generally, the presence of inorganic ions during the assembly process will

affect the result of the assembly.^{46,47} In our system, in the process of dissolving LDH, metal

ions (Mg^{2+} and Al^{3+}) are introduced. In order to eliminate their influence, we designed a
series of comparative samples treated by the same process as assembly of SG7-LDH,
including SG7 powder, Cl-LDH and SG7+LDH (physically mixed SG7 and Cl-LDH). For
the SG7 and the SG7+LDH, both can only form isotropic spherical structures as shown in
SEM image (Supplementary Fig. 20a, 20c). For Cl-LDH, SEM image shows irregular
morphology of metal salt (Supplementary Fig. 20b). These results indicate that the ion of
dissolved LDH is not the key to facilitate rectangular morphology of LSP.

**p. 20 Line 4:** As a proof of concept to validate the LDH confinement-driven LSP,
various simple-structural molecules with aromatic rings and negative charge (Fig. 1a) were
chosen to orderly array in confinement space of LDH (step 1, Supplementary Fig. 52–66) and
further performed the assembly process same as SG7 (Supplementary Fig. 67–72). By the
assembly results, the basic principles are summarized as follows: 1) the formation of H-bond
net among released $-NH_2$, $-HSO_3$ and $-OH$. For example, the benzenesulfonic acid (BSA)
with insufficient active hydrogen bond (H-bond) cannot form H-bond net (Supplementary Fig.
53, 56 and 59), leading to the failure in formation of LSP (Supplementary Fig. 68). In
contrast, the appearance of multiple active H-bonds easily forms the LSPs (e.g.,
3-aminobenzene sulfonic acid, 3-ABSA; 2,5-diaminobenzenesulfonic Acid, 2,5-DABSA;
Congo red, CR); 2) oriented long-range π - π stacking *between* neighboring molecules. For
example, the 6,7-dihydroxynaphthalene-2-sulfonate (DHNS) shows the failed formation of
LSP (Supplementary Fig. 71) . The reasons originate from that although the appearance of
π - π stacking of DHNS molecules in the interlayer of LDH as confirmed by UV spectrum
(Supplementary Fig. 54), the π - π stacking is negligible long-range ordered arrangement as

confirmed by a low FL anisotropic value ($r = 0.0214$) (Supplementary Fig. 57). AIMD
simulations, which are referred to XRD, ICP-MS and element analysis results
(Supplementary Fig. 52, Table 1 and Table 6), also further proved the adjacent DHNS
molecules in the interlayer of LDH do have π - π stacking in short-range, attributing to three
functional groups and the large aromatics ring of DHNS. However, they show disordered
arrangement in the long range (Supplementary Fig. 62). Therefore, our approach provides a
good alternative to form oversized LSP by simple-structural molecules.

6. Page 6, line 119: “SG7 is chosen as a model because its favorable optical properties allow
naked eye visualizing of assembly events”. To support this author only provides the
spectroscopic data in the SI without any explanation. More information is needed.

Response: As suggested by the reviewer, we have explained in detail the reasons for choosing
SG7 in revised manuscript.

SG7 is a kind of typical optical molecule with quite big difference in fluorescence
properties between monomer and various stacking states, providing the prerequisite for naked
eye visualization under UV lamp, which can be confirmed by following experimental data. In
FL spectra, SG7 monomer dissolved in CH₃OH has one main peak at $\lambda = 420\sim 435$ nm
(Supplementary Fig. 2). When the concentration of SG7 in CH₃OH approaches saturation, a
new FL peak appears at 511 nm, attributing to SG7 excimer (*ACS Appl. Mater. Interfaces* **10**,
18012–18020 (2018)). Compared with SG7 monomer and excimer, FL emission of
SG7-LDH shows red-shift ($\lambda = 560\sim 600$ nm) because confinement space of LDH endows
SG7 with ordered arrangement and long-range π - π stacking (Supplementary Fig. 7). After

removal of the LDH layers, FL spectra of LSM ($\lambda = 460$ nm) and LSP ($\lambda = 530\sim 545$ nm)
 show quite different emission wavelength (Supplementary Fig. 8–9). Under UV lamp,
 SG7-LDH is orange light, SG7 LSM is blue light and SG7 LSP is yellow light. Therefore, the
 successful formation of LSP can be preliminarily judged by naked eye (Fig. 1).

 **Supplementary Figure 2.** FL emission spectra of SG7 solution in CH₃OH: **(a)** with a set of
 concentrations and **(b)** saturated solution.

 **Supplementary Figure 7.** **(a)** FL emission spectra and **(b)** corresponding normalized FL
 spectra of SG7-LDH with different sizes.

 **Supplementary Figure 8.** **(a)** FL emission spectra and **(b)** corresponding normalized FL

spectra of metastable LSM₂₀–LSM₃₀₀₀ after removal of LDH layers.

**Supplementary Figure 9.** FL spectra of LSP₂₀–LSP₃₀₀₀: (a) freshly sonicated metastable

LSP and (b) stable LSP kept for 12 h after ultrasound.

**Figure 1.** (a) Schematic presentation for the formation of the LSP. Normalized FL spectra

and photos under UV light of (b) SG7 and SG7-LDH in CH₃OH; (c) fresh LSM and

metastable LSP (LSP_{seed}); (d) fresh SSP and LSP. SEM images of (e) metastable LSP (f) LSP

after aging for 12 h (g) SG7_{agg} and (h) fresh SSP.

**p. 5 Line 14 in manuscript:** Here, 8-hydroxypyrene-1,3,6-trisulfonate (solvent green 7,
SG7) is chosen as a model to study for the LSP because it possesses a pyrene molecule
bearing hydrogen-bonding moieties and negative charges, which is expected to self-assemble
*via* π - π stacking of the pyrene planes and hydrogen-bond (H-bond) of sulfonyl group.
Importantly, there exist remarkable different fluorescence (FL) properties between monomer
and various stacking states (Supplementary Fig. 1–2 for a detailed analysis), providing the
prerequisite for naked eye visualization of assembly events. Fig. 1a depicts the formation
process of LSP (and seed-induced supramolecular polymerization, SSP) from SG7 monomer.
Upon intercalating SG7 monomer into confinement space of LDH, named as SG7-LDH,
which is confirmed by powder X-ray Diffraction (XRD) measurements (step 1,
Supplementary Fig. 3), the ordered arrangement of SG7 were achieved. As confirmed by FL
anisotropy, SG7-LDH showed a high anisotropic value ($r = 0.565$) compared with untreated
SG7 powder ($r = 0.0201$) (Supplementary Fig. 4–5). The *ab initio* molecular dynamics
(AIMD) simulations further testified the ordered arrangement of SG7 with an orientation
angle of $\theta = 9^\circ$ (θ is the inclined angle between the principal axis of molecule and the
elongation axis of LSM, Supplementary Fig. 6). After removal of LDH by using
methanol/trifluoroacetic acid (CH₃OH/TFA) mixture solvent (5:3 v/v), the pre-assembly as
seed are obtained, which is a metastable and living supramonomer (LSM) (step 2). As
confirmed by FL spectra and photos under UV light (Fig. 1b–c, Supplementary Fig. 7–8),
compared with SG7 monomer ($\lambda_{em} = 430$ nm), the pre-assembly LSM ($\lambda_{em} = 460$ nm) showed
red-shifted wavelength, indicating emergence of long-range π - π stacking among LSM;
Compared with SG7-LDHs ($\lambda_{em} = 590$ nm), the pre-assembly LSM ($\lambda_{em} = 460$ nm) showed a

blue-shift, indicating the changes in stacking structure. On the other hand, the metastable and
living properties of LSM can be proved by that LSM can continue to grow to metastable LSP
driven by crystallization with increase poor solvents TFA to CH₃OH/TFA (1:3 v/v), that is,
the step 3. The assembly process results in the larger π - π stacking of metastable LSP than
LSM, which is testified by the redshift of fresh metastable LSP ($\lambda_{em} = 535$ nm) compared
with LSM ($\lambda_{em} = 460$ nm) in FL emission spectra (Fig. 1c, Supplementary Fig. 8–9). The
wide excitation range of LSP is also in line with the long-range π - π stacking rather than a
single molecule (Supplementary Fig. 10). Furthermore, the theoretical calculation shows
LSM prefers to continuously elongate to thermodynamically favorable LSP product due to
the high binding energy of -52.64 kJ mol⁻¹ per SG7 molecule (Supplementary Fig. 11). The
step 4 is an elongation process of metastable LSP by self-replication, as confirmed by
scanning electron microscopy (SEM) images (Fig. 1 e–f and Supplementary Fig. 12).
Compared with metastable LSP, the FL emission spectra of LSP (Supplementary Fig. 9)
show that the decreased FL intensity is attributed to the FL quenching by elongation;
unchanged wavelength ($\lambda_{em} = 535$ nm) is attributed to the maintenance of stacking structure
type. The step 5 is to confirm the character of living polymerization by adding inactive
SG7_{agg} (SG7 dissolved in CH₃OH/TFA (1:3 v/v), 2.5 mM) into metastable LSP (LSP_{seed}) to
form SSP. In FL emission spectra (Supplementary Fig. 13–14), the formed SSP ($\lambda_{em} = 525$
499 nm) shows redshift compared with SG7_{agg} ($\lambda_{em} = 450$ nm), indicating the rearrangement of
500 SG7_{agg} induced by LSP_{seed}. In addition, SSP shows a emission color change and blue shift of
501 10 nm compared with LSP (Fig. 1d), attributing to the tiny difference in stacking states
between SSP and LSP.

**p. 20 Line 14 in Supporting Information:** SG7 is a kind of typical optical molecule with
quite big difference in fluorescence properties between monomer and excimer, providing the
prerequisite for naked eye visualization under UV lamp, which can be confirmed by above
experimental data. In FL spectra, SG7 monomer dissolved in CH₃OH has one main peak at λ
= 420~435 nm (Supplementary Fig. 2). When the concentration of SG7 in CH₃OH
approaches saturation, a new FL peak appears at 511 nm, attributing to SG7 excimer.⁵

*7. The scale bar of both sets of SEM images in Figure 2 is different, which makes it difficult*
*to appreciate the increased length and width of the living supramolecular polymeric*
*structures. Can the authors provide the SEM images for both sets with an identical scale bar?*

Response: Thank you very much for your suggestion.

Indeed, it is difficult to appreciate the increased length and width under the different
scale bar in original Fig. 2. Therefore, as you suggested, all SEM images in Fig. 2 have been
revised to an identical scale.

Moreover, in order to characterize the SEM performance of the LSM, we used a simple
solvent evaporation method for drying sample in silicon substrate, which resulted in severe
aggregation (shown in original Fig 2) and wrong judgment of LSM size. To solve these
problem, we use PSS (poly(sodium-p-styrenesulfonate)) to modify silicon substrate. Once the
modified substrate is immersed in the fresh LSM solution, the sufficient $-\text{SO}_3^-$ of PSS will
form $-\text{SO}_3\text{H}$, which is beneficial to form H-bond between PSS and LSM. Moreover, the
immersing time of substrate should be as short as possible to avoid too many LSM dip-coated
on the modified substrate. However, even with such a sophisticated treatment, it should be

pointed out that for nanoscale structures, aggregation cannot be completely avoided. Thus, in
order to observe clearly and accurately, we etched part of the LDH to expose the LSM before
aggregation, that is, the LSM is still on the etched fragments of LDH.

As seen in revised Fig. 2a–c and their insets, the hexagonal sheet structure of etched
SG7-LDH₂₀, SG7-LDH₅₀ and SG7-LDH₁₀₀ are difficult to observe because the small size of
LDH leads to uncontrollable rapid dissolution. Even so, it can be concluded that as the
increase of LDH size, the size of LSM also increases because the LSM is still on the etched
fragments of LDH. Besides, the etched piece of SG7-LDH₃₀₀₀ remained the hexagonal sheet
structure due to the slower dissolution, where the size of 200 nm attributed to LSM₃₀₀₀ can be
clearly seen (Fig. 2d).

In addition, the size of stable LSP is determined by two factors: 1) the starting LSM size.
It can be confirmed by the LSP₅₀, LSP₁₀₀ and LSP₃₀₀₀ samples, where with increase of the
LSM size from 10 to 200 nm (mark the size in the inset in Fig 2 with a yellow line), the
rectangular LSP₅₀, LSP₁₀₀ and LSP₃₀₀₀ display increasing area from 0.32 to 4.00 μm^2 (Fig.
2e–h); 2) the activity of LSM. It can be proved that although the starting size of LSM₂₀ is the
smallest, the corresponding LSP₂₀ area can indeed reach $\sim 1.50 \mu\text{m}^2$. In the cycle experiment,
only SSP₂₀ showed an increase in area, while SSP₅₀–SSP₃₀₀₀ did not increase (Supplementary
Fig. 15 and 35), so LSP₂₀ has the best activity endowed with smallest LDH₂₀.

**Figure 2. Tuning the size and activity of LSP by LDH confinement. (a–d)** SEM images of

SG7 LSM from different size SG7-LDHs: (a) 20 nm, (b) 50 nm, (c) 100 nm and (d) 3 μ m

(insets are corresponding magnifications and the LSM sizes in the insets are marked with a

yellow line). (e–h) SEM images of SG7 LSP from the corresponding LSM in a–d,

respectively. (i–l) Debye plot of metastable LSP made of corresponding LSM in a–d,

respectively.

**Supplementary Figure 15.** SEM images of (a) fresh metastable LSP made of SG7,
 corresponding fresh SSPs in (b) Cycle 1 and (c) Cycle 2, and (d) contrast sample: physically
 mixing pure SG7 and Cl-LDH in CH₃OH/TFA (1:3 v/v) with the same dosage as LSP.

**Supplementary Figure 35.** SEM images of fresh (a) SSP₅₀, (b) SSP₁₀₀ and (c) SSP₃₀₀₀ in
 Cycle 1.

**p. 20 Line 14:** It should be pointed out that for nanoscale structures, aggregation cannot
 be completely avoided.^{50,51} Thus, in order to observe accurately, we etched part of the LDH
 to expose the LSM before aggregation, that is, the LSM is still on the etched fragments of
 LDH. As confirmed by SEM images (Fig. 2a–d and their insets), the sizes of metastable

LSMs gradually increase with the raising of LDH sizes from 20 nm to 3 μm (as-prepared
SMs from different size LDHs are named as LSM₂₀, LSM₅₀, LSM₁₀₀ and LSM₃₀₀₀,
respectively). Note that, metastable LSMs are easily destroyed by new H-bond between
CH₃OH and SG7 molecule, preventing continued growth confirmed by SEM (Supplementary
Fig. 23). Thus, by increasing the amount of poor solvent TFA, combined with the
crystallization driving principle,⁵²⁻⁵⁴ LSMs can continue to grow into LSP (as-prepared LSPs
made from different size LDHs are named as LSP₂₀, LSP₅₀, LSP₁₀₀ and LSP₃₀₀₀, respectively).
The highly ordered structure of LSP₂₀-LSP₃₀₀₀ can be confirmed by high FL anisotropic
values ($r > 0.45$, Supplementary Fig. 24). With increase size of LDH from 50 nm to 3 μm ,
LSP₅₀-LSP₃₀₀₀ display successively increased area from 0.32 μm^2 to 4.0 μm^2 (Fig. 2f-h).
Interestingly, although LSP₂₀ does not comply with the above rules, it shows surprising
results of unexpected extension with an area of $\sim 1.50 \mu\text{m}^2$ (Fig. 2e), testifying activity of
LSP₂₀ is the strongest, which is discussed in the following cycle experiments.

8. *Throughout the manuscript, in all the SEM images the fibers appear in agglomerate and*
*interconnected state. Because it is hard to clearly see the fibers discretely, the corresponding*
*statics on their length and width profiles will be less reliable or accurate. Can the authors*
*show examples of more diluted fibers to a better display of their length and width?*

Response: Thank you for your comments. Your comments make us aware of this problem
existing in SEM images throughout the manuscript. Therefore, all SEM images are revised by
improving sample preparation.

Generally, to observe the SEM image, the simple solvent evaporation method is used for

drying sample on silicon substrate, which resulted in severe aggregation (just like the SEM
image we obtained in original Fig 2). However, the conventional dilution operation is not
suitable for our current system, because when a good solvent is added, the assembly will
change. To solve the problem, we use PSS (poly(sodium-p-styrenesulfonate)) to modify
silicon substrate. Once the modified substrate is immersed in the acidic environment, the
sufficient $-\text{SO}_3^-$ of PSS will form $-\text{SO}_3\text{H}$, which is beneficial to form H-bond between PSS
and SG7 LSM (or LSP and SSP). Moreover, the immersing time of substrate should be as
short as possible to avoid too many LSM (or LSP and SSP) dip-coated on the modified
substrate. However, even with such a sophisticated treatment, it should be pointed out that
aggregation cannot be completely avoided due to the high surface activity derived from the
inherent characteristics of nanostructures. Despite the existence of these problems, by using
the improved preparation method, the size of the as-prepared sample can be clearly and
accurately observed and compared.

In revised manuscript, all SEM images in Fig. 2 and Fig. 4 are revised to clearly show
the sizes of LSM, LSP and SSP. In supporting information (Supplementary Fig. 12, 25, 27,
32–34, 69–70 and 72), SEM images of many samples are revised by modified substrate, such
as fresh metastable LSP_{20} – LSP_{3000} , LSP_{20} without ultrasound kept for 12 h, LSP_{20} with
ultrasound kept for different time, LSP_{20} kept for 12 h with different ultrasound time, fresh
SSP_{50} – SSP_{3000} , fresh SSP_{20} (controlled by different SG7_{agg} concentration, SG7_{agg} ratio and
LSP_{seed} concentration), various LSP and corresponding SSP (made of SG7, ABSA, DABSA
and CR) in universality part.

Figure 2. Tuning the size and activity of LSP by LDH confinement. (a–d) SEM images of SG7 SM from different size SG7-LDHs: (a) 20 nm, (b) 50 nm, (c) 100 nm and (d) 3 μm (insets are corresponding magnifications and the LSM sizes in the insets are marked with a yellow line). (e–h) SEM images of SG7 LSP from the corresponding LSM in a–d, respectively. (i–l) Debye plot of metastable LSP made of corresponding LSM in a–d, respectively.

**Figure 4. Living supramolecular polymerization. (a)** The size of SSP₂₀ as a function of
 cycle number x ($x = 1, 2$ and 3). Error bars were calculated using the standard error formalism
 for the data of three repeated experiments (insets: corresponding SEM images of SSP₂₀
 obtained in Cycle 1–3, respectively). **(b–c)** FL spectra of SSP₂₀ obtained in Cycle 1–3.

**Supplementary Figure 12. SEM images of LSP₂₀ with ultrasound kept for different time: (a)**
 **0 h, (b) 1 h, (c) 3 h and (d) 12 h.**

**Supplementary Figure 25.** SEM images of fresh metastable (a) LSP₂₀, (b) LSP₅₀, (c) LSP₁₀₀

and (d) LSP₃₀₀₀.

**Supplementary Figure 27.** SEM images of LSP₂₀ kept for 12 h with different ultrasound

time: (a) 50 min, (b) 60 min and (c) 120 min.

**Supplementary Figure 32.** SEM images of fresh SSP₂₀ made from LSP_{seed} and SG7_{agg} with

different concentrations: (a) 0.400, (b) 2.00 and (c) 3.00 g L⁻¹.

**Supplementary Figure 33.** SEM images of fresh SSP₂₀ by different volume ratio of

SG7_{agg}/LSP_{seed}: (a) 1:1, (b) 2:1 and (c) 4:1.

**Supplementary Figure 34.** SEM images of fresh SSP₂₀ prepared with metastable LSP₂₀ with

different concentrations: (a) 0.250, (b) 2.50 and (c) 5.00 g L⁻¹.

**Supplementary Figure 69.** SEM images of (a) fresh metastable LSP made of ABSA,

corresponding fresh SSPs in (b) Cycle 1 and (c) Cycle 2, and (d) contrast sample: physically

mixing pure ABSA and Cl-LDH in CH₃OH/TFA (1:3 v/v) with the same dosage as LSP.

**Supplementary Figure 70.** SEM images of (a) fresh metastable LSP made of DABSA,

corresponding fresh SSPs in (b) Cycle 1 and (c) Cycle 2, and (d) contrast sample: physically

mixing pure DABSA and Cl-LDH in CH₃OH/TFA (1:3 v/v) with the same dosage as LSP.

**Supplementary Figure 72.** SEM images of (a) fresh metastable LSP made of CR,

corresponding fresh SSPs in (b) Cycle 1 and (c) Cycle 2, and (d) contrast sample: physically

mixing pure CR and Cl-LDH in CH₃OH/TFA (1:3 v/v) with the same dosage as LSP.

9. *The LSP seed was prepared by sonication of metastable LSP. In addition to the duration,*
*the temperature of sonication should be also reported in the supporting information.*

Response: Thank you for your comments. We have supplemented the temperature details in
preparation method of LSP, SSP and co-polymer in supporting information. During
ultrasound, in order to avoid the effect of temperature on the LSP, SSP and co-polymer, the
mixture of ice and water was used to keep a constant temperature at 0 °C.

**p. S8 Line 10:** The metastable LSMs (LSM₂₀, LSM₅₀, LSM₁₀₀ and LSM₃₀₀₀) prepared
from LDHs with different size were synthesized by the same method mentioned above, so
were LSP₂₀–LSP₃₀₀₀. During ultrasound, the mixture of ice and water was used to keep a
constant temperature at 0 °C.

**p. S8 Line 20:** Further adding TFA to CH₃OH/TFA = 1:3 v/v and applying ultrasound for
1 h, the fresh mixed SP₂₀ were formed. During ultrasound, the mixture of ice and water was
used to keep a constant temperature at 0 °C.

**p. S9 Line 6:** All FITS results were tested under magnetic stirring on F7000 ($\lambda_{\text{ex}} = 450$
662 nm, $\lambda_{\text{em}} = 540$ nm, $V_{\text{PMT}} = 700$ V). During ultrasound, the mixture of ice and water was used
to keep a constant temperature at 0 °C.

**p. S9 Line 13:** SSPs (SSP₂₀, SSP₅₀, SSP₁₀₀ and SSP₃₀₀₀) were prepared by the same
method mentioned above from fresh sonicated metastable LSP₂₀–LSP₃₀₀₀. During ultrasound,
the mixture of ice and water was used to keep a constant temperature at 0 °C.

**p. S9 Line 19:** Herein, LSP_{seed} (2.50 g L⁻¹) was made from SG7-LDH (20.0 g L⁻¹ in
CH₃OH), and LSP_{seed} with the other concentrations could be achieved by diluting or
concentrating SG7-LDH in CH₃OH. During ultrasound, the mixture of ice and water was

used to keep a constant temperature at 0 °C.

**p. S10 Line 10:** All samples of SSP were freshly made for SEM. During ultrasound, the
mixture of ice and water was used to keep a constant temperature at 0 °C.

**p. S10 Line 20:** FL spectra were used to precisely test the existence of normally
elongated LSP (yellow light at 530–545 nm), metastable LSM (blue light at ~480 nm) and
polymer-L/D (525–530 nm). During ultrasound, the mixture of ice and water was used to
keep a constant temperature at 0 °C.

**p. S11 Line 7:** Corresponding metastable LSP products or LSP_{seed} were prepared by
adding TFA (300 μL) into the as-prepared intercalated LDH (20.0 g L⁻¹, 100 μL) with
ultrasound for 1 h. During ultrasound, the mixture of ice and water was used to keep a
constant temperature at 0 °C.

10. *In the last part of the result and discussions, the authors demonstrated that ‘The different*
*affinity of chiral molecules during supramolecular polymerization process endows metastable*
*SM the possibility of chiral recognition’. However, as both the monomer and the confined*
*space are achiral, what is the origin of the different affinity of chiral molecules with*
*metastable SM? Is it consistent that Polymer-L₃₀₀₀ always has a shorter lag time than*
*Polymer-D₃₀₀₀, or does the order of lag time varies with different batches? Results of*
*repetitive tests should be presented.*

Response: Thank you for your comments.

● The origin of the different affinity between LSM₃₀₀₀ and chiral Arg (marked as
(LSM₃₀₀₀+L and LSM₃₀₀₀+D, respectively) is the difference in stereo conformation of L-Arg

and D-Arg, which is amplified by our system. For Arg, there are three functional groups (–
NH₂, –OH and –COOH) to form H-bond with LSM. The different stereo conformation of Arg
caused in different steric hindrance in co-polymer formation, which are confirmed by
corresponding optimized geometries in the theoretical calculations (Supplementary Fig. 47).
The results show that LSM₃₀₀₀+L and LSM₃₀₀₀+D have different assembly forms. The
binding energy of LSM₃₀₀₀+D (–471.64 kJ mol^{–1}) is much stronger than LSM₃₀₀₀+SG7
(–52.64 kJ mol^{–1}). The higher (or unmatched) binding energy in co-assembly indicates that
the polymer-D₃₀₀₀ is very slow, even difficult, to form. In contrast, the lower binding energy
of LSM₃₀₀₀+L (–191.50 kJ mol^{–1}) provides basic conditions for polymer-L₃₀₀₀.

● Based on above energy principle, the polymer-L₃₀₀₀ always has a shorter lag time than
polymer-D₃₀₀₀, even in multiple batches of repeated experiments. To confirm that, three
batches LSM₃₀₀₀ samples were prepared and the FL excitation spectra (Supplementary Fig. 48)
were used to judge the co-assembly progress and evaluate the lag time.

For polymer-L₃₀₀₀ in batch 1, the lag time is ~50 min, confirmed by the disappearance of
LSM₃₀₀₀ peak ($\lambda = 409$ nm) and increase of polymer-L₃₀₀₀ peak ($\lambda = 470$ nm) in FL excitation
spectra (Supplementary Fig. 48a). In contrast, polymer-D₃₀₀₀ do not form at the same time yet,
confirmed by the FL excitation spectra with almost no change (Supplementary Fig. 48d). For
polymer-L₃₀₀₀ in batch 2 and 3, although the intensity of LSM₃₀₀₀ in FL excitation spectra
decreases with different rates, the corresponding lag time are always ~50 min,
(Supplementary Fig. 48a–c). Meanwhile, in batch 2 and 3, polymer-D₃₀₀₀ do not form at the
same time yet (Supplementary Fig. 48d–f), making it easy to achieve chiral recognition.
Therefore, chiral recognition results do not vary in different batches.

 **Supplementary Figure 47.** Optimized geometries of (a) LSM₃₀₀₀+L, (b) LSM₃₀₀₀+D, (c)
 highlighted binding area in a, and (d) highlighted binding area in b.

 **Supplementary Figure 48.** FL excitation spectra of chiral recognition products with different
 ultrasound time: (a–c) LSM₃₀₀₀+L in batch 1–3 and (d–f) LSM₃₀₀₀+D in batch 1–3.

 **p. 18 Line 21:** The mechanism of the chiral recognition is that the presence of different
 competitiveness units exerts a strong and different retardation during the assembly kinetics,
 where –NH₂, –OH and –COOH on L- and D-Arg have different stereo conformation and

binding energy (Supplementary Fig. 47). The theoretical calculation shows binding energy
between LSM₃₀₀₀ with SG7 (-52.64 kJ mol⁻¹) is weaker than that of LSM₃₀₀₀ with L-Arg
(-191.50 kJ mol⁻¹) and LSM₃₀₀₀ with D-Arg (-471.64 kJ mol⁻¹), providing basic conditions
for co-assembly (Supplementary Fig. 11h and 47). Further, the difference in the binding
energy between L-Arg and D-Arg gives LSM₃₀₀₀ the ability to recognize the Arg type (Fig.
5c–d).⁶⁴ To verify the reproducibility of chiral recognition, three batches of LSM₃₀₀₀ samples
were prepared, and the results showed that the lag time of corresponding polymer-L₃₀₀₀
formation was 50 min (Supplementary Fig. 48).

11. *The referencing is generally OK but a very early paper on LSP that should be considered*
*for citation as it predates many of the papers referenced is Robinson et al Chem. Comm. 2015,*
*51, 15921-15924.*

Response: Thank you for your comments and positive evaluation of our work. We have
supplemented relevant paper into our ref. 22.

**p. 25 Line 13:**

[22] Robinson, M. E. et al. Length control of supramolecular polymeric nanofibers based on
stacked planar platinum(II) complexes by seeded-growth. *Chem. Comm.* **51**,
15921–15924 (2015).

**Responses to Reviewer 2**

***Overview:** This manuscript reports living supramolecular polymerization (LSP) by arresting*
*the metastable state of the monomer using confinement in a commercially available LDH.*
*They demonstrate well controlled supramolecular polymerization of simple molecules to*
*produce supramolecular polymers with controlled chain length and low dispersity. The*
*confinement approach is indeed interesting and should make lasting impact in the field. I*
*recommend publication after addressing following issues.*

Response: Thank you for your comments. We appreciate that you agree with our preparation
method of well controlled LSP through confinement of LDH. The size and activity of LSP in
our work can be regulated by LDH with different sizes, so as to chiral recognition. Your
comments are greatly helpful. As you suggested, all of the changes in the revised manuscript
are highlighted in red.

*Some other comments:*

*1. The paper is written in rather complex manner which is difficult to read at times. For*
*example, in Fig 1, authors show several molecules which may not be required. Author could*
*discuss the main molecule and the results while at the end to demonstrate general*
*applicability, they may show other structures and results in the subsequent discussion.*

Response: Thank you for your suggestion. Your suggestions are of great help to the
improvement of the article logic. According to your suggestion, the model molecule,
8-hydroxypyrene-1,3,6-trisulfonate (solvent green 7, SG7), is first discussed to demonstrate
the controllability of LSP. In the last section, the universality and its principles of this method

are discussed by employing several small molecules, including benzenesulfonic acid (BSA),
 3-aminobenzene sulfonic acid (3-ABSA), 2,5-diaminobenzenesulfonic acid (2,5-DABSA),
 6,7-dihydroxynaphthalene-2-sulfonate (DHNS) and Congo red (CR). Fig. 1 and Fig. 6 in
 manuscript have been revised.

 **Figure 1.** (a) Schematic presentation for the formation of the LSP. Normalized FL spectra
 and photos under UV light of (b) SG7 and SG7-LDH in CH₃OH; (c) fresh LSM and
 metastable LSP (LSP_{seed}); (d) fresh SSP and LSP. SEM images of (e) metastable LSP (f) LSP
 after aging for 12 h (g) SG7_{agg} and (h) fresh SSP.

Figure 6. Various molecules chosen in our proposed method.

p. 5 Line 14 in manuscript: Here, 8-hydroxypyrene-1,3,6-trisulfonate (solvent green 7,

SG7) is chosen as a model to study for the LSP because it possesses a pyrene molecule

bearing hydrogen-bonding moieties and negative charges, which is expected to self-assemble

via π - π stacking of the pyrene planes and hydrogen-bond (H-bond) of sulfonyl group.

Importantly, there exist remarkable different fluorescence (FL) properties between monomer

and various stacking states (Supplementary Fig. 1–2 for a detailed analysis), providing the

prerequisite for naked eye visualization of assembly events. Fig. 1a depicts the formation

process of LSP (and seed-induced supramolecular polymerization, SSP) from SG7 monomer.

Upon intercalating SG7 monomer into confinement space of LDH, named as SG7-LDH,

which is confirmed by powder X-ray Diffraction (XRD) measurements (step 1,

Supplementary Fig. 3), the ordered arrangement of SG7 were achieved. As confirmed by FL

anisotropy, SG7-LDH showed a high anisotropic value ($r = 0.565$) compared with untreated

SG7 powder ($r = 0.0201$) (Supplementary Fig. 4–5). The *ab initio* molecular dynamics

(AIMD) simulations further testified the ordered arrangement of SG7 with an orientation
angle of $\theta = 9^\circ$ (θ is the inclined angle between the principal axis of molecule and the
elongation axis of LSM, Supplementary Fig. 6). After removal of LDH by using
methanol/trifluoroacetic acid (CH₃OH/TFA) mixture solvent (5:3 v/v), the pre-assembly as
seed are obtained, which is a metastable and living supramonomer (LSM) (step 2). As
confirmed by FL spectra and photos under UV light (Fig. 1b–c, Supplementary Fig. 7–8),
compared with SG7 monomer ($\lambda_{em} = 430$ nm), the pre-assembly LSM ($\lambda_{em} = 460$ nm) showed
red-shifted wavelength, indicating emergence of long-range π - π stacking among LSM;
Compared with SG7-LDHs ($\lambda_{em} = 590$ nm), the pre-assembly LSM ($\lambda_{em} = 460$ nm) showed a
blue-shift, indicating the changes in stacking structure. On the other hand, the metastable and
living properties of LSM can be proved by that LSM can continue to grow to metastable LSP
driven by crystallization with increase poor solvents TFA to CH₃OH/TFA (1:3 v/v), that is,
the step 3. The assembly process results in the larger π - π stacking of metastable LSP than
LSM, which is testified by the redshift of fresh metastable LSP ($\lambda_{em} = 535$ nm) compared
with LSM ($\lambda_{em} = 460$ nm) in FL emission spectra (Fig. 1c, Supplementary Fig. 8–9). The
wide excitation range of LSP is also in line with the long-range π - π stacking rather than a
single molecule (Supplementary Fig. 10). Furthermore, the theoretical calculation shows
LSM prefers to continuously elongate to thermodynamically favorable LSP product due to
the high binding energy of -52.64 kJ mol⁻¹ per SG7 molecule (Supplementary Fig. 11). The
step 4 is an elongation process of metastable LSP by self-replication, as confirmed by
scanning electron microscopy (SEM) images (Fig. 1 e–f and Supplementary Fig. 12).
Compared with metastable LSP, the FL emission spectra of LSP (Supplementary Fig. 9)

show that the decreased FL intensity is attributed to the FL quenching by elongation;
unchanged wavelength ($\lambda_{em} = 535$ nm) is attributed to the maintenance of stacking structure
type. The step 5 is to confirm the character of living polymerization by adding inactive
SG7_{agg} (SG7 dissolved in CH₃OH/TFA (1:3 v/v), 2.5 mM) into metastable LSP (LSP_{seed}) to
form SSP. In FL emission spectra (Supplementary Fig. 13–14), the formed SSP ($\lambda_{em} = 525$
817 nm) shows redshift compared with SG7_{agg} ($\lambda_{em} = 450$ nm), indicating the rearrangement of
818 SG7_{agg} induced by LSP_{seed}. In addition, SSP shows a emission color change and blue shift of
819 10 nm compared with LSP (Fig. 1d), attributing to the tiny difference in stacking states
between SSP and LSP.

**p. 20 Line 4:** As a proof of concept to validate the LDH confinement-driven LSP,
various simple-structural molecules with aromatic rings and negative charge (Fig. 1a) were
chosen to orderly array in confinement space of LDH (step 1, Supplementary Fig. 52–66) and
further performed the assembly process same as SG7 (Supplementary Fig. 67–72). By the
assembly results, the basic principles are summarized as follows: 1) the formation of H-bond
net among released $-NH_2$, $-HSO_3$ and $-OH$. For example, the benzenesulfonic acid (BSA)
with insufficient active hydrogen bond (H-bond) cannot form H-bond net (Supplementary Fig.
53, 56 and 59), leading to the failure in formation of LSP (Supplementary Fig. 68). In
contrast, the appearance of multiple active H-bonds easily forms the LSPs (e.g.,
3-aminobenzene sulfonic acid, 3-ABSA; 2,5-diaminobenzenesulfonic Acid, 2,5-DABSA;
Congo red, CR); 2) oriented long-range π - π stacking *between* neighboring molecules. For
example, the 6,7-dihydroxynaphthalene-2-sulfonate (DHNS) shows the failed formation of
LSP (Supplementary Fig. 71) . The reasons originate from that although the appearance of

π - π stacking of DHNS molecules in the interlayer of LDH as confirmed by UV spectrum
(Supplementary Fig. 54), the π - π stacking is negligible long-range ordered arrangement as
confirmed by a low FL anisotropic value ($r = 0.0214$) (Supplementary Fig. 57). AIMD
simulations, which are referred to XRD, ICP-MS and element analysis results
(Supplementary Fig. 52, Table 1 and Table 6), also further proved the adjacent DHNS
molecules in the interlayer of LDH do have π - π stacking in short-range, attributing to three
functional groups and the large aromatics ring of DHNS. However, they show disordered
arrangement in the long range (Supplementary Fig. 62). Therefore, our approach provides a
good alternative to form oversized LSP by simple-structural molecules.

*2. I am not at all convinced whether confinement in LDH should be so much compared with*
*biomimicking! This is a well-known thing that molecules intercalate in such inorganic layered*
*structures or even clays. Those should not be compared with biomimicking.*

Response: Thank you very much for your suggestion. After our cautious consideration, we
decide to follow your advice and change the original title to “Supramolecular Polymerization
with Simple-Structured Monomers: Confinement Bring It to Life”. In addition, the
Introduction has been rewritten.

**p. 4 Line 9:** The confinement space that can impact all chemical events taking place in a
small cavity have been well documented in the field of nanoreactors, biosensors and drug
delivery vehicles, leading to a contrasting outcome than in the bulk.^{29,30} In terms of
self-assembly, there is evidence that confinement space can promote the formation and
stability of self-assembled complexes held together by intermolecular interactions.³¹⁻³⁶

**p. 4 Line 22:** To validate our hypothesis, the confinement effect of the layered double
hydroxide (LDH) nanomaterial was tried to guide the assembly of simple-structured
monomers by intercalating method.^{41,42} Theoretically, after removal of LDHs template, the
ordered assembly of guests will be destroyed because it has been reported that activation
barrier (30 kcal mol⁻¹) for similar process, spontaneous initiation of styrene polymerization⁴³,
is far beyond the energies associated with non-covalent bond formation.

*3. Referencing should be balanced, recent examples on LSP and particular review articles*
*should be included.*

Response: Thank you for your suggestion. Recent particular examples about LSP have been
supplemented and highlighted in red in the revised manuscript.

**p. 24 Line 3:**

[11] Wagner, W., Wehner, M., Stepanenko, V., Ogi, S. & Würthner, F. Living supramolecular
polymerization of a perylene bisimide dye into fluorescent J-aggregates.
*Angew.Chem.Int. Ed.* **56**, 16008–16012 (2017).

[12] Gao, L., Lin, J., Zhang, L. & Wang, L. Living supramolecular polymerization of rod–
coil block copolymers: Kinetics, origin of uniformity, and its implication. *Nano Lett.* **19**,
3, 2032–2036 (2019).

[13] Qin, B. et al. Supramolecular interfacial polymerization: A controllable method of
fabricating supramolecular polymeric materials. *Angew.Chem. Int.Ed.* **56**, 7639–7643
(2017).

**p. 29 Line 13:**

- [63] Sarkar, S., Sarkar, A. and George, S. J. Stereoselective seed-induced living
supramolecular polymerization. *Angew. Chem. Int. Ed.* **59**, 19841–19845 (2020).
- [64] Ma, X. et al. Fabrication of chiral-selective nanotubular heterojunctions through living
supramolecular polymerization. *Angew. Chem. Int. Ed.* **55**, 9539–9543 (2016).

**Responses to Reviewer 3**

***Overview:** This manuscript reports the controlled self-assembly of small charged molecules. I*
*agree with the latter part of this manuscript on the seeded growth of the assemblies. However,*
*most of the main claims are not supported. Furthermore, it is difficult to understand the*
*contents of this manuscript because the authors mixed up their assumptions (or imaginations)*
*and experimental results. Hence, I recommend the authors to fully revise the manuscript and*
*submit to other journals such as scientific report.*

Response: We are very grateful for your affirmation about the part of our work, which gives
891 us great motivation to further improve this work. We especially thank you for pointing out the
892 shortcomings of our work from a professional perspective, including the distinction of the
893 definition, the idea of articles, writing skills, etc. Your comments have promoted our new
knowledge and understanding for the field of supramolecular polymerization and
self-assembly, and will be of great help to our future work. According to your suggestion, we
have added many new experiments and explanation. The entire article was rewritten, and all
of the changes are highlighted in red.

*Some other comments on the definition of supramolecular polymerization:*

*1. In this manuscript, the authors controlled the self-assemblies of small organic molecules*
*with their developed method using LDH. They considered these assembling processes as*
*supramolecular polymerization. However, it is just crystallization rather than supramolecular*
*polymerization. In particular, the authors named their monomers as MSS (monomer with*
*simple structure). I don't understand why they need to emphasize their monomers by using*

*such special abbreviation. I guess the authors thought that their monomer designs were*
*different from monomers used for the conventional supramolecular polymers. Simply, the*
*assemblies, reported in this manuscript, are not supramolecular polymers and, as a result,*
*the monomer designs are different. I recommend not to describe the reported self-assembling*
*processes as supramolecular polymerization.*

Response: Thank you for your comments. They are very helpful to revise our manuscript
more clearly.

● Thank you very much for proposing the concept of crystallization. Actually, our living
assembly (or polymerization) is a kind of crystallization-driven process. In our original Fig. 1,
we mistakenly used ‘crystallize’ word in the step 3 to demonstrate the crystallization-driven
assembly process, leading to the misunderstanding of readers. To describe this assembly (or
polymerization) process more clearly, we revised the Fig. 1 and supplemented explanation in
revised manuscript. It can be observed that the process includes 5 steps, namely intercalation
of small SG7 molecules (step 1), the release of pre-assembled seed (LSM) (step 2), the
formation of metastable LSP by crystallization-driven assembly, also called as LSP_{seed} (step
3), the elongation of LSP by self-replication of metastable LSP (step 4) and the formation of
SSP induced by LSP_{seed} (step 5).

The step 1 is the order array of intercalated SG7 molecules by confinement space, as
confirmed by XRD, UV, polarized FL profiles, and AIMD simulations (Supplementary Fig. 1,
3, 4 and 6).

The step 2 is to obtain the pre-assembled seed (LSM) made of oriented intercalated SG7
after removing LDH layers. This can be confirmed by comparing the XRD results of samples

(Supplementary Fig. 16), including LSM, untreated SG7 powder, treated SG7 powder in
CH₃OH/TFA (1:3 v/v), Cl-LDH dissolved in TFA and SG7 crystal (JCPDS data base card No:
42-1955). Among all samples, only the XRD pattern of SG7_{agg}, treated SG7 powder in
CH₃OH/TFA (1:3 v/v), shows two peaks in line with SG7 crystal. The XRD pattern of LSM
shows several weak peaks, which are attributed to the metal salts of the LDH dissolved in
TFA. Thus, the SG7 crystal did not appear, indicating the formation process of LSM is an
assembly process, rather than a crystallization process.

The step 3 is to obtain metastable LSP through the crystallization-driven assembly of
LSM, which is confirmed by the redshift of fresh metastable LSP ($\lambda_{em} = 535$ nm) compared
with LSM ($\lambda_{em} = 460$ nm) in FL emission spectra (Fig. 1 and Supplementary Fig. 8–9).
Moreover, the wide excitation range of LSP is also in line with the long-range π - π stacking
rather than a single molecule (Supplementary Fig. 10).

The step 4 is an elongation process of metastable LSP by self-replication. The increased
area in step 4 can be monitored by SEM (Supplementary Fig. 12). Compared with metastable
LSP, the FL emission spectra of LSP (Supplementary Fig. 9) show that the decreased FL
intensity is attributed to the FL quenching by elongation; unchanged wavelength ($\lambda_{em} = 535$
943 nm) is attributed to the maintenance of stacking structure type.

The step 5 is to confirm the character of living polymerization by adding inactive SG7_{agg}
(SG7 dissolved in CH₃OH/TFA (1:3v/v), 2.5 mM) into metastable LSP (LSP_{seed}) to form SSP.
In FL emission spectra (Fig. 1 and Supplementary Fig. 13–14), the formed SSP ($\lambda_{em} = 525$
947 nm) shows redshift compared with SG7_{agg} ($\lambda_{em} = 450$ nm), indicating the rearrangement of
948 SG7_{agg} induced by LSP_{seed}. In addition, SSP shows a blue shift of 10 nm compared with LSP,

attributing to the tiny difference in stacking states between SSP and LSP. In SEM images (Fig.
3–4 and Supplementary Fig. 32–34), the transformation of morphology from spherical
(SG7_{agg}) to rectangular (SSP) and controlled size of SSP in cycle experiments also confirm
the living polymerization induced by LSP. In addition, in the XRD results of LSP (step 4) and
SSP (step 5) (Supplementary Fig. 16), the weak peaks are similar to XRD results of LSM,
attributing to metal salts from LDH, rather than SG7 crystal. Therefore, all steps are not the
formation process of crystal.

● Indeed, it is not appropriate to emphasize these monomers by using such special
abbreviation (MSS). Thus, we have revised the special abbreviation and to simple-structured
monomers in the full text. As you pointed out, we have chosen simple-structured molecules to
realize the living self-assembly (or supramolecular polymerization). Generally, this design
strategy is hardly possible to succeed from an energy principle, because the activation barrier
of simple-structured molecules in the nucleation step is insufficiently high in comparison
with that of the spontaneous nucleation to control the kinetics of subsequent elongation (*Nat.*
*Chem.* **2014**, *6*, 188–195). Therefore, the conventional living supramolecular polymers are
mainly based on complex-structured larger molecules, which require painstaking regulation
and multi-step modification of the monomer structure (*Prog. Polym. Sci.* **2020**, *100*, 101167;
*Chem. Sci.* **2019**, *10*, 6770–6776; *PNAS* **2017**, *114*, 11844–11849). To sum up, we used the
confinement effect of LDH to overcome a huge energy barrier to inhibit spontaneous
nucleation of simple-structured molecules and disassembly of metastable states, which
provided the prerequisites of living polymerization (assembly). Through our preparation
method, simple-structured molecules (ABSA, DABSA, SG7 and CR) can form LSP because

of easily achieved long-range interactions.

**Figure 1.** (a) Schematic presentation for the formation of the LSP. Normalized FL spectra
and photos under UV light of (b) SG7 and SG7-LDH in CH₃OH; (c) fresh LSM and
metastable LSP (LSP_{seed}); (d) fresh SSP and LSP. SEM images of (e) metastable LSP (f) LSP
after aging for 12 h (g) SG7_{agg} and (h) fresh SSP.

**Supplementary Figure 1.** UV-vis spectra of (a) SG7-LDH aqueous dispersion (20.0 g L⁻¹)

with different sizes (optical path = 0.2 mm), **(b)** SG7 solution in CH₃OH with a set of
 concentrations (optical path = 10 mm) and **(c)** SG7 solution in CH₃OH with different
 concentrations (optical path = 0.2 mm).

 **Supplementary Figure 3.** XRD patterns of **(a)** pure LDH precursor powder and **(b)**
 SG7-LDH with different sizes.

 **Supplementary Figure 4.** Polarized FL profiles and anisotropic value (*r*) for **(a)** SG7-LDH₂₀
 (*r* = 0.565), **(b)** SG7-LDH₅₀ (*r* = 0.589), **(c)** SG7-LDH₁₀₀ (*r* = 0.531) and **(d)** SG7-LDH₃₀₀₀ (*r*
 = 0.398) in solid state on the quartz plate, respectively.

 62

**Supplementary Figure 6. (a)** Schematic illustration for the definition of orientation angle, θ .

**(b)** The snapshot of SG7-LDH after AIMD simulation of 100 ps.

**Supplementary Figure 16.** XRD patterns of SG7 SSP (orange), SG7 LSP (red), SG7 LSM

(blue), destroyed CI-LDH in TFA (light gray), SG7_{agg} (gray), SG7 powder (dark gray) and

SG7 crystal (black).

**Supplementary Figure 8. (a)** FL emission spectra and **(b)** corresponding normalized FL

spectra of metastable LSM₂₀–LSM₃₀₀₀ after removal of LDH layers.

**Supplementary Figure 9.** FL spectra of LSP₂₀–LSP₃₀₀₀: (a) freshly sonicated metastable
LSP and (b) stable LSP kept for 12 h after ultrasound.

**Supplementary Figure 10.** FL excitation spectra of LSP₂₀–LSP₃₀₀₀.

**Supplementary Figure 12.** SEM images of LSP₂₀ with ultrasound kept for different time: (a)

0 h, (b) 1 h, (c) 3 h and (d) 12 h.

**Supplementary Figure 13.** FL spectra of fresh SSP₂₀ made from LSP_{seed} with different

concentrations.

Supplementary Figure 14. FL spectrum of SG7_{agg} (2.5 mM).

**Figure 3. Kinetics behavior of LSP₂₀.** (a–b) CLSM images of LSP₂₀: (a) metastable and (b)

kept for 12 h from a. (c) Average size of LSP₂₀ kept for 12 h by different ultrasound time. (d)

FITS of metastable LSP₂₀ without ultrasound. (e–g) The size of SSP as a function of (e) the

concentrations of SG7_{agg}, (f) the volume ratio of added SG7_{agg} to the LSP_{seed}, (g) the amounts

of SG7-LDH. (h) Log–log plot of the rate of increased absorbance at 460 nm as a function of

the LSP_{seed} concentration. Error bars are calculated using the standard error formalism for the

data of three replicate experiments. (i) Time scan of absorbance of SSP₂₀, where LSP_{seed} are

metastable LSP₂₀ with different concentrations (optical path = 1 mm). **(j)** Energy landscapes
 of products during step 1–5.

 **Figure 4. Living supramolecular polymerization.** (a) The size of SSP₂₀ as a function of
 cycle number x ($x = 1, 2$ and 3). Error bars were calculated using the standard error formalism
 for the data of three repeated experiments (insets: corresponding SEM images of SSP₂₀
 obtained in Cycle 1–3, respectively). (b–c) FL spectra of SSP₂₀ obtained in Cycle 1–3.

 **Supplementary Figure 32.** SEM images of fresh SSP₂₀ made from LSP_{seed} and SG7_{agg} with
 different concentrations: (a) 0.400, (b) 2.00 and (c) 3.00 g L⁻¹.

**Supplementary Figure 33.** SEM images of fresh SSP₂₀ by different volume ratio of

SG7_{agg}/LSP_{seed}: (a) 1:1, (b) 2:1 and (c) 4:1.

**Supplementary Figure 34.** SEM images of fresh SSP₂₀ prepared with metastable LSP₂₀ with

different concentrations: (a) 0.250, (b) 2.50 and (c) 5.00 g L⁻¹.

**p. 5 Line 14:** Here, 8-hydroxypyrene-1,3,6-trisulfonate (solvent green 7, SG7) is chosen

as a model to study for the LSP because it possesses a pyrene molecule bearing

hydrogen-bonding moieties and negative charges, which is expected to self-assemble *via* π–π

stacking of the pyrene planes and hydrogen-bond (H-bond) of sulfonyl group. Importantly,

there exist remarkable different fluorescence (FL) properties between monomer and various

stacking states (Supplementary Fig. 1–2 for a detailed analysis), providing the prerequisite

for naked eye visualization of assembly events. Fig. 1a depicts the formation process of LSP

(and seed-induced supramolecular polymerization, SSP) from SG7 monomer. Upon
intercalating SG7 monomer into confinement space of LDH, named as SG7-LDH, which is
confirmed by powder X-ray Diffraction (XRD) measurements (step 1, Supplementary Fig. 3),
the ordered arrangement of SG7 were achieved. As confirmed by FL anisotropy, SG7-LDH
showed a high anisotropic value ($r = 0.565$) compared with untreated SG7 powder ($r =$
0.0201) (Supplementary Fig. 4–5). The *ab initio* molecular dynamics (AIMD) simulations
further testified the ordered arrangement of SG7 with an orientation angle of $\theta = 9^\circ$ (θ is the
inclined angle between the principal axis of molecule and the elongation axis of LSM,
Supplementary Fig. 6). After removal of LDH by using methanol/trifluoroacetic acid
($\text{CH}_3\text{OH}/\text{TFA}$) mixture solvent (5:3 v/v), the pre-assembly as seed are obtained, which is a
metastable and living supramonomer (LSM) (step 2). As confirmed by FL spectra and photos
under UV light (Fig. 1b–c, Supplementary Fig. 7–8), compared with SG7 monomer ($\lambda_{\text{em}} =$
430 nm), the pre-assembly LSM ($\lambda_{\text{em}} = 460$ nm) showed red-shifted wavelength, indicating
emergence of long-range π - π stacking among LSM; Compared with SG7-LDHs ($\lambda_{\text{em}} = 590$
1058 nm), the pre-assembly LSM ($\lambda_{\text{em}} = 460$ nm) showed a blue-shift, indicating the changes in
stacking structure. On the other hand, the metastable and living properties of LSM can be
proved by that LSM can continue to grow to metastable LSP driven by crystallization with
increase poor solvents TFA to $\text{CH}_3\text{OH}/\text{TFA}$ (1:3 v/v), that is, the step 3. The assembly process
results in the larger π - π stacking of metastable LSP than LSM, which is testified by the
redshift of fresh metastable LSP ($\lambda_{\text{em}} = 535$ nm) compared with LSM ($\lambda_{\text{em}} = 460$ nm) in FL
emission spectra (Fig. 1c, Supplementary Fig. 8–9). The wide excitation range of LSP is also
in line with the long-range π - π stacking rather than a single molecule (Supplementary Fig.

10). Furthermore, the theoretical calculation shows LSM prefers to continuously elongate to
thermodynamically favorable LSP product due to the high binding energy of $-52.64 \text{ kJ mol}^{-1}$
1068 per SG7 molecule (Supplementary Fig. 11). The step 4 is an elongation process of metastable
LSP by self-replication, as confirmed by scanning electron microscopy (SEM) images (Fig. 1
e–f and Supplementary Fig. 12). Compared with metastable LSP, the FL emission spectra of
LSP (Supplementary Fig. 9) show that the decreased FL intensity is attributed to the FL
quenching by elongation; unchanged wavelength ($\lambda_{\text{em}} = 535 \text{ nm}$) is attributed to the
maintenance of stacking structure type. The step 5 is to confirm the character of living
polymerization by adding inactive SG7_{agg} (SG7 dissolved in $\text{CH}_3\text{OH}/\text{TFA}$ (1:3 v/v), 2.5 mM)
into metastable LSP (LSP_{seed}) to form SSP. In FL emission spectra (Supplementary Fig.
13–14), the formed SSP ($\lambda_{\text{em}} = 525 \text{ nm}$) shows redshift compared with SG7_{agg} ($\lambda_{\text{em}} = 450 \text{ nm}$),
indicating the rearrangement of SG7_{agg} induced by LSP_{seed} . In addition, SSP shows a
emission color change and blue shift of 10 nm compared with LSP (Fig. 1d), attributing to the
tiny difference in stacking states between SSP and LSP. In SEM images (Fig. 1g–h and
Supplementary Fig. 15), the transformation of morphology from spherical (SG7_{agg}) to
rectangular (SSP) and increased size of SSP in cycle experiments also confirm the living
polymerization induced by LSP. In addition, in the XRD pattern of LSM, LSP and SSP
(Supplementary Fig. 16), the absence of the diffraction peak for SG7 crystal (JCPDS data
base card No: 42-1955) demonstrates their formations were not a crystallization process, but
rather an assembly process.

*2. Even if the reported self-assemblies were considered as supramolecular polymerization,*

*LDH or biomimetic confinement did not induce nor promote supramolecular polymerization.*
*The authors should reconsider the title “Biomimetic confinement driven supramolecular*
*polymerization”.*

Response: Thank you very much for your suggestion. We have revised the title to
“Supramolecular Polymerization with Simple-Structured Monomers: Confinement Bring It to
Life”.

Actually, the confinement of LDH promoted the formation of pre-assembly seed (LSM)
with highly ordered anisotropy and the acquirement of high activity, which provided the
prerequisites of living polymerization (assembly).

Firstly, the highly ordered anisotropy of pre-assembly seed (LSM) in the confinement
space of LDH was confirmed by the XRD, UV, polarized FL profiles and AIMD simulations
(Supplementary Fig. 1, 3, 4, 6 and 52–63).

Secondly, the high activity of LSM was confirmed by the elongation of metastable LSP
and chiral recognition. Due to highly active LSM, metastable LSP can continue to growth
with an increased area over time (Supplementary Fig. 12). Meanwhile, highly active LSM
show unique ability in co-assembly and can amplify the difference between chiral molecules.
The lag time of co-assemble products, polymer-L₃₀₀₀, is always 50 min (Supplementary Fig.
47–48). Meanwhile, polymer-D₃₀₀₀ always does not form. In our system, the activity of LSM
is tunable by LDH with different sizes, leading to the access of co-assembly (by
LSM₂₀–LSM₅₀, (Supplementary Fig. 51)) and amplification of chiral difference (by LSM₃₀₀₀,
(Fig.5 and Supplementary Fig. 44–46)). Therefore, the confinement of LDH not only
endowed LSM with highly ordered arrangement and high activity, but also makes the activity

of LSM controllable by LDH size.

**Supplementary Figure 1.** UV-vis spectra of (a) SG7-LDH aqueous dispersion (20.0 g L⁻¹)

with different sizes (optical path = 0.2 mm), (b) SG7 solution in CH₃OH with a set of

concentrations (optical path = 10 mm) and (c) SG7 solution in CH₃OH with different

concentrations (optical path = 0.2 mm).

**Supplementary Figure 3.** XRD patterns of (a) pure LDH precursor powder and (b)

SG7-LDH with different sizes.

**Supplementary Figure 52.** XRD patterns of BSA-LDH, ABSA-LDH, DABSA-LDH,

DHNS-LDH and CR-LDH.

**Supplementary Figure 53.** UV-vis spectra of intercalated LDH (20 g/L in CH₃OH) and
 corresponding contrast samples (5 mM in CH₃OH) (optical path = 0.1 mm): **(a)** BSA-LDH,
 **(b)** ABSA-LDH and **(c)** DABSA-LDH.

**Supplementary Figure 54.** UV-vis spectra of DHNS-LDH (10 g/L) and DHNS (5 mM) in
 CH₃OH. (optical path = 0.1 mm).

**Supplementary Figure 55.** UV-vis spectra of CR-LDH (2 g/L in CH₃OH) and corresponding
 contrast samples (1 mM in CH₃OH) (optical path = 0.1 mm).

**Supplementary Figure 4.** Polarized FL profiles and anisotropic value (r) for **(a)** SG7-LDH₂₀

($r = 0.565$), **(b)** SG7-LDH₅₀ ($r = 0.589$), **(c)** SG7-LDH₁₀₀ ($r = 0.531$) and **(d)** SG7-LDH₃₀₀₀ (r

= 0.398) in solid state on the quartz plate, respectively.

**Supplementary Figure 56.** Polarized FL profiles and anisotropic value (r) for **(a)** BSA-LDH

($r = 0.104$), **(b)** ABSA-LDH ($r = 0.586$) and **(c)** DABSA-LDH ($r = 0.760$) in solid state on

the quartz plate.

**Supplementary Figure 57.** Polarized FL profiles and anisotropic value (r) for DHNS-LDH

($r = 0.0214$) in solid state on the quartz plate.

**Supplementary Figure 58.** Polarized FL profiles and anisotropic value (r) for CR-LDH ($r =$

0.691) in solid state on the quartz plate.

**Supplementary Figure 6.** (a) Schematic illustration for the definition of orientation angle, θ .

(b) The snapshot of SG7-LDH after AIMD simulation of 100 ps.

**Supplementary Figure 59.** The snapshot of BSA-LDH after AIMD simulation of 100 ps.

**Supplementary Figure 60.** The snapshot of ABSA-LDH after AIMD simulation of 100 ps.

**Supplementary Figure 61.** The snapshot of DABSA-LDH after AIMD simulation of 100 ps.

**Supplementary Figure 62.** The snapshot of DHNS-LDH after AIMD simulation of 100 ps.

**Supplementary Figure 63.** The snapshot of CR-LDH after AIMD simulation of 100 ps.

**Supplementary Figure 64.** Schematic illustration for the definition of θ for (a) BSA, (b)

ABSA, (c) DABSA, (d) DHNS and (e) CR.

**Supplementary Figure 12.** SEM images of LSP₂₀ with ultrasound kept for different time: (a)

0 h, (b) 1 h, (c) 3 h and (d) 12 h.

**Supplementary Figure 47.** Optimized geometries of (a) LSM₃₀₀₀+L, (b) LSM₃₀₀₀+D, (c)

highlighted binding area in a, and (d) highlighted binding area in b.

**Supplementary Figure 48.** FL excitation spectra of chiral recognition products with different

ultrasound time: (a–c) LSM₃₀₀₀+L in batch 1–3 and (d–f) LSM₃₀₀₀+D in batch 1–3.

**Supplementary Figure 51.** Invalid chiral recognition of metastable LSM₂₀ and LSM₅₀ to L-
 or D-Arg. **(a–b)** FL emission spectra and **(c)** excitation spectra of polymer-L₂₀ and
 polymer-D₂₀. **(d–e)** FL emission spectra and **(f)** excitation spectra of polymer-L₅₀ and
 polymer-D₅₀.

 **Figure 5. Application in the chiral recognition.** Photos under UV light of chiral recognition
 products of LSM₃₀₀₀ to **(a)** L-Arg and **(b)** D-Arg during co-polymerization process, noted that
 only ultrasound for 50 min. **(c–d)** Optimized geometries of LSM₃₀₀₀ to **(c)** L-Arg and **(d)**
 D-Arg.

 **Supplementary Figure 44.** FL emission spectra of chiral recognition products of LSM₃₀₀₀+L
 and LSM₃₀₀₀+D after ultrasound for 50 min (insets: corresponding normalized spectra).

**Supplementary Figure 45.** FL excitation spectra of chiral recognition products of

LSM₃₀₀₀+L and LSM₃₀₀₀+D.

**Supplementary Figure 46.** CLSM images of formed polymer-L₃₀₀₀ at 50 min by the

co-assembly of LSM₃₀₀₀ and L-Arg: (a) $\lambda_{\text{laser diode}} = 405$ nm, (b) $\lambda_{\text{laser diode}} = 488$ nm. CLSM

images of chiral products of LSP₃₀₀₀+D at 50 min: (c) $\lambda_{\text{laser diode}} = 405$ nm and (d) $\lambda_{\text{laser diode}} =$

488 nm.

**p. 5 Line 20:** Fig. 1a depicts the formation process of LSP (and seed-induced

supramolecular polymerization, SSP) from SG7 monomer. Upon intercalating SG7 monomer

into confinement space of LDH, named as SG7-LDH, which is confirmed by powder X-ray

diffraction (XRD) measurements (step 1, Supplementary Fig. 3), the ordered arrangement of

SG7 were achieved. As confirmed by FL anisotropy, SG7-LDH showed a high anisotropic

value ($r = 0.565$) compared with untreated SG7 powder ($r = 0.0201$) (Supplementary Fig.
4–5). The *ab initio* molecular dynamics (AIMD) simulations further testified the ordered
arrangement of SG7 with an orientation angle of $\theta = 9^\circ$ (θ is the inclined angle between the
principal axis of molecule and the elongation axis of LSM, Supplementary Fig. 6). After
removal of LDH by using methanol/trifluoroacetic acid (CH₃OH/TFA) mixture solvent (5:3
v/v), the pre-assembly as seed are obtained, which is a metastable and living supramonomer
(LSM) (step 2). As confirmed by FL spectra and photos under UV light (Fig. 1b–c,
Supplementary Fig. 7–8), compared with SG7 monomer ($\lambda_{em} = 430$ nm), the pre-assembly
LSM ($\lambda_{em} = 460$ nm) showed red-shifted wavelength, indicating emergence of long-range π - π
stacking among LSM; Compared with SG7-LDHs ($\lambda_{em} = 590$ nm), the pre-assembly LSM
($\lambda_{em} = 460$ nm) showed a blue-shift, indicating the changes in stacking structure. On the other
hand, the metastable and living properties of LSM can be proved by that LSM can continue
to grow to metastable LSP driven by crystallization with increase poor solvents TFA to
CH₃OH/TFA (1:3 v/v), that is, the step 3.

**p. 19 Line 9:** Importantly, the chiral recognition effectiveness from LSM₂₀, LSM₅₀ and
LSM₁₀₀ and LSM₃₀₀₀ is quite distinct (Supplementary Fig. 49–51 and the detailed analysis).
For the smaller LSM₂₀ and LSM₅₀ with higher activity, chiral recognition is difficult to realize
due to indistinguishable lag time. For LSM₁₀₀, the identification can only be seen within 20
1216 min. Thus, the higher recognition ability of LSM₃₀₀₀ maybe origin in relatively maximized
kinetics effects and minimal activity.

*Some other comments on the scientific discussion:*

*3. I am an expert of supramolecular polymerization have read so many related papers. This*
*manuscript is one of the worst papers in terms of writing skill. This manuscript contains some*
*interesting results and is grammatically fine. However, it is very difficult to understand the*
*contents scientifically because I could not easily understand with which data the authors*
*support their claims.*

Response: Thank you very much for your comments. They are greatly helpful for improving
the quality of our manuscript. Therefore, we sufficiently revised our manuscript according to
your comments. Especially, thank you for pointing out our shortcomings in writing skills,
which prompted us to modify the layout of the entire article.

For example, Fig. 1 in original manuscript shows several molecules, which is difficult to
read at times. Thus, we have chosen the 8-hydroxypyrene-1,3,6-trisulfonate (solvent green 7,
SG7) as model molecule, which is first discussed to demonstrate the controllability of LSP
(Fig. 1). Then, in the last section in the revised manuscript, the universality and its principle
of this method are discussed by employing several small molecules, including
benzenesulfonic acid (BSA), 3-aminobenzene sulfonic acid (3-ABSA),
2,5-diaminobenzenesulfonic acid (2,5-DABSA), 6,7-dihydroxynaphthalene-2-sulfonate
(DHNS) and Congo red (CR) (Fig. 6).

**Figure 1. (a)** Schematic presentation for the formation of the LSP. Normalized FL spectra

and photos under UV light of **(b)** SG7 and SG7-LDH in CH₃OH; **(c)** fresh LSM and

metastable LSP (LSP_{seed}); **(d)** fresh SSP and LSP. SEM images of **(e)** metastable LSP **(f)** LSP

after aging for 12 h **(g)** SG7_{agg} and **(h)** fresh SSP.

**Figure 6.** Various molecules chosen in our proposed method.

3.1 For example, from the beginning of the “Results and discussion”, the authors claim that
MSS are orderly arrayed in LDH. I think most of the readers wonder why and how the
authors probed it. As far as I found, their claims are based on their previous work in 2009
(ref 36). However, this previous work just discusses the orientation of the other small
molecule. Hence, it is difficult to claim that MSSs used in this manuscript were also arrayed
orderly in LDH. This point may not be critically important to discuss the possible mechanism.
However, such scientifically inappropriate discussion makes us difficult trust the results and
discussions.

Response: Thank you very much for your comments. We have supplemented the data about
the order array of all small molecules in the interlayer of LDH, including XRD, UV-vis
spectra, polarized FL profiles and anisotropy, ICP-MS, element analysis and AIMD
simulations.

XRD results show the successful intercalation of small molecules into the interlayer of
LDH, named as SG7-LDH, BSA-LDH, ABSA-LDH, DABSA-LDH, DHNS-LDH and
CR-LDH (Supplementary Fig. 1, 3 4, 6 and 52–63). Before intercalation, Cl-LDH shows
diffraction peaks of (003), (006), (009) and (110) at ca. 11.3°, 22.4°, 35.0° and 60.9°,
respectively (Supplementary Fig. 3). According to the Bragg equation of $2d\sin\theta=n\lambda$, where d
is the spacing between crystal plane, θ is the angle between the incident X-ray and the
corresponding crystal plane, n is diffraction series ($n = 1$) and λ is wavelength of X-ray ($\lambda =$
1.54 Å). Therefore, the d -spacing of Cl-LDH was calculated to 0.782 nm. After intercalation,
BSA-LDH, ABSA-LDH, DABSA-LDH, DHNS-LDH and CR-LDH show the d -spacing

increased to 1.48 nm, 1.57 nm, 1.45 nm, 1.22 nm and 0.877 nm, respectively. The increased
*d*-spacing indicates the successful intercalation of small molecules into the interlayer of LDH.

UV spectra of SG7-LDH ($\Delta\lambda = \sim 100$ nm), ABSA-LDH ($\Delta\lambda = \sim 5$ nm), DABSA-LDH ($\Delta\lambda =$
~ 75 nm), DHNS-LDH ($\Delta\lambda = \sim 10$ nm) and CR-LDH ($\Delta\lambda = \sim 5$ nm) showed varying degrees of
red shift relative to their respective monomers (Supplementary Fig. 1 and 53–55). The
redshifts are attributed to stronger π - π stacking interactions between intercalated molecules
than isotropic monomers in CH₃OH. For intercalated BSA in LDH, there is no enhanced π - π
stacking due to lack of functional group, confirmed by the absence of red shift in UV spectra.

From the polarized FL profiles in Supplementary Fig. 4, SG7-LDH ($r = 0.565$) shows
higher anisotropic value than untreated SG7 powder ($r = 0.0201$, Supplementary Fig. 5),
indicating the ordered arrangement of SG7 molecules in the interlayer of LDH. Similarly,
ABSA-LDH ($r = 0.586$), DABSA-LDH ($r = 0.760$) and CR-LDH ($r = 0.691$) show high
anisotropic value, confirming their ordered arrangement in the interlayer of LDHs
(Supplementary Fig. 56–58). The low anisotropic value of BSA-LDH ($r = 0.104$) and
DHNS-LDH ($r = 0.0214$) demonstrate the lack of ordered arrangement.

To further confirm the above experimental results, the AIMD simulations were performed
on these intercalated LDH. The chemical formula were $\text{Mg}_{48}\text{Al}_{16}(\text{OH})_{128}(\text{C}_{16}\text{H}_6\text{O}_{10}\text{S}_3)_4$,
$\text{Mg}_{48}\text{Al}_{16}(\text{OH})_{128}(\text{C}_6\text{H}_5\text{SO}_3)_{16}$, $\text{Mg}_{48}\text{Al}_{16}(\text{OH})_{128}(\text{C}_6\text{H}_6\text{NSO}_3)_{16}$,
$\text{Mg}_{48}\text{Al}_{16}(\text{OH})_{128}(\text{C}_6\text{H}_7\text{N}_2\text{SO}_3)_{16}$, $\text{Mg}_{48}\text{Al}_{16}(\text{OH})_{128}(\text{C}_{10}\text{H}_5\text{SO}_5)_5\text{Cl}$ and
$\text{Mg}_{48}\text{Al}_{16}(\text{OH})_{128}(\text{C}_{32}\text{H}_{22}\text{N}_6\text{S}_2\text{O}_6)_2(\text{Cl})_{12}$ according to the results from ICP-MS, element
analysis, respectively (Supplementary Table 1 and 6). The lattice parameters ($a = b = 2d_{(110)}$, c
$= d_{(003)}$) were referred to the corresponding XRD results. From the AIMD simulations in

Supplementary Fig. 6 and 56–58, the adjacent intercalated molecules (SG7, ABSA, DABSA
and CR) in the interlayer of LDH have π - π stacking in short-range, attributing to functional
groups and the large aromatics ring of intercalated molecules. In addition, these π - π stacking
shows highly ordered arrangement in the long range. The AIMD simulations of BSA-LDH
and DHNS-LDH show the disordered arrangement in long range, which are in consistent with
the low anisotropic value of BSA-LDH ($r = 0.104$) and DHNS-LDH ($r = 0.0214$) in polarized
FL profiles. Furthermore, we also give the precise angle θ of the arrangement of these small
molecules in interlayer of LDH. The θ for SG7, ABSA, DABSA and CR can be calculated to
be 9° , 43° , 86° and 0° , respectively.

On the basis of above results, the successful formation of LSP is based on the ordered
arrangement of intercalated molecules in the confinement space of LDH. In contrast,
disordered intercalated molecules fail to promote the formation process of LSP.

*3.2 There are so many similar inappropriate discussion (or scientifically not supported*
*claims) in this manuscript. I don't have enough time to specially mention one by one but here*
*I mention another example. The authors claimed that DHNS monomer could not realize*
*desired supramolecular polymerization due to insufficient π - π stackings owing to its*
*orientational angle with Supplementary figure 8. However, it is impossible to discuss the*
*orientational angle with these figures in FigS.8. I could not understand why the authors could*
*get such conclusion. In particular, DHNS has two hydroxy groups while none of the other*
*monomers possess hydroxy group. These catechol moieties are known to strongly coordinate*
*to transition metal ions. Hence, although I don't know the true reason, the authors have to*

*show the solid evidence to support their claim.*

● Thank you very much for your comments. It is indeed hard to explain the failed
formation of DHNS LSP solely based on SEM images. Thus, we have supplemented the data
about the array of DHNS in LDH, including XRD, UV-vis spectra, polarized fluorescence
and anisotropy, ICP-MS, element analysis and AIMD simulations.

XRD results show the successful intercalation of DHNS into LDHs layers, named as
DHNS-LDH (Supplementary Fig. 3 and 52). Before intercalation, the *d*-spacing of Cl-LDH
was calculated to be 0.782 nm. The *d*-spacing was increased to 1.22 nm after the DHNS
intercalation. The increased *d*-spacing indicates the successful intercalation of DHNS into
LDH layers. UV spectrum of the DHNS-LDH (20 g/L in CH₃OH) shows red shift of 10 nm
than DHNS (5 mM in CH₃OH), demonstrating the appearance of π - π stacking from DHNS
molecules in the interlayer of LDH (Supplementary Fig. 54). Nevertheless, the low FL
anisotropic value ($r = 0.0214$) displays that the π - π stacking structure is negligible long-range
ordered arrangement (Supplementary Fig. 5).

To further confirm the above experimental results, the AIMD simulations of DHNS-LDH
were performed. The chemical formula of DHNS-LDH was Mg₄₈Al₁₆(OH)₁₂₈(C₁₀H₅SO₅)₅Cl
according to ICP-MS and element analysis (Supplementary Table 1–2). The lattice
parameters ($a = b = 2d_{(110)} = 3.07 \text{ \AA}$, $c = d_{(003)} = 7.82 \text{ \AA}$) were referred to the corresponding
XRD results. From the AIMD simulations results in Supplementary Fig. 62, the adjacent
DHNS molecules in the interlayer of LDH do have π - π stacking in short-range, attributing to
three functional groups and the large aromatics ring of DHNS. However, they show
disordered arrangement in the long range.

On the basis of above results, DHNS molecules fail to orderly arrange in the
confinement space of LDH. Therefore, it is impossible for disordered DHNS molecules to
overcome a huge energy barrier from spontaneous nucleation to living elongate, leading to
the failure of formation of LSP.

**Supplementary Figure 3.** XRD patterns of (a) pure LDH precursor powder and (b)
SG7-LDH with different sizes.

**Supplementary Figure 52.** XRD patterns of BSA-LDH, ABSA-LDH, DABSA-LDH,
DHNS-LDH and CR-LDH.

**Supplementary Figure 1.** UV-vis spectra of (a) SG7-LDH aqueous dispersion (20.0 g L^{-1})

with different sizes (optical path = 0.2 mm), **(b)** SG7 solution in CH₃OH with a set of
concentrations (optical path = 10 mm) and **(c)** SG7 solution in CH₃OH with different
concentrations (optical path = 0.2 mm).

**Supplementary Figure 53.** UV-vis spectra of intercalated LDH (20 g/L in CH₃OH) and
corresponding contrast samples (5 mM in CH₃OH) (optical path = 0.1 mm): **(a)** BSA-LDH,
**(b)** ABSA-LDH and **(c)** DABSA-LDH.

**Supplementary Figure 54.** UV-vis spectra of DHNS-LDH (10 g/L) and DHNS (5 mM) in
CH₃OH. (optical path = 0.1 mm).

**Supplementary Figure 55.** UV-vis spectra of CR-LDH (2 g/L in CH₃OH) and corresponding
contrast samples (1 mM in CH₃OH) (optical path = 0.1 mm).

**Supplementary Figure 4.** Polarized FL profiles and anisotropic value (r) for **(a)** SG7-LDH₂₀

($r = 0.565$), **(b)** SG7-LDH₅₀ ($r = 0.589$), **(c)** SG7-LDH₁₀₀ ($r = 0.531$) and **(d)** SG7-LDH₃₀₀₀ (r

$= 0.398$) in solid state on the quartz plate, respectively.

**Supplementary Figure 56.** Polarized FL profiles and anisotropic value (r) for **(a)** BSA-LDH

($r = 0.104$), **(b)** ABSA-LDH ($r = 0.586$) and **(c)** DABSA-LDH ($r = 0.760$) in solid state on

the quartz plate, respectively.

**Supplementary Figure 57.** Polarized FL profiles and anisotropic value (r) for DHNS-LDH

in solid state on the quartz plate. ($r = 0.0214$) in solid state on the quartz plate.

**Supplementary Figure 58.** Polarized FL profiles and anisotropic value (r) for CR-LDH ($r =$

0.691) in solid state on the quartz plate.

**Supplementary Figure 5.** Polarized FL profiles and anisotropic value (r) for untreated SG7

powder ($r = 0.0201$).

**Supplementary Table 1.** The ICP-MS results of pure Cl-LDH precursors with different sizes

in our work, the calculated ratio of Mg/Al and corresponding chemical formula.

Various LDHs	Mg (ppm)	Al (ppm)	Mg/Al	$[\text{Mg}_a\text{Al}_b(\text{OH})_x](\text{A}^{n-})_y \cdot z\text{H}_2\text{O}$	
				a	b
LDH ₂₀	3.89	1.41	3.06	0.75	0.25
LDH ₅₀	5.02	1.77	3.15	0.76	0.24
LDH ₁₀₀	6.49	2.44	2.95	0.75	0.25
LDH ₃₀₀₀	4.19	1.65	2.82	0.74	0.26

**Supplementary Table 2.** The elemental analysis results of all intercalated LDHs in our work
 and the calculated weight fraction of intercalated molecules and corresponding chemical
 formula.

Various LDHs	S (wt.%)	O (wt.%)	H (wt.%)	A ⁿ⁻ (wt.%)	[Mg _a Al _b (OH) _x](A ⁿ⁻) _y ·zH ₂ O		
					x	y	z
SG7-LDH ₂₀	5.47	33.40	4.73	25.93	1.93	0.08	2.2
SG7-LDH ₅₀	5.72	33.90	4.40	27.11	1.80	0.11	2.9
SG7-LDH ₁₀₀	5.69	33.61	4.54	26.97	1.85	0.10	2.5
SG7-LDH ₃₀₀₀	5.70	33.63	4.60	27.02	1.90	0.09	2.4
BSA-LDH ₂₀	5.07	32.01	5.45	25.03	2.03	0.22	2.1
ABSA-LDH ₂₀	5.21	29.94	5.11	28.17	1.98	0.27	2.3
DABSA-LDH ₂₀	4.90	29.34	4.98	28.79	1.96	0.29	2.6
DHNS-LDH ₂₀	3.32	24.05	3.94	24.80	2.04	0.08	2.5
CR-LDH ₂₀	3.54	29.19	5.22	35.95	1.80	0.10	2.7

**Supplementary Figure 6.** (a) Schematic illustration for the definition of orientation angle, θ .

(b) The snapshot of SG7-LDH after AIMD simulation of 100 ps.

**Supplementary Figure 59.** The snapshot of BSA-LDH after AIMD simulation of 100 ps.

**Supplementary Figure 60.** The snapshot of ABSA-LDH after AIMD simulation of 100 ps.

**Supplementary Figure 61.** The snapshot of DABSA-LDH after AIMD simulation of 100 ps.

**Supplementary Figure 62.** The snapshot of DHNS-LDH after AIMD simulation of 100 ps.

**Supplementary Figure 63.** The snapshot of CR-LDH after AIMD simulation of 100 ps.

**Supplementary Figure 64.** Schematic illustration for the definition of θ for (a) BSA, (b)

ABSA, (c) DABSA, (d) DHNS and (e) CR.

**p. 5 Line 14 in manuscript:** Here, 8-hydroxypyrene-1,3,6-trisulfonate (solvent green 7,
 SG7) is chosen as a model to study for the LSP because it possesses a pyrene molecule
 bearing hydrogen-bonding moieties and negative charges, which is expected to self-assemble
 *via* π - π stacking of the pyrene planes and hydrogen-bond (H-bond) of sulfonyl group.
 Importantly, there exist remarkable different fluorescence (FL) properties between monomer
 and various stacking states (Supplementary Fig. 1–2 for a detailed analysis), providing the
 prerequisite for naked eye visualization of assembly events. Fig. 1a depicts the formation
 process of LSP (and seed-induced supramolecular polymerization, SSP) from SG7 monomer.
 Upon intercalating SG7 monomer into confinement space of LDH, named as SG7-LDH,
 which is confirmed by powder X-ray Diffraction (XRD) measurements (step 1,

Supplementary Fig. 3), the ordered arrangement of SG7 were achieved. As confirmed by FL
anisotropy, SG7-LDH showed a high anisotropic value ($r = 0.565$) compared with untreated
SG7 powder ($r = 0.0201$) (Supplementary Fig. 4–5). The *ab initio* molecular dynamics
(AIMD) simulations further testified the ordered arrangement of SG7 with an orientation
angle of $\theta = 9^\circ$ (θ is the inclined angle between the principal axis of molecule and the
elongation axis of LSM, Supplementary Fig. 6). After removal of LDH by using
methanol/trifluoroacetic acid (CH₃OH/TFA) mixture solvent (5:3 v/v), the pre-assembly as
seed are obtained, which is a metastable and living supramonomer (LSM) (step 2). As
confirmed by FL spectra and photos under UV light (Fig. 1b–c, Supplementary Fig. 7–8),
compared with SG7 monomer ($\lambda_{em} = 430$ nm), the pre-assembly LSM ($\lambda_{em} = 460$ nm) showed
red-shifted wavelength, indicating emergence of long-range π - π stacking among LSM;
Compared with SG7-LDHs ($\lambda_{em} = 590$ nm), the pre-assembly LSM ($\lambda_{em} = 460$ nm) showed a
blue-shift, indicating the changes in stacking structure. On the other hand, the metastable and
living properties of LSM can be proved by that LSM can continue to grow to metastable LSP
driven by crystallization with increase poor solvents TFA to CH₃OH/TFA (1:3 v/v), that is,
the step 3. The assembly process results in the larger π - π stacking of metastable LSP than
LSM, which is testified by the redshift of fresh metastable LSP ($\lambda_{em} = 535$ nm) compared
with LSM ($\lambda_{em} = 460$ nm) in FL emission spectra (Fig. 1c, Supplementary Fig. 8–9). The
wide excitation range of LSP is also in line with the long-range π - π stacking rather than a
single molecule (Supplementary Fig. 10). Furthermore, the theoretical calculation shows
LSM prefers to continuously elongate to thermodynamically favorable LSP product due to
the high binding energy of -52.64 kJ mol⁻¹ per SG7 molecule (Supplementary Fig. 11). The

step 4 is an elongation process of metastable LSP by self-replication, as confirmed by
scanning electron microscopy (SEM) images (Fig. 1 e–f and Supplementary Fig. 12).
Compared with metastable LSP, the FL emission spectra of LSP (Supplementary Fig. 9)
show that the decreased FL intensity is attributed to the FL quenching by elongation;
unchanged wavelength ($\lambda_{\text{em}} = 535 \text{ nm}$) is attributed to the maintenance of stacking structure
type. The step 5 is to confirm the character of living polymerization by adding inactive
SG7_{agg} (SG7 dissolved in CH₃OH/TFA (1:3 v/v), 2.5 mM) into metastable LSP (LSP_{seed}) to
form SSP. In FL emission spectra (Supplementary Fig. 13–14), the formed SSP ($\lambda_{\text{em}} = 525$
1441 nm) shows redshift compared with SG7_{agg} ($\lambda_{\text{em}} = 450 \text{ nm}$), indicating the rearrangement of
1442 SG7_{agg} induced by LSP_{seed}. In addition, SSP shows a emission color change and blue shift of
1443 10 nm compared with LSP (Fig. 1d), attributing to the tiny difference in stacking states
between SSP and LSP.

**p. 20 Line 4:** As a proof of concept to validate the LDH confinement-driven LSP,
various simple-structural molecules with aromatic rings and negative charge (Fig. 1a) were
chosen to orderly array in confinement space of LDH (step 1, Supplementary Fig. 52–66) and
further performed the assembly process same as SG7 (Supplementary Fig. 67–72). By the
assembly results, the basic principles are summarized as follows: 1) the formation of H-bond
net among released $-\text{NH}_2$, $-\text{HSO}_3$ and $-\text{OH}$. For example, the benzenesulfonic acid (BSA)
with insufficient active hydrogen bond (H-bond) cannot form H-bond net (Supplementary Fig.
53, 56 and 59), leading to the failure in formation of LSP (Supplementary Fig. 68). In
contrast, the appearance of multiple active H-bonds easily forms the LSPs (e.g.,
3-aminobenzene sulfonic acid, 3-ABSA; 2,5-diaminobenzenesulfonic Acid, 2,5-DABSA;

Congo red, CR); 2) oriented long-range π - π stacking *between* neighboring molecules. For
example, the 6,7-dihydroxynaphthalene-2-sulfonate (DHNS) shows the failed formation of
LSP (Supplementary Fig. 71) . The reasons originate from that although the appearance of
π - π stacking of DHNS molecules in the interlayer of LDH as confirmed by UV spectrum
(Supplementary Fig. 54), the π - π stacking is negligible long-range ordered arrangement as
confirmed by a low FL anisotropic value ($r = 0.0214$) (Supplementary Fig. 57). AIMD
simulations, which are referred to XRD, ICP-MS and element analysis results
(Supplementary Fig. 52, Table 1 and Table 6), also further proved the adjacent DHNS
molecules in the interlayer of LDH do have π - π stacking in short-range, attributing to three
functional groups and the large aromatics ring of DHNS. However, they show disordered
arrangement in the long range (Supplementary Fig. 62). Therefore, our approach provides a
good alternative to form oversized LSP by simple-structural molecules.

*4. The authors have not quantified the amount of the incorporated MSSs in LDH. However,*
*the authors must confirm that how much monomers were included in LDH, otherwise it is*
*difficult to discuss the mechanisms with these materials.*

Response: Thank you for your comment. As you suggested, we quantify the amount of
intercalated molecules (Supplementary Table 1–2). Based on above results, the universal
chemical formula of intercalated LDH, $[\text{Mg}_a\text{Al}_b(\text{OH})_x](\text{A}^{n-})_y \cdot z\text{H}_2\text{O}$ (A stands for the
intercalated molecules), can be calculated in detail. As seen in the Supplementary Table 1–2,
SG7-LDH with different sizes show similar weight fraction of SG7 in SG7-LDH (~25 wt.%),
indicating that amount of intercalation does not vary with LDH size.

In detail, according to the results of ICP and element analysis, for the SG7-LDH₂₀,
SG7-LDH₅₀, SG7-LDH₁₀₀, SG7-LDH₃₀₀₀, BSA-LDH₂₀, ABSA-LDH₂₀, DABSA-LDH₂₀,
DHNS-LDH₂₀ and CR-LDH₂₀, the chemical formula are calculated to be
$[\text{Mg}_{0.75}\text{Al}_{0.25}(\text{OH})_{1.93}](\text{SG7})_{0.08}\cdot 2.2\text{H}_2\text{O}$, $[\text{Mg}_{0.76}\text{Al}_{0.24}(\text{OH})_{1.80}](\text{SG7})_{0.11}\cdot 2.9\text{H}_2\text{O}$,
$[\text{Mg}_{0.75}\text{Al}_{0.25}(\text{OH})_{1.85}](\text{SG7})_{0.10}\cdot 2.5\text{H}_2\text{O}$, $[\text{Mg}_{0.74}\text{Al}_{0.26}(\text{OH})_{1.90}](\text{SG7})_{0.09}\cdot 2.4\text{H}_2\text{O}$,
$[\text{Mg}_{0.75}\text{Al}_{0.25}(\text{OH})_{2.03}](\text{BSA})_{0.22}\cdot 2.1\text{H}_2\text{O}$, $[\text{Mg}_{0.75}\text{Al}_{0.25}(\text{OH})_{1.98}](\text{ABSA})_{0.27}\cdot 2.3\text{H}_2\text{O}$,
$[\text{Mg}_{0.75}\text{Al}_{0.25}(\text{OH})_{1.96}](\text{DABSA})_{0.29}\cdot 2.6\text{H}_2\text{O}$, $[\text{Mg}_{0.75}\text{Al}_{0.25}(\text{OH})_{2.04}](\text{DHNS})_{0.08}\cdot 2.5\text{H}_2\text{O}$ and
$[\text{Mg}_{0.75}\text{Al}_{0.25}(\text{OH})_{1.80}](\text{CR})_{0.1}\text{Cl}_{0.25}\cdot 2.7\text{H}_2\text{O}$, respectively. In addition, the theoretical model
diagram of these intercalated LDHs are presented in Supplementary Fig. 65–66.

**Supplementary Table 1.** The ICP-MS results of pure Cl-LDH precursors with different sizes
in our work, the calculated ratio of Mg/Al and corresponding chemical formula.

Various LDHs	Mg (ppm)	Al (ppm)	Mg/Al	$[\text{Mg}_a\text{Al}_b(\text{OH})_x](\text{A}^{n-})_y\cdot z\text{H}_2\text{O}$	
				a	b
LDH ₂₀	3.89	1.41	3.06	0.75	0.25
LDH ₅₀	5.02	1.77	3.15	0.76	0.24
LDH ₁₀₀	6.49	2.44	2.95	0.75	0.25
LDH ₃₀₀₀	4.19	1.65	2.82	0.74	0.26

**Supplementary Table 2.** The elemental analysis results of all intercalated LDHs in our work
 and the calculated weight fraction of intercalated molecules and corresponding chemical
 formula.

Various LDHs	S	O	H	A ⁿ⁻	[Mg _a Al _b (OH) _x](A ⁿ⁻) _y ·zH ₂ O		
	(wt.%)	(wt.%)	(wt.%)	(wt.%)	x	y	z
SG7-LDH ₂₀	5.47	33.40	4.73	25.93	1.93	0.08	2.2
SG7-LDH ₅₀	5.72	33.90	4.40	27.11	1.80	0.11	2.9
SG7-LDH ₁₀₀	5.69	33.61	4.54	26.97	1.85	0.10	2.5
SG7-LDH ₃₀₀₀	5.70	33.63	4.60	27.02	1.90	0.09	2.4
BSA-LDH ₂₀	5.07	32.01	5.45	25.03	2.03	0.22	2.1
ABSA-LDH ₂₀	5.21	29.94	5.11	28.17	1.98	0.27	2.3
DABSA-LDH ₂₀	4.90	29.34	4.98	28.79	1.96	0.29	2.6
DHNS-LDH ₂₀	3.32	24.05	3.94	24.80	2.04	0.08	2.5
CR-LDH ₂₀	3.54	29.19	5.22	35.95	1.80	0.10	2.7

**Supplementary Figure 65.** The top view of intercalated LDHs: **(a)** SG7-LDH, **(b)**

BSA-LDH, **(c)** ABSA-LDH, **(d)** DABSA-LDH, **(e)** DHNS-LDH and **(f)** CR-LDH.

**Supplementary Figure 66.** The side view of intercalated LDHs: **(a)** SG7-LDH, **(b)**
 BSA-LDH, **(c)** ABSA-LDH, **(d)** DABSA-LDH, **(e)** DHNS-LDH and **(f)** CR-LDH.

**p. S30 Line 10:** In SG7-LDH₂₀ system, SG7 accounts for 26.32 wt.%. Among them, the
 adsorption capacity of SG7 on the outer surface of CO₃-LDH (named as SG7-LDH₂₀-surface)
 is 3.11 wt.%; the intercalation capacity of SG7 in interlayer of LDH is 23.21 wt.%. That is,
 the adsorption capacity only accounts for 13.41% of the total SG7.

**p. 7 Line 18:** To quantitatively describe the effect of SG7 adsorption on assembly, we
 employed the CO₃-LDH₂₀ (the interlayer anions of LDH is CO₃²⁻) to adsorb SG7 (marked as
 SG7-LDH₂₀-surface, detailed in Supporting Information), where the intercalation does not

occur because the affinity between CO_3^{2-} and LDH layer is much greater than that between
organic simple-structured molecules and LDH layer.^{44,45} ICP-MS and element analysis results
show the adsorption capacity is very low and only accounts for 13.41% of the total SG7
(Supplementary Table 1–2).

**p. 20 Line 19:** AIMD simulations, which are referred to XRD, ICP-MS and element
analysis results (Supplementary Fig. 52, Table 1 and Table 6), also further proved the
adjacent DHNS molecules in the interlayer of LDH do have π - π stacking in short-range,
attributing to three functional groups and the large aromatics ring of DHNS.

*5. This is the most import comment from me. The authors claimed that they could control the*
*size of metastable LSP with the size of LDH.*

*5.1 In their proposed mechanism, they imagined that the sizes of SM were controlled by the*
*size of LDH. However, there is no direct evidence to support this claim, because SMs were*
*aggregated into big objects in Fig. 2a-d whose sizes are way much bigger than those of the*
*corresponding LDHs.*

Response: Thank you for your comments. Your comments make us aware of this problem
existing in SEM images throughout the manuscript. Therefore, including Fig 2, all SEM
images are revised by improving sample preparation.

Generally, to observe the SEM image, the simple solvent evaporation method is used for
drying sample on silicon substrate, which resulted in severe aggregation (just like the SEM
image we obtained in original Fig 2). However, the conventional dilution operation is not
suitable for our current system, because when a good solvent is added, the assembly will

change. To solve the problem, we use PSS (poly(sodium-p-styrenesulfonate)) to modify
silicon substrate. Once the modified substrate is immersed in the fresh LSM solution, the
sufficient $-\text{SO}_3^-$ of PSS will form $-\text{SO}_3\text{H}$, which is beneficial to form H-bond between PSS
and LSM (LSP and SSP). Moreover, the immersing time of substrate should be as short as
possible to avoid too many LSM dip-coated on the modified substrate. However, even with
such a sophisticated treatment, it should be pointed out that for nanoscale structures,
aggregation cannot be completely avoided. Thus, in order to observe clearly and accurately,
we etched part of the LDH to expose the LSM before aggregation, that is, the LSM is still on
the etched fragments of LDH.

As seen in revised Fig. 2a–c and their insets, the hexagonal sheet structure of etched
SG7-LDH₂₀, SG7-LDH₅₀ and SG7-LDH₁₀₀ are difficult to observe because the small size of
LDH leads to uncontrollable rapid dissolution. Even so, it can be concluded that as the
increase of LDH size, the size of LSM also increases because the LSM is still on the etched
fragments of LDH. Besides, the etched piece of SG7-LDH₃₀₀₀ remained the hexagonal sheet
structure due to the slower dissolution, where the size of 200 nm attributed to LSM₃₀₀₀ can
be clearly seen (Fig. 2d).

*5.2 Even if the sizes of SMs were controlled by the size of LDHs, why the size of metastable*
*LSPs could be controlled? In theory, the size of supramolecular polymers were controlled by*
*the ratio of the monomers and nuclei (or seeds). Given that all the added monomers were*
*used for the growth of the seeds (or short supramolecular polymers), the sizes can be*
*controlled. To the best of my knowledge, none of the systems could control the size of the*

*assemblies by changing the size of monomers. How could the authors explain?*

Response: Thanks for your question. This is a very enlightening question. It makes us realize
that the naming of the products in each assembly process is not clear enough. Especially the
step 2, after removal of LDH, the released ordered molecule array forms LSM, acting as
pre-assembled seeds, leading to the formation of thermodynamically favorable LSP. Thus, we
renamed original supramonomer (SM) as living supramonomer (LSM) (Fig. 1). Since the
high active of LSM as a pre-assembled seed can induce LSP to start to grow, the size of LSM
can influences the starting size of metastable LSP rather than the final size of stable LSP. This
is confirmed by the degree of polymerization in Fig. 2 and sizes in SEM images
(Supplementary Fig. 25) of fresh metastable LSP. At the same time, this phenomenon is
similar to the size control theory of conventional supramolecular polymers. In the step 2 of
our system, the LSM is pre-assembled seed composed of small ordered molecule array, which
are released from the interlayer of LDH, rather than a single monomer (Fig. 2). Therefore,
under the formation conditions of the supramolecular polymer, the change in the size of LSM
endowed by the controllable size of LDH means that the ratio of the nucleus (or seed) to
monomer has changed, which in turn influences the size of the final supramolecular polymer.
To conclude, the sizes of LSMs were controlled by the size of LDHs, which gives the
supramolecular polymer the size-controllable property. That is, in our system, the size of LSP
is controlled by the size of pre-assembled seeds (LSM with ordered arrangement), rather than
the size of the monomers.

*5.3 Furthermore, there is no scientific evidence that SMs are further assembled into*

*metastable LSP or SMs are once disassembled into free MSS and re-assembled into*
*metastable LSP.*

Response: Actually, LSMs prefer to further assemble into metastable LSP. Firstly, the
theoretical calculations can confirm that preference of further assembly of LSMs. The
binding energy for the further assembly of LSM is $-52.64 \text{ kJ mol}^{-1}$ per SG7 molecule,
indicating that it is thermodynamically favorable (Supplementary Fig. 11h). Meanwhile, the
dissociation energy of LSM disassembly is calculated to be $52.64 \cdot (n-1) \text{ kJ mol}^{-1}$ by
subtracting every single SG7 from LSM (n represents the number of SG7 molecule in LSM).
Therefore, in the sight of the energy, LSMs prefer to further assemble into metastable LSP. It
should also be pointed out that there are indeed reversible assembly processes in our system,
but they appear between LSM and LSP, rather than between LSM and monomer.

Secondly, LSMs would not disassemble into free SG7 and then re-assemble into
metastable LSP. This can be confirmed by the following experiment. If LSMs disassemble to
free SG7, the following LSP will fail to form. Because LSMs disassemble into free SG7 in
good solvent, the H-bond net among LSM is replaced by the stronger H-bond between
solvent CH_3OH and SG7 (Supplementary Fig. 23). Moreover, this LSM disassembly is not
reversible to form LSP, as confirmed by the absence of rectangular morphology in SEM
images (Supplementary Fig. 23).

To sum up, the LSMs prefer to further assemble into metastable LSP, rather than LSMs
once disassembled into free MSS and re-assembled into metastable LSP.

**Figure 2. Tuning the size and activity of LSP by LDH confinement.** (a–d) SEM images of
 SG7 LSM from different size SG7-LDHs: (a) 20 nm, (b) 50 nm, (c) 100 nm and (d) 3 μm
 (insets are corresponding magnifications and the LSM sizes in the insets are marked with a
 yellow line). (e–h) SEM images of SG7 LSP from the corresponding LSM in a–d,
 respectively. (i–l) Debye plot of metastable LSP made of corresponding LSM in a–d,
 respectively.

**Supplementary Figure 25.** SEM images of fresh metastable (a) LSP₂₀, (b) LSP₅₀, (c) LSP₁₀₀

and (d) LSP₃₀₀₀.

**Supplementary Figure 11.** (a) Schematic illustration for the definition of orientation angle, θ .

(b) Energy diagram of SG7 LSM with different orientation. Optimized geometries of (c)

SG7-LDH, (d) inactive SG7 aggregates ($\theta = 35^\circ$), (e) SG7 LSM+SG7 monomer ($\theta = 35^\circ$) and

(f) SG7 LSM+SG7 monomer ($\theta = 9^\circ$), together with the highlighted binding areas in e and f:
(g) $\theta = 35^\circ$ and (h) $\theta = 9^\circ$. The color of each element is labeled in c.

**Supplementary Figure 23.** SEM images of aged SG7 LSM₂₀ for 20 min after removal of
LDHs in CH₃OH/TFA (5:3 v/v).

**p. 9 Line 20:** It should be pointed out that for nanoscale structures, aggregation cannot
be completely avoided.^{50,51} Thus, in order to observe accurately, we etched part of the LDH to
expose the LSM before aggregation, that is, the LSM is still on the etched fragments of LDH.
As confirmed by SEM images (Fig. 2a–d and their insets), the sizes of metastable LSMs
gradually increase with the raising of LDH sizes from 20 nm to 3 μm (as-prepared SMs from
different size LDHs are named as LSM₂₀, LSM₅₀, LSM₁₀₀ and LSM₃₀₀₀, respectively). Note
that, metastable LSMs are easily destroyed by new H-bond between CH₃OH and SG7
molecule, preventing continued growth confirmed by SEM (Supplementary Fig. 23). Thus,
by increasing the amount of poor solvent TFA, combined with the crystallization driving
principle,^{52–54} LSMs can continue to grow into LSP (as-prepared LSPs made from different
size LDHs are named as LSP₂₀, LSP₅₀, LSP₁₀₀ and LSP₃₀₀₀, respectively). The highly ordered
structure of LSP₂₀–LSP₃₀₀₀ can be confirmed by high FL anisotropic values ($r > 0.45$,
Supplementary Fig. 24). With increase size of LDH from 50 nm to 3 μm, LSP₅₀–LSP₃₀₀₀

display successively increased area from $0.32 \mu\text{m}^2$ to $4.0 \mu\text{m}^2$ (Fig. 2f–h). Interestingly,
although LSP₂₀ does not comply with the above rules, it shows surprising results of
unexpected extension with an area of $\sim 1.50 \mu\text{m}^2$ (Fig. 2e), testifying activity of LSP₂₀ is the
strongest, which is discussed in the following cycle experiments. Therefore, the realization of
living polymerization for SG7 LSM is attributed to synergy between diverse activities and
starting LSM sizes endowed by LDHs. These phenomena demonstrate the dimensions of
ordered metastable states (LSMs and LSPs) can be regulated by changing the size of LDH
confined space.

*6. Line 168 in p9; the authors described that “the activity of metastable LSP₂₀ not only*
*recovered, but also increased exponentially by gently shaking, confirmed by FL intensity*
*increases to four times. It is reasonable that the activity of metastable LSP₂₀ was recovered by*
*shaking. In this case, the aggregated LSP₂₀ are supposed to be disassembled into free LSP₂₀*
*and, as a result, could initiate the seeded polymerization. On the other hand, the FL intensity*
*change implies more than that. Basically, FL intensity changes due to the changes of the*
*packing of SG7 accompanied by supramolecular polymerization. Generally, FL intensity does*
*not changes so much even if supramolecular polymers, like SMs, were aggregated, because*
*their packings of monomers in supramolecular polymers were not changed. Hence, the*
*observed increment in FL intensity may suggest that the aggregation of LSP₂₀s were localized*
*at the bottom of the optical cell and their FL intensity was underestimated. In that case,*
*Figure 3D would be not be trustable. Hence, authors should monitor FL intensity under the*
*gentle stirring and reconsider this part.*

Response: Thank you very much for your suggestion. They are greatly helpful for precisely
monitoring the FL intensity of metastable LSP under magnetic stirring. All of the revisions
are highlighted in red.

Using the newly designed device (Supplementary Fig. 28), we retested the fluorescence
properties for the unsonicated LSP₂₀, LSP₅₀, LSP₁₀₀, LSP₃₀₀₀ and sonicated LSP₂₀. In this case,
the phenomenon of LSP aggregate depositing to the bottom of the optical cell will be ruled
out.

All unsonicated LSP₂₀–LSP₃₀₀₀ (Fig. 3d and Supplementary Fig. 29) show the same
changed trend, where FL intensity first rises and then falls. Firstly, FL intensity of
unsonicated metastable LSP₂₀ (Fig. 3d) increases in the initial stage, attributing to the
formation of metastable LSP₂₀ ($\lambda_{em} = 535$ nm) by crystallization-driven assembly of LSM₂₀
($\lambda_{em} = 460$ nm), which is confirmed by their different wavelength in FL emission spectra
(Supplementary Fig. 8–9). Secondly, the decrease in FL intensity of unsonicated metastable
LSP₂₀ (Fig. 3d) can be attributed to the elongation of metastable LSP₂₀, which is confirmed
by stable LSP with lower FL intensity (Supplementary Fig. 9) and larger area in SEM images
(Supplementary Fig. 12).

In the initial stage, the FL intensity LSP₂₀ increased with a highest rate (117.8 a.u./min)
compared with LSP₅₀ (13.56 a.u./min), LSP₁₀₀ (16.73 a.u./min) and LSP₃₀₀₀ (29.71 a.u./min)
(Fig. 3d and Supplementary Fig. 29). This phenomenon indicates that LSM₂₀ is most active to
assembly to metastable LSP₂₀. Meanwhile, LSP₂₀ shows the highest FL intensity at 0 min
(Supplementary Fig. 3d and Supplementary Fig. 29), indicating the smallest starting size of
metastable LSP₂₀ due to the smallest LDH₂₀ and LSM₂₀. In addition, during the elongation of

metastable LSP₃₀₀₀ with decreased FL intensity, there was a period when the decline rate was
 very low. Considering the fact that LDH₃₀₀₀ shows a slower dissolution in TFA from Fig. 2,
 LSM₃₀₀₀ is continuously and slowly released from the LDH₃₀₀₀ and acts as new nuclei (seed)
 to form LSP₃₀₀₀ again, resulting in the change of decline rate. Compared with unsonicated
 LSP₂₀, the FL intensity of freshly sonicated LSP₂₀ only shows decline with low rate,
 attributing to the elongation (Supplementary Fig. 31). This result is attributed to that the
 assembly of LSM is quickly finished during the ultrasound and only elongation can be
 observed during FITS test. Meanwhile, the FL intensity of freshly sonicated LSP₂₀ is much
 weaker than that of unsonicated LSP₂₀, indicating that sonicated LSP₂₀ has larger size. As
 confirmed in SEM images, sonicated LSP₂₀ can elongate to $\sim 1.50 \mu\text{m}^2$ while unsonicated
 LSP₂₀ shows uneven and smaller size ($0.01\text{--}0.20 \mu\text{m}^2$) due to the suppressed elongation
 activity. (Fig. 2 and Supplementary Fig. 30).

 **Supplementary Figure 28.** The schematic presentation of experimental set-up of FITS under
 magnetic stirring.

**Supplementary Figure 29.** FITS of metastable LSP₅₀, LSP₁₀₀ and LSP₃₀₀₀ without
 ultrasound.

**Supplementary Figure 8.** (a) FL emission spectra and (b) corresponding normalized FL
 spectra of metastable LSM₂₀–LSM₃₀₀₀ after removal of LDH layers.

**Supplementary Figure 9.** FL spectra of LSP₂₀–LSP₃₀₀₀: (a) freshly sonicated metastable
 LSP and (b) stable LSP kept for 12 h after ultrasound.

**Figure 3. Kinetics behavior of LSP₂₀.** (a–b) CLSM images of LSP₂₀: (a) metastable and (b)

kept for 12 h from a. (c) Average size of LSP₂₀ kept for 12 h by different ultrasound time. (d)

FITS of metastable LSP₂₀ without ultrasound. (e–g) The size of SSP as a function of (e) the

concentrations of SG7_{agg}, (f) the volume ratio of added SG7_{agg} to the LSP_{seed}, (g) the amounts

of SG7-LDH. (h) Log–log plot of the rate of increased absorbance at 460 nm as a function of

the LSP_{seed} concentration. Error bars are calculated using the standard error formalism for the

data of three replicate experiments. (i) Time scan of absorbance of SSP₂₀, where LSP_{seed} are

metastable LSP₂₀ with different concentrations (optical path = 1 mm). (j) Energy landscapes

of products during step 1–5.

**Supplementary Figure 12.** SEM images of LSP₂₀ with ultrasound kept for different time: **(a)**

0 h, **(b)** 1 h, **(c)** 3 h and **(d)** 12 h.

**Supplementary Figure 31.** FITS of freshly sonicated LSP₂₀ under magnetic stirring at 540

1710 nm.

**Figure 2. Tuning the size and activity of LSP by LDH confinement. (a–d)** SEM images of

SG7 LSM from different size SG7-LDHs: (a) 20 nm, (b) 50 nm, (c) 100 nm and (d) 3 μm

(insets are corresponding magnifications and the LSM sizes in the insets are marked with a

yellow line). (e–h) SEM images of SG7 LSP from the corresponding LSM in a–d,

respectively. (i–l) Debye plot of metastable LSP made of corresponding LSM in a–d,

respectively.

**Supplementary Figure 30.** SEM image of LSP₂₀ without ultrasound kept for 12 h.

**p. 12 Line 9:** The fluorescence intensity time scan (FITS) was monitored by using
specially designed fluorescent device with magnetic stirring (Supplementary Fig. 28). All
unsonicated LSP₂₀–LSP₃₀₀₀ (Fig. 3d and Supplementary Fig. 29) show the same changed
trend, where FL intensity first rises and then falls. Firstly, FL intensity of unsonicated
metastable LSP₂₀ (Fig. 3d) increases in the initial stage, attributing to the formation of
metastable LSP₂₀ ($\lambda_{em} = 535$ nm, Supplementary Fig. 9) by crystallization-driven assembly of
LSM₂₀ ($\lambda_{em} = 460$ nm, Supplementary Fig. 8). Secondly, the decreased FL intensity and
increased area of sonicated metastable LSP₂₀ (Supplementary Fig. 9, 12 and 31) compared
with unsonicated metastable LSP₂₀ (Fig. 4d, Supplementary Fig. 30) can be attributed to the
elongation of metastable LSP₂₀.

In the initial stage, the FL intensity LSP₂₀ increased with a highest rate (117.8 a.u./min)
compared with LSP₅₀ (13.56 a.u./min), LSP₁₀₀ (16.73 a.u./min) and LSP₃₀₀₀ (29.71 a.u./min)
(Supplementary Fig. 29). This phenomenon indicates that LSM₂₀ is most active to assembly
to metastable LSP₂₀. Meanwhile, LSP₂₀ shows the highest FL intensity at 0 min (Fig. 4d and
Supplementary Fig. 29), indicating the smallest starting size of metastable LSP₂₀ due to the

smallest LDH₂₀ and LSM₂₀. In addition, during the elongation of metastable LSP₃₀₀₀ with
decreased FL intensity, there was a period when the decline rate was very low. Considering
the fact that LDH₃₀₀₀ shows a slower dissolution in TFA (Fig. 2), LSM₃₀₀₀ is continuously and
slowly released from the LDH₃₀₀₀ and acts as new nuclei (seed) to form LSP₃₀₀₀ again,
resulting in the change of decline rate. Compared with unsonicated LSP₂₀, the FL intensity of
freshly sonicated LSP₂₀ only shows decline with low rate, attributing to the elongation (Fig.
4d and Supplementary Fig. 31). This result is attributed to that the assembly of LSM is
quickly finished during the ultrasound and only elongation can be observed during FITS test.
Meanwhile, the FL intensity of freshly sonicated LSP₂₀ is much weaker than that of
unsonicated LSP₂₀ (Fig. 4d and Supplementary Fig. 12 and 31), indicating that sonicated
LSP₂₀ has larger size. As confirmed in SEM images, sonicated LSP₂₀ can elongate to ~1.50
1748 μm^2 while unsonicated LSP₂₀ shows uneven and smaller size (0.01–0.20 μm^2) due to the
1749 suppressed elongation activity. (Fig. 2 and Supplementary Fig. 30).

*Some other comments on the experiments:*

*7. I can understand that it is not easy to discuss the size of the assemblies in particular if they*
*were aggregated. With the current data set such as figure S19 and Figure 3a,b, it is difficult*
*to provide the trustable data. If the authors would like to discuss the sizes, they should find a*
*condition where the assemblies are not aggregated and characterize their individual sizes,*
*otherwise it's too risky.*

Response: Thank you very much for your suggestion. Indeed, it is difficult to discuss the
size of assemblies if they are aggregated. In fact, in our system, the obtained LSP is easy to

aggregate because of their sufficient functional groups, leading to inaccurate assessment of
the LSP size through the CLSM. Thus, the CLSM was only used to observe the appearance
and movement of LSP in the solution state.

In order to monitor the size of all samples (LSM, LSP, and SSP) more accurately, the SEM
was used. At the same time, before preparing the samples for SEM images, we use PSS
(poly(sodium-p-styrenesulfonate)) to modify silicon substrate. Once the modified substrate is
immersed into the fresh sample (LSM, LSP or SSP) solution, the sufficient $-\text{SO}_3^-$ of PSS will
form $-\text{SO}_3\text{H}$, which is beneficial to form H-bond between PSS and samples. Moreover, the
immersing time of substrate should be as short as possible to avoid too many samples being
dip-coated on the modified substrate. All SEM images have been revised. In revised
manuscript, all SEM images in Fig. 2 and Fig. 4 are revised to clearly show the sizes of LSM,
LSP and SSP. In supporting information (Supplementary Fig. 12, 15, 25, 27, 32–34, 69–70
and 72), SEM images of many samples are revised by modified substrate, such as fresh
metastable LSP_{20} – LSP_{3000} , LSP_{20} without ultrasound kept for 12 h, LSP_{20} with ultrasound
kept for different time, LSP_{20} kept for 12 h with different ultrasound time, fresh
SSP_{50} – SSP_{3000} , fresh SSP_{20} (controlled by different SG7_{agg} concentration, SG7_{agg} ratio and
LSP_{seed} concentration), various LSP and corresponding SSP (made of SG7, ABSA, DABSA
and CR) in universality part.

**Figure 2. Tuning the size and activity of LSP by LDH confinement.** (a–d) SEM images of

SG7 SM from different size SG7-LDHs: (a) 20 nm, (b) 50 nm, (c) 100 nm and (d) 3 μm

(insets are corresponding magnifications and the LSM sizes in the insets are marked with a

yellow line). (e–h) SEM images of SG7 LSP from the corresponding LSM in a–d,

respectively. (i–l) Debye plot of metastable LSP made of corresponding LSM in a–d,

respectively.

**Figure 4. Living supramolecular polymerization.** (a) The size of SSP₂₀ as a function of
 cycle number x ($x = 1, 2$ and 3). Error bars were calculated using the standard error formalism
 for the data of three repeated experiments (insets: corresponding SEM images of SSP₂₀
 obtained in Cycle 1–3, respectively). (b–c) FL spectra of SSP₂₀ obtained in Cycle 1–3.

**Supplementary Figure 12.** SEM images of LSP₂₀ with ultrasound kept for different time: (a)
 0 h, (b) 1 h, (c) 3 h and (d) 12 h.

**Supplementary Figure 25.** SEM images of fresh metastable (a) LSP₂₀, (b) LSP₅₀, (c) LSP₁₀₀

and (d) LSP₃₀₀₀.

**Supplementary Figure 27.** SEM images of LSP₂₀ kept for 12 h with different ultrasound

time: (a) 50 min, (b) 60 min and (c) 120 min.

**Supplementary Figure 32.** SEM images of fresh SSP₂₀ made from LSP_{seed} and SG7_{agg} with

different concentrations: (a) 0.400, (b) 2.00 and (c) 3.00 g L⁻¹.

**Supplementary Figure 33.** SEM images of fresh SSP₂₀ by different volume ratio of

SG7_{agg}/LSP_{seed}: **(a)** 1:1, **(b)** 2:1 and **(c)** 4:1.

**Supplementary Figure 34.** SEM images of fresh SSP₂₀ prepared with metastable LSP₂₀ with

different concentrations: **(a)** 0.250, **(b)** 2.50 and **(c)** 5.00 g L⁻¹.

**Supplementary Figure 69.** SEM images of **(a)** fresh metastable LSP made of ABSA,

corresponding fresh SSPs in **(b)** Cycle 1 and **(c)** Cycle 2, and **(d)** contrast sample: physically

mixing pure ABSA and Cl-LDH in CH₃OH/TFA (1:3 v/v) with the same dosage as LSP.

**Supplementary Figure 70.** SEM images of (a) fresh metastable LSP made of DABSA,

corresponding fresh SSPs in (b) Cycle 1 and (c) Cycle 2, and (d) contrast sample: physically

mixing pure DABSA and Cl-LDH in CH₃OH/TFA (1:3 v/v) with the same dosage as LSP.

**Supplementary Figure 72.** SEM images of (a) fresh metastable LSP made of CR,

corresponding fresh SSPs in (b) Cycle 1 and (c) Cycle 2, and (d) contrast sample: physically

mixing pure CR and Cl-LDH in CH₃OH/TFA (1:3 v/v) with the same dosage as LSP.

**p. 20 Line 13:** The real-time changes in optical signals are powerful method to monitor
the formation of metastable states and *dynamic* elongation process due to the structure-depended
luminescence behavior of SG7 molecule. Taking the LSM₂₀ without ultrasound as an
example, the formed LSP₂₀ move vigorously before 10 min (Supplementary vedio 1), but
severe stack can be observed in confocal laser scanning microscope (*CLSM*) image as the time
increase (Supplementary Fig. 26). To address this problem, mechanical agitation is often used,
which creates smaller structures with fully exposed active sites as initiators for living
polymerization. When LSM₂₀ is sonicated for 1 h, the as-obtained product (metastable LSP₂₀)
shows uniform distribution of area (Fig. 3a). After being kept for 12 h, kinetics capture
extends the area from 0.03 μm^2 to 1.50 μm^2 (Supplementary Fig. 12), while retaining uniform
characteristics (Fig. 3b). Importantly, a competitive growth of LSP₂₀ at anisotropic active
sites can be observed. The average width increased by 200%, while the length only increased
by 48% by prolonging ultrasound time (Fig. 3c and Supplementary Fig. 27).The real-time
changes in optical signals are powerful method to monitor the formation of metastable states
and *dynamic* elongation process due to the structure-depended luminescence behavior of SG7
molecule. Taking the LSM₂₀ without ultrasound as an example, the formed LSP₂₀ move
vigorously before 10 min (Supplementary vedio 1), but severe stack can be observed in
confocal laser scanning microscope (*CLSM*) image as the time increase (Supplementary Fig. 26).
To address this problem, mechanical agitation is often used, which creates smaller structures
with fully exposed active sites as initiators for living polymerization. When LSM₂₀ is
sonicated for 1 h, the as-obtained product (metastable LSP₂₀) shows uniform distribution of
area (Fig. 3a). After being kept for 12 h, kinetics capture extends the area from 0.03 μm^2 to

1.50 μm^2 (Supplementary Fig. 12), while retaining uniform characteristics (Fig. 3b).
Importantly, a competitive growth of LSP₂₀ at anisotropic active sites can be observed. The
average width increased by 200%, while the length only increased by 48% by prolonging
ultrasound time (Fig. 3c and Supplementary Fig. 27).

REVIEWER COMMENTS

Reviewer #1 (Remarks to the Author):

Comments on NCOMMS-20-26877 (Revised)

Authors: Wenying Shi and Chao Lu et al

The authors have made a great effort to enhance the quality of the paper and the revised manuscript is certainly much improved. The results are now much more convincing and lend support for the assertions.

Further comments:

- 1) It is important that someone with English as a first language critically reads the MS before publication.
The Title, in particular, reads very poorly in English (it is more accurate now). "Simple-structured" is not good wording.
Maybe something like:
Enabling Living Supramolecular Polymerization of Simple Monomers through Confinement
- 2) Examples of the "simple monomers" should be stated in the abstract.

Reviewer #2 (Remarks to the Author):

The revised manuscript has addressed the reviewer's comments satisfactorily. However still the reference section has not been updated, for example recent review article (Chem. Commun. 2020, 56, 6757–6769) in this topic is missing, the title (particularly the term "Simple-structured") may be modified. Once these minor issues are addressed, the manuscript may be accepted for publication.

Reviewer #3 (Remarks to the Author):

The authors sincerely tackled reviewers' comments including a lot of additional data. I highly appreciate it. However, as all the reviewers commented in their first comments, the writing style of the manuscript was very unclear and it is still difficult to understand the revised manuscript (though the revised one became better than the first one). Unfortunately, there are still several fundamental errors, unclear and misleading descriptions, strongly indicating that the authors do not understand controlled supramolecular polymerization correctly. Hence, I still cannot recommend this manuscript publication for Nature Communications. Below I have described two particularly important points together with minor comments on the authors' responses.

(1) Proposed mechanism for the controlled supramolecular polymerization

In my previous comments, I asked why the size of metastable LSPs (not LSPs!!) could be controlled. However, the authors as the following;

=====

Since the high active of LSM as a pre-assembled seed can induce LSP to start to grow, the size of LSM can influence the starting size of metastable LSP rather than the final size of stable LSP. This is confirmed by the degree of polymerization in Fig. 2 and sizes in SEM images (Supplementary Fig. 25) of fresh metastable LSP. At the same time, this phenomenon is similar to the size control

theory of conventional supramolecular polymers. In the step 2 of our system, the LSM is pre-assembled seed composed of small ordered molecule array, which are released from the interlayer of LDH, rather than a single monomer (Fig. 2). Therefore, under the formation conditions of the supramolecular polymer, the change in the size of LSM endowed by the controllable size of LDH means that the ratio of the nucleus (or seed) to monomer has changed, which in turn influences the size of the final supramolecular polymer.

=====

Here, the authors explained the controlled polymerization for LSP rather than metastable LSP. In the former part, they only said that this phenomenon is similar to the size control theory of conventional supramolecular polymers. However, I don't think this result is conventional one. Once again, I explain the reason why the authors' explanation is inappropriate. I can understand that LDH, as a template, can control of the size of LSM (pre-assembled seed). On the other hand, the authors explained that this LSM is further assembled to form metastable LSP, meaning that LSM worked as a "monomer" and metastable LSP is formed as a "supramolecular polymer". In this process, there is no reason that the size of the resultant supramolecular polymer (metastable LSP) is controlled by the size of LSM. In conventional theory, the size of metastable LSP is decided by concentration. In the case of the conventional supramolecular polymers, polymerization happens one-dimensionally. As a result, only the length increases as the degree of polymerization increases. Hence, if the authors have only observed that the width of metastable LSP was uniquely decided by the size of LDH (and this width must be same with the size of LDH), I could understand. However, apparently, the width of metastable LSP was much larger than those of LDH. Hence, I cannot imagine that the size of metastable LDH is really controlled.

As clearly mentioned in my previous comments, it is not easy to properly evaluate the size of assemblies. Hence, we often evaluate about 100 of supramolecular polymers by AFM together with other data such as SLS. Of course, in the previous papers for controlled living supramolecular polymerization, the mechanism is very clear and well substantiated. I accepted the explanation of the step 4, which is conventional one. However, it is very puzzling to trust the fact that the sizes of metastable LSPs were well controlled by the size of LDH. I strongly recommend the authors to confirm the reproducibility of SLS and SEM characterization by other members in their labs. And, the authors must evaluate more than 50 of metastable LPS by SEM and calculate the polydispersity.

Because the size of final LSP is controlled by the ratio of seeds and monomers, as in the case of previously reported systems, I think there is no necessity to include the discussion on the size effect of LDH, which is not trustable at present. I recommend the authors to remove these discussion, which make the story more simple and understandable.

(2) Energy Diagram of Figure 3j

This schematic illustration well explains that the authors do not understand the basics of supramolecular assemblies. I request the authors to well consider the first step depicted here, showing that molecular dispersed SG7 (mentioned as SG7 solution) is the most energetically stable state. If the molecularly dispersed state of SG7 is the most stable, what is the driving force to be incorporated into LDH? The authors drew arrows that the release of SG7 from LDH is irreversible because it cannot exceeds the energy hill. However, in this diagram, in order for SG7 to reach the intercalated state in LDH, SG7 must exceed more higher energy hill. Hence, this diagram is apparently wrong. The authors have to study the previous papers such as Nat. Chem. 9, 493, 2017. In Fig.2, there is an energy diagram to explain the mechanism. When we talk about the kinetic assembly and thermodynamic assembly, the energy level of the latter one is lower than the former one. However, in either case, both energies levels are lower than that of their monomeric state. Furthermore, it is possible to compare the energies under the same condition (concentration, temperature, solvents and so on). The authors compared the energies levels in different states. However, in their experiments, solvents are changed. Hence, it is scientifically wrong to compare all the energy states like this.

(3) Definition of supramolecular polymerization

In my previous comment, I pointed out this is just crystallization rather than supramolecular polymerization. However, I could not understand what the authors wanted to insist. In my opinion, supramolecular polymerization is either one-dimensional or two-dimensional assembling processes. If the authors wanted to claim that their work can be regarded as supramolecular polymerization rather than crystallization, the authors simply explain the differences of these two

in their definitions with references. The unrelated discussion together with data in the answer made me strongly hesitate to continue to review this manuscript.

(4) Confinement effect

In my previous comment, I mentioned that the confinement effect never induced supramolecular polymerization. Of course, the confinement effect itself is necessary for the observed results. I don't deny it. Simply, the original title sounded mis-leading. However, here, the authors' answer something unrelated to my comment.

(5) AIMD

MD simulation becomes an important tool to discuss how molecules behave. However, it is still not conclusive one. In particular, the molecular structures of LSM and metastable LSP are not yet understood by experiments. It is very risky to discuss the energy levels with these calculations.

Responses to Reviewer 1

Overview: The authors have made a great effort to enhance the quality of the paper and the revised manuscript is certainly much improved. The results are now much more convincing and lend support for the assertions.

Response:

Thank you for your recognition of our work. Your previous suggestions have been great helpful to improve the academic rigorousness of our manuscript.

Further comments:

1. It is important that someone with English as a first language critically reads the MS before publication. The Title, in particular, reads very poorly in English (it is more accurate now).

“Simple-structured” is not good wording. Maybe something like: Enabling Living Supramolecular Polymerization of Simple Monomers through Confinement.

Response:

Thank you for your comments. As you suggested, we invited a specialist in supramolecular polymer with English as a first language to polish our article and give some professional advice.

The title has been revised to: “Oriented Arrangement of Simple Monomers Enabled by Confinement: Towards Living Supramolecular Polymerization.”

2. Examples of the “simple monomers” should be stated in the abstract.

Response:

Thank you for your patience. As you suggested, corresponding detailed examples of “simple monomers” have been supplemented into our abstract and highlighted in red.

p. 2 Line 16 in abstrat:

Here, with the benefit of the confinement from the layered double hydroxide (LDH) nanomaterial, various simple monomers (such as benzene, naphthalene and pyrene derivatives) successfully form living supramolecular polymer (LSP) with length control and narrow dispersity.

Responses to Reviewer 2

Overview: The revised manuscript has addressed the reviewer's comments satisfactorily. However still the reference section has not been updated, for example recent review article (Chem. Commun. 2020, 56, 6757–6769) in this topic is missing, the title (particularly the term "Simple-structured") may be modified. Once these minor issues are addressed, the manuscript may be accepted for publication.

Response:

Thank you for your comments. We appreciate your positive comments for our work. As you suggested, new reference and modified title in the revised manuscript are highlighted in red. In addition, the term “Simple-structured” has been modified to “Simple monomers”.

The title has been revised to: “Oriented Arrangement of Simple Monomers Enabled by Confinement: Towards Living Supramolecular Polymerization.”

p. 24 Line 435 in manuscript:

[9]Ghosh, G., Dey, P. & Ghosh, S. Controlled supramolecular polymerization of π -systems. Chem. Commun. 56, 6757–6769 (2020).

Responses to Reviewer 3

Overview: The authors sincerely tackled reviewers' comments including a lot of additional data. I highly appreciate it. However, as all the reviewers commented in their first comments, the writing style of the manuscript was very unclear and it is still difficult to understand the revised manuscript (though the revised one became better than the first one). Unfortunately, there are still several fundamental errors, unclear and misleading descriptions, strongly indicating that the authors do not understand controlled supramolecular polymerization correctly. Hence, I still cannot recommend this manuscript publication for Nature Communications. Below I have described two particularly important points together with minor comments on the authors' responses.

Response:

We are particularly thankful for your previous suggestion, which helped us to acquire a deeper understanding about the controlled supramolecular polymerization. Therefore, after receiving this suggestion, we are very willing to continue to make revisions for improving the quality of our manuscript. The changes are highlighted in red.

Further comments:

1. *Proposed mechanism for the controlled supramolecular polymerization.*

In my previous comments, I asked why the size of metastable LSPs (not LSPs!!) could be controlled. However, the authors as the following;

=====

Since the high active of LSM as a pre-assembled seed can induce LSP to start to grow, the

size of LSM can influences the starting size of metastable LSP rather than the final size of stable LSP. This is confirmed by the degree of polymerization in Fig. 2 and sizes in SEM images (Supplementary Fig. 25) of fresh metastable LSP. At the same time, this phenomenon is similar to the size control theory of conventional supramolecular polymers. In the step 2 of our system, the LSM is pre-assembled seed composed of small ordered molecule array, which are released from the interlayer of LDH, rather than a single monomer (Fig. 2). Therefore, under the formation conditions of the supramolecular polymer, the change in the size of LSM endowed by the controllable size of LDH means that the ratio of the nucleus (or seed) to monomer has changed, which in turn influences the size of the final supramolecular polymer.

=====

Here, the authors explained the controlled polymerization for LSP rather than metastable LSP. In the former part, they only said that this phenomenon is similar to the size control theory of conventional supramolecular polymers. However, I don't think this result is conventional one. Once again, I explain the reason why the authors' explanation is inappropriate. I can understand that LDH, as a template, can control of the size of LSM (pre-assembled seed). On the other hand, the authors explained that this LSM is further assembled to form metastable LSP, meaning that LSM worked as a "monomer" and metastable LSP is formed as a "supramolecular polymer". In this process, there is no reason that the size of the resultant supramolecular polymer (metastable LSP) is controlled by the size of LSM. In conventional theory, the size of metastable LSP is decided by concentration. In the case of the conventional supramolecular polymers, polymerization happens one-dimensionally. As a result, only the length increases as the degree of polymerization

increases. Hence, if the authors have only observed that the width of metastable LSP was uniquely decided by the size of LDH (and this width must be same with the size of LDH), I could understand. However, apparently, the width of metastable LSP was much larger than those of LDH. Hence, I cannot imagine that the size of metastable LDH is really controlled.

As clearly mentioned in my previous comments, it is not easy to properly evaluate the size of assemblies. Hence, we often evaluate about 100 of supramolecular polymers by AFM together with other data such as SLS. Of course, in the previous papers for controlled living supramolecular polymerization, the mechanism is very clear and well substantiated. I accepted the explanation of the step 4, which is conventional one. However, it is very puzzling to trust the fact that the sizes of metastable LSPs were well controlled by the size of LDH. I strongly recommend the authors to confirm the reproducibility of SLS and SEM characterization by other members in their labs. And, the authors must evaluate more than 50 of metastable LPS by SEM and calculate the polydispersity.

Because the size of final LSP is controlled by the ratio of seeds and monomers, as in the case of previously reported systems, I think there is no necessity to include the discussion on the size effect of LDH, which is not trustable at present. I recommend the authors to remove these discussion, which make the story more simple and understandable.

Response:

We regret that we did not give you a satisfactory reply in the last revision due to our misunderstanding. Thank you for your patience to describe your comment again. According to your questions and comments, we have made explanation one by one as follows.

1.1 The controlled metastable LSM and metastable LSP

On this question, first, we prove the existence of metastable states. And then we proved their controllability.

1.1.1 The existence of metastable states

The metastable states, which is kinetic and non-equilibrium structures, are prone to spontaneously convert into the equilibrium state over time or by application of some kind of stimulus that aids in overcoming the activation barrier to ease into the global minimum (*Angew. Chem. Int. Ed* **58**, 16730–16740 (2019)).

1) Metastable LSM

The SG7-LDH (20 nm, 4 mg) is dissolved in 320 μL mixture solvent $\text{CH}_3\text{OH}/\text{TFA}$ (5:3 v/v) under stirring for 0–10 min, which is defined as the metastable LSM_{20} . Under this condition, LDH can be completely etched away, confirmed by the disappearance of LDH diffraction peaks in XRD patterns (Supplementary Fig. 3 and 25).

LSM_{20} is a metastable state, which can be confirmed by FL spectra. As seen in Supplementary Fig. 9, FL spectra of metastable LSM_{20} ($\lambda_{\text{em}} = 453 \text{ nm}$) showed no obvious change in wavelength and intensity in $\text{CH}_3\text{OH}/\text{TFA}$ (5:3 v/v) in the period of 0–10 min. This state can be maintained for 10 min. After 10 min, the metastable LSM_{20} began a relaxation to its energetically favored state confirmed by the decreased FL intensity at 453 nm.

2) Metastable LSP

The SG7-LDH (20 nm, 4 mg) is dissolved in 800 μL mixture solvent $\text{CH}_3\text{OH}/\text{TFA}$ (1:3 v/v), followed with ultrasound, which is defined as the metastable LSP_{20} . As seen in the FITS of metastable LSP_{20} (Supplementary Fig. 12), the FL intensity remained constant in the first 5 min, indicating metastable LSP_{20} was in the energy well for a while. After 5 min, metastable

LSP₂₀ began a relaxation to its energetically favored state, confirmed by the slowly decreased FL intensity.

1.1.2 Controlled metastable LSPs

Indeed, in the conventional supramolecular polymerization system, which is a thermodynamic equilibrium system, the final outcome (morphology, size, structure, etc.) of supramolecular polymer is only dictated by the final conditions (the ratio of seeds and monomers, concentration, temperature, and solvent composition, etc.). Because the thermodynamic equilibrium is represented by the global energy minimum of the energy landscape and the overall system does not change over time (*Nat. Nanotechnol.* **10**, 111–119 (2015); *Chem. Soc. Rev.* **47**, 3788–3803 (2018)).

In contrast, in a kinetic and non-equilibrium system (metastable), the preparation protocol leads to structurally different aggregates from the same building blocks under the same final conditions (solvent composition, temperature, pH, anionic strength, etc). In other words, the final energy landscape is fixed, as a function of molecular design and final conditions, while the final outcome of the aggregation process depends on the path by which monomers self-assemble into aggregates (*Chem. Soc. Rev.* **46**, 5476–5490 (2017)).

In order to control metastable LSPs, we tried to change the confinement size by tuning LDH size. When the assembly of SG7 passes through different confinement pathway, the size of the obtained product has undergone significant changes (**detailed in Example 1**). In fact, such parameters determine the actual conditions at which the nucleation takes place, and consequently its rate. In addition, we have also designed conventional control methods for realizing controllability of metastable LSP by changing experimental parameters, such as

timing of addition of the bad solvent (**detailed in Example 2**) and mechanical agitation (**detailed in Example 3**). To sum up, the size of non-equilibrium LSP (metastable LSP) is not simply controlled by concentration, but by the pathway.

Example 1: Pathway selection *via* confinement size

The final condition: solvent composition = CH₃OH/TFA (1:3 v/v); temperature = 0 °C; [SG7] = 2.5 mM (by dissolving 4 mg SG7-LDH into 800 μL CH₃OH/TFA (1:3 v/v) at 0 min); cationic concentration (ion strength) [Mg²⁺] = 75 mM, [Al³⁺] = 25 mM; ultrasound time = 60 min; SG7-LDH with different sizes showed similar weight fraction of SG7 in SG7-LDH (~26 wt.%) by elemental analysis.

Control the confinement size (LDH size) to change the size of metastable LSP. The experimental results showed that with the increase of LDH size (the size of confinement pathway) from 20 to 3000 nm, the metastable LSP₂₀–LSP₃₀₀₀ had increased sizes in SEM images, while they are distinct in energy and consequently its activity in elongation (Supplementary Fig. 34–35). The different outcomes are closely related to the different pathways (Supplementary Fig. 36). See the following explanation for details:

Due to going through increased confinement space (different pathways), the metastable LSM₂₀–LSM₃₀₀₀ ($\lambda_{\text{ex}} = 370$ nm) showed increased sizes (Fig. 2), accompanied by increased emission wavelengths at 470, 472, 474 and 503 nm, respectively (Supplementary Fig. 35).

The appearance of metastable LSM with different sizes leads to different rate of primary nucleation of metastable LSP, as confirmed by the different appearance time at 20, 30, 25 and 10 min, respectively (Supplementary Fig. 35). During the process, the kinetics of transformation and the intermolecular interaction in structure of LSP₂₀–LSP₃₀₀₀ are different.

As seen in the time-dependent FL spectra of LSP₂₀–LSP₃₀₀₀ (Supplementary Fig. 35), the rate of FL intensity decline and degree of red-shift are different from each other, indicating that the kinetics of transformation depend on confinement space due to pathway complexity.

In addition, the pathway not only influenced the rate of primary nucleation of metastable LSP but also in energy, confirmed by their different emission wavelength at 525, 510, 516 and 523 nm, respectively) ($\lambda_{\text{ex}} = 370$ nm, Supplementary Fig. 35–36). Interestingly, metastable LSP₂₀–LSP₃₀₀₀ had different activity in elongation during the relaxation to the energetically favored state, confirmed by the different area of elongated LSP₂₀–LSP₃₀₀₀ (1.50, 0.32, 2.0 and 4.0 μm^2).

Example 2: Pathway selection *via* solvent processing

The final condition: solvent composition = CH₃OH/TFA (1:3 v/v); temperature = 0 °C; [SG7] = 2.5 mM (by dissolving 4 mg SG7-LDH₂₀ into 800 μL CH₃OH/TFA (1:3 v/v)); cationic concentration (ion strength) [Mg²⁺] = 75 mM, [Al³⁺] = 25 mM; size of LDH = 20 nm. The different kinetic assemblies of metastable LSP were obtained by change timing of addition of the bad solvent to SG7-LDH (4 mg in 200 μL CH₃OH) during self-assembly. Condition (1) 0 min with 600 μL ; Condition (2) 0 min with 200 μL and 30 min with 400 μL ; Condition (3) 0 min with 120 μL and 30 min 480 μL ; Condition (4) 0 min with 120 μL and 24 h with 480 μL (Fig. 3a). The fluorescence properties of samples were tested under magnetic stirring (Supplementary Fig. 13) within the experimental timescale of 60 min. SEM images of products in Condition (1)–(4) were prepared from samples at the same experimental time (60 min) (Supplementary Fig. 38).

Control the timing of addition of the bad solvent to change the size of metastable LSP.

The size of metastable LSP₂₀ is dependent on the timing of addition of TFA. Similar results have been reported by Rybtchinski and co-worker, whom studied on the pathway dependent self-assembly of a perylene diimide terpyridine trapping (*Nat. Nanotechnol.* **6**, 141–146 (2011)).

Condition (1): As seen in the time-dependent FL spectra ($\lambda_{\text{ex}} = 370 \text{ nm}$) (Supplementary Fig. 35), metastable LSP₂₀ ($\lambda_{\text{em}} = 525 \text{ nm}$) appeared at 20 min, which was the most fast compared with other conditions. SEM image showed that the size of metastable LSP was $\sim 0.04 \mu\text{m}^2$ (Supplementary Fig. 38). Condition (2): The metastable LSP₂₀ appeared at 45 min, which was slower than that of Condition (1) (Supplementary Fig. 37). The size of metastable LSP₂₀ was $\sim 0.006 \mu\text{m}^2$, which was smaller than that of condition (1) (Supplementary Fig. 38). Condition (3): The metastable LSP₂₀ appeared at 48 min, which was slower than that of condition (1) and (2) (Supplementary Fig. 9). The size of metastable LSP is $\sim 0.0036 \mu\text{m}^2$ (Supplementary Fig. 38), which is smaller than that of condition (1) and (2). Within the experimental practical timescale (60 min) of condition (1)–(3), the change in timing of addition of the bad solvent determines the actual conditions at which the nucleation takes place, and consequently its rate. Therefore, the metastable LSP₂₀ showed decreased size. In condition (4), although the final condition is similar, the metastable LSP₂₀ undergone different pathway. In CH₃OH/TFA (5:3 v/v), metastable LSM₂₀ has accessed to its energetically favored state before 24 h, leading to the failed transformation to LSP₂₀ in CH₃OH/TFA (1:3 v/v) due to serious stacking (Supplementary Fig. 9 and 38).

Example 3: Pathway selection *via* mechanical agitation

The final condition: solvent composition = CH₃OH/TFA (1:3 v/v); temperature = 0 °C;

[SG7] = 2.5 mM (by dissolving 4 mg SG7-LDH₂₀ into 800 μL CH₃OH/TFA (1:3 v/v) at 0 min); cationic concentration (ion strength) [Mg²⁺] = 75 mM, [Al³⁺] = 25 mM; size of LDH = 20 nm. The different kinetic assemblies of metastable LSP obtained with ultrasound or without ultrasound.

The size of metastable LSP₂₀ at 60 min is influenced by mechanical agitation. The experimental results show metastable LSP obtained with ultrasound and without ultrasound showed distinct sizes in SEM images. The sonicated metastable LSP₂₀ shows smaller but more uniform size than that of without ultrasound (Supplementary Fig. 15a and 39a). Due to ultrasound, active bonding sites were exposed sufficiently to form heterogeneous nuclei of metastable LSP₂₀. Similar results have been reported by Takeuchi and Sugiyasu (*Nat. Chem.* **6**, 188–195 (2014)), whom studied on the effect of mechanical agitation on rate-determining nucleation process of porphyrin-based supramolecular assembly.

1.2 The sizes of metastable LSP and LDH cannot be directly compared, because the latter has been removed by etching before formation of metastable LSP.

It is worthy to note that LDH template only exists in step 1 and 2. It can be concluded that the size of metastable LSM is controlled by LDH size. As shown in the SEM images of partly etched LDH, metastable LSM is much smaller than LDH (Fig. 2). However, during the transformation from metastable LSM to metastable LSP, LDH template has been completely removed. Hence, the elongation process of metastable LSP will not be restricted by original LDH size. The key of controlling the size of metastable LSP is its pathway of formation. In our work, one of the most important parameters of pathway is the size of LDH (detailed in

response 1.1. Example 1).

1.3 Last but not least, as you suggested, we supplemented the repeated SEM and SLS of metastable LSP to make our results more convincing (Supplementary Fig. 34 and Table 3).

Herein, the final condition: solvent composition = CH₃OH/TFA (1:3 v/v); temperature = 0 °C; [SG7] = 2.5 mM (by dissolving 4 mg SG7-LDH into 800 μL CH₃OH/TFA (1:3 v/v) at 0 min); cationic concentration (ion strength) [Mg²⁺] = 75 mM, [Al³⁺] = 25 mM; ultrasound time = 60 min. The different metastable LSP₂₀–LSP₃₀₀₀ obtained from SG7-LDH with a set of size (20 nm, 50 nm, 100 nm and 3 μm) showed distinct sizes in SEM images.

The corresponding polydispersity index (PDI) has been calculated by evaluating over 50 metastable LSP (Supplementary Table 3). For metastable LSP₂₀, the number-average area (A_n), weight-average area (A_w), and PDI (A_w/A_n) were 0.03880 μm², 0.04035 μm² and 1.040, respectively. For metastable LSP₅₀, the number-average area (A_n), weight-average area (A_w), and PDI (A_w/A_n) were 0.07115 μm², 0.07657 μm² and 1.076, respectively. For metastable LSP₁₀₀, the number-average area (A_n), weight-average area (A_w), and PDI (A_w/A_n) were 0.1131 μm², 0.1192 μm² and 1.054, respectively. For metastable LSP₃₀₀₀, the number-average area (A_n), weight-average area (A_w), and PDI (A_w/A_n) were 0.7639 μm², 0.8144 μm² and 1.066, respectively.

1.4 The size of final LSP and SSP is not simply controlled by ratio of seeds and monomers, but by the formation pathway.

As the reviewer said, owing to the living ends, for the metastable LSP going through the

same pathway, the size of final LSP should be controlled by ratio of seeds and monomers. However, our purpose is to change the assembly pathway by changing the LDH size to achieve the controlled outcome. Thus, after careful consideration and extensive investigation of non-equilibrium system (*Chem. Soc. Rev.* **46**, 5476–5490 (2017); *J. Am. Chem. Soc.* **136**, 8540–8543 (2014); *J. Am. Chem. Soc.* **130**, 15176–15184 (2008)), we have reservations about the previous conclusion, that is, the effect of LDH size on the size of metastable LSP and its energetically favored state in **response 1.1. Example 1**. The key of controlling the size of metastable LSP and consequently its energetically favored state (Fig. 2 and Supplementary Fig. 34) is the pathway of formation. One of the most important parameters of pathway is the size of LDH (detailed in **response 1.1. Example 1**)

Certainly, if metastable LSP is used as a seed, it goes through the same pathway, the final size of as-obtained SSP is controlled by the ratio of seeds and monomer. As shown in SEM images in Supplementary Fig. 43, the size of SSP increased with the ratio of LSP_{seed} and monomer.

Figure 2. Tuning the size of LSP *via* LDH confinement. (a–d) SEM images of metastable LSM prepared from SG7-LDHs with different size: (a) 20 nm, (b) 50 nm, (c) 100 nm and (d) 3 μm (insets are corresponding magnifications and the sizes of metastable LSM in the insets are marked with a yellow line). **(e–h)** SEM images of elongated LSP from the corresponding metastable LSM in a–d, respectively. **(i–l)** Debye plot of metastable LSP made of corresponding metastable LSM in a–d, respectively. Error bars are calculated using the standard error formalism for the data of three replicate experiments.

Figure 3. Tuning the size of LSP via solvent processing and mechanical agitation. (a) Schematic illustration for condition (1)–(4) to study different kinetic assemblies via changing timing of addition of TFA (0, 30 and 1440 min). (b–c) CLSM images of: (b) metastable LSP₂₀ and (c) elongated LSP₂₀ for 12 h from b. (d) Average size of elongated LSP₂₀ prepared by different ultrasound time.

Supplementary Figure 3. XRD patterns of (a) pure LDH precursor powder and (b) SG7-LDH with different sizes.

Supplementary Figure 9. FL spectra of metastable LSM₂₀ in CH₃OH/TFA (5:3 v/v) within (a) 0–10 min and (b) 10–30 min. FL spectra of the transformation of metastable LSM₂₀ in CH₃OH/TFA (1:3 v/v), after keeping metastable LSM₂₀ in CH₃OH/TFA (5:3 v/v) for (c) 30 min and (d) 24 h.

Supplementary Figure 10. Temperature-dependent absorption of metastable LSP with a set of low concentration and supersaturated SG7 solution in CH₃OH/TFA (1:3 v/v) (5 mM) at 475 nm in UV-vis spectra.

Supplementary Figure 12. FITS of freshly sonicated LSP₂₀ under magnetic stirring at 535 nm.

Supplementary Figure 13. Schematic presentation of experimental set-up of FITS under magnetic stirring.

Supplementary Figure 15. SEM images of LSP₂₀ with ultrasound elongated for different time: (a) 0 h, (b) 1 h, (c) 3 h and (d) 12 h.

Supplementary Figure 25. XRD patterns of SG7 SSP (orange), SG7 LSP (red), SG7 LSM (blue), destroyed CI-LDH in TFA (light gray), SG7_{agg} (gray), SG7 powder (dark gray) and SG7 crystal (black).

Supplementary Figure 34. SEM images of **(a₁–a₃)** metastable LSP₂₀, **(b₁–b₃)** metastable LSP₅₀, **(c₁–c₃)** metastable LSP₁₀₀ and **(d₁–d₃)** metastable LSP₃₀₀₀ in batch 1-3, respectively. **(a₄–d₄)** Area distribution of LSP₂₀–LSP₃₀₀₀, respectively.

Supplementary Table 3. The number-average area (A_n), weight-average area (A_w), and PDI (A_w/A_n) of LSP₂₀–LSP₃₀₀₀, respectively, which was obtained by evaluating over 50 objects in SEM images.

Various LSPs	A_n (μm^2)	A_w (μm^2)	PDI (A_w/A_n)
LSP ₂₀	0.03880	0.04035	1.040
LSP ₅₀	0.07115	0.07657	1.076
LSP ₁₀₀	0.1131	0.1192	1.054
LSP ₃₀₀₀	0.7639	0.8144	1.066

Supplementary Figure 35. FL spectra of transformation from metastable (a) LSM₂₀; (b) LSM₅₀; (c) LSM₁₀₀ and (d) LSM₃₀₀₀ to corresponding LSP in CH₃OH/TFA (1:3 v/v). (e–h) Normalized FL spectra in a–d, respectively. (i–l) Time-dependent FL intensity and wavelength in a–d, respectively.

Supplementary Figure 36. Schematic illustration of pathway for the formation of metastable LSP₂₀–LSP₃₀₀₀.

Supplementary Figure 37. (a) FL spectra of metastable LSM₂₀ in CH₃OH/TFA (1:1 v/v). (b) FL spectra of metastable LSM₂₀ from (a), transforming to metastable LSP₂₀ after adding TFA until CH₃OH/TFA (1:3 v/v).

Supplementary Figure 38. SEM images of (a–c) metastable LSP₂₀ prepared from Condition (1)–(3), respectively and (d) failed metastable LSP₂₀ of Condition (4).

Supplementary Figure 39. SEM image of LSP₂₀ without ultrasound kept for (a) 1 h and (b) 12 h.

p. 6 Line 110 in manuscript:

[revised manuscript text omitted]

2. Energy Diagram of Figure 3j

This schematic illustration well explains that the authors do not understand the basics of supramolecular assemblies. I request the authors to well consider the first step depicted here, showing that molecular dispersed SG7 (mentioned as SG7 solution) is the most energetically stable state. If the molecularly dispersed state of SG7 is the most stable, what is the driving

force to be incorporated into LDH? The authors drew arrows that the release of SG7 from LDH is irreversible because it cannot exceed the energy hill. However, in this diagram, in order for SG7 to reach the intercalated state in LDH, SG7 must exceed more higher energy hill. Hence, this diagram is apparently wrong. The authors have to study the previous papers such as Nat. Chem. 9, 493, 2017. In Fig.2, there is an energy diagram to explain the mechanism. When we talk about the kinetic assembly and thermodynamic assembly, the energy level of the latter one is lower than the former one. However, in either case, both energies levels are lower than that of their monomeric state. Furthermore, it is possible to compare the energies under the same condition (concentration, temperature, solvents and so on). The authors compared the energies levels in different states. However, in their experiments, solvents are changed. Hence, it is scientifically wrong to compare all the energy states like this.

Response:

Thank you for this valuable comment. As you suggested, we have read the reference carefully (*Nat. Chem.* **9**, 493 (2017)) and cited as a reference. We have corrected the mistake in the energy level of SG7 solution and compared them under the same condition, respectively, as displayed in the corrected Fig. 4i–k.

As shown in the revised Fig. 4i–4j, the energy level of SG7 solution should be higher than that of thermodynamic equilibrium $SG7_{agg}$ and ordered SG7 in LDH. The energy level of SG7 monomer is recalculated to be $167.02 \text{ kJ mol}^{-1}$ higher than that of $SG7_{agg}$ (Fig. 4i) and $126.92 \text{ kJ mol}^{-1}$ higher than that of ordered SG7 in LDH (Fig. 4j). The driving force of incorporating SG7 into LDH is mainly Coulomb force and hydrogen bond (*Chem. Rev.* **112**,

4124-4155 (2012)). The computational methods to obtain the energy levels of SG7 monomer, SG7_{agg}, and ordered SG7 in LDH are listed as follows:

First of all, the models representing SG7 monomer and ordered SG7 in LDH are constructed (detailed information for model construction is listed in the methods section). In SG7 monomer model, eight SG7 molecules are dispersed in solvent ([SG7] = 2.5 mM in CH₃OH/TFA (5:3 v/v)). Firstly, the geometry of SG7 monomer is optimized to obtain its energy, where the arrangement of SG7 molecule is disordered. Secondly, the geometry of SG7-LDH is also optimized. It is found that in LDH, SG7 molecules are orderly arranged with an orientation angle of 9°. Finally, the energy of ordered SG7 in LDH and SG7 monomer is also obtained. The energy level of SG7 monomer is calculated to be 126.9 kJ·mol⁻¹ higher than that of ordered SG7 in LDH.

After that, the potential energy surface of SG7_{agg} composed of eight SG7 molecules is completely searched with orientation angle (θ) of SG7 ranging from 0 to 90° with the step of 1°. The global energy minima is calculated to be at $\theta = 35^\circ$, which is deduced to be the geometry of thermodynamic equilibrium SG7_{agg}. The energy level of SG7 monomer is 167.0 kJ·mol⁻¹ higher than that of SG7_{agg}. The conversion from SG7_{agg} ($\theta = 35^\circ$) to ordered SG7 in LDH ($\theta = 9^\circ$) needs to overcome an energy barrier of 70.55 kJ mol⁻¹ at $\theta = 18^\circ$ (the maximum of potential energy surface from 35° to 9°). Furthermore, the effect of the number of SG7 molecules on the energy diagram is investigated. The potential energy surface of ordered SG7 composed of four molecules is also searched, as displayed in Supplementary Fig. 46b. The global energy minima also lies at $\theta = 35^\circ$, the same with that of ordered SG7 with eight molecules. The ordered SG7 in LDH, corresponding to $\theta = 9^\circ$, is also a local minima, in

accordance with the ordered SG7 with eight molecules. For ordered SG7 with four molecules, the conversion of SG7_{agg} ($\theta = 35^\circ$) to ordered SG7 in LDH ($\theta = 9^\circ$) needs to overcome an energy barrier of 36.22 kJ mol⁻¹ at $\theta = 18^\circ$. In general, the energy barrier overcome by LDH confinement space is influenced by the number of SG7 molecules, but the locations of global minima and local minima are independent of the number of SG7 molecules.

In the revised Fig. 4k, calculating the energy level of SSP needs unaffordable computational cost due to its large number of SG7 molecules, which is beyond the state of art. Thus, the schematic illustration for energy level of metastable LSM, metastable LSP, elongated LSP and SSP is presented according to the experimental results. The FL spectra of the transformation from metastable LSM to LSP were obtained by dissolving 4 mg SG7-LDH₂₀ into 800 μ L CH₃OH/TFA (1:3 v/v) (Supplementary Fig. 35). It should be noted that under the above condition, the system is adiabatic, *i.e.*, without external energy input. During the transformation, metastable LSP ($\lambda_{em} = 525$ nm) has lower energy than metastable LSM ($\lambda_{em} = 470$ nm), confirmed by the red shift (Supplementary Fig. 35). After then, as seen in SEM images (Supplementary Fig. 15 and 19), the elongated LSP showed larger size than that of metastable LSP, indicating that metastable LSP undergone the energetically favored elongation step (*Chem. Soc. Rev.* **46**, 5476–5490 (2017)), so is the case for SSP. Thus, both the energy of elongated LSP and SSP are lower than that of metastable LSP. Due to the addition of SG7_{agg}, the energy of SSP ($\lambda_{em} < 525$ nm) is higher than that of elongated LSP ($\lambda_{em} = 535$ nm), confirmed by blue shift ($\Delta\lambda = 10$ nm) in FL spectra (Supplementary Fig. 11 and 16).

Figure 4. Kinetics behavior and Living supramolecular polymerization of SSP. (a–c) The size of SSP as a function of (a) the concentrations of SG7_{agg} , (b) the volume ratio of added SG7_{agg} to the LSP_{seed} , (c) the concentrations of LSP_{seed} . (d) Log–log plot of the rate of increased absorbance at 460 nm as a function of the LSP_{seed} concentration. (e) Time scan of absorbance of SSP_{20} prepared by LSP_{seed} with different concentrations. (f–g) FL spectra of SSP_{20} obtained in Cycle 1–3. (h) The size of SSP_{20} as a function of cycle number x ($x = 1, 2$ and 3) (insets: corresponding SEM images of SSP_{20} obtained in Cycle 1–3, respectively). Schematic illustration for the energy diagrams of (j) SG7 monomer and SG7_{agg}, (k) SG7 monomer and SG7-LDH, together with (l) the products during the formation of LSP and SSP. All error bars are calculated using the standard error formalism for the data of three replicate experiments.

Supplementary Figure 11. FL spectra of LSP₂₀–LSP₃₀₀₀: **(a)** freshly sonicated metastable LSP and **(b)** elongated LSP (12 h) after ultrasound.

Supplementary Figure 15. SEM images of LSP₂₀ with ultrasound elongated for different time: **(a)** 0 h, **(b)** 1 h, **(c)** 3 h and **(d)** 12 h.

Supplementary Figure 16. FL spectra of fresh SSP₂₀ made from LSP_{seed} with different concentrations.

Supplementary Figure 19. SEM images of (a) metastable LSP made of SG7, corresponding fresh SSPs in (b) Cycle 1 and (c) Cycle 2, and (d) contrast sample: physically mixing pure SG7 and Cl-LDH in CH₃OH/TFA (1:3 v/v) with the same dosage as LSP.

Supplementary Figure 35. FL spectra of transformation from metastable (a) LSM₂₀; (b) LSM₅₀; (c) LSM₁₀₀ and (d) LSM₃₀₀₀ to corresponding LSP in CH₃OH/TFA (1:3 v/v). (e–h) Normalized FL spectra in a–d, respectively. (i–l) Time-dependent FL intensity and wavelength in a–d, respectively.

Supplementary Figure 46. (a) Schematic illustration for the definition of orientation angle, θ . (b) Energy diagram of SG7 LSM with different orientation. Optimized geometries of (c) SG7-LDH, (d) inactive SG7 aggregates ($\theta = 35^\circ$), (e) SG7 LSM+SG7 monomer ($\theta = 35^\circ$) and (f) SG7 LSM+SG7 monomer ($\theta = 9^\circ$), together with the highlighted binding areas in e and f: (g) $\theta = 35^\circ$ and (h) $\theta = 9^\circ$. The color of each element is labeled in c.

p. 16 Line 298 in manuscript:

The above experiments have fully proved the mechanism of the transformation from metastable LSM to LSP to be energetically favored. This beneficial elongation behavior derives from the huge energy barrier overcome by LDH, confirmed by theoretical calculation and optical information. As proved by theoretical calculation (Supplementary Fig. 46,

detailed information for model construction is listed in the methods section), the locations of global minima and local minima are independent of the number of SG7 molecules, the energies of SG7 monomer, SG7_{agg} and ordered SG7 in LDH can be calculated and compared by using the representative models composed of eight SG7 molecules, as displayed in Fig. 4i–j. The energy of SG7 monomer is calculated to be 167.02 kJ mol⁻¹ higher than that of SG7_{agg} and 126.92 kJ mol⁻¹ higher than that of ordered SG7 in LDH (Supplementary Fig. 46). The driving force of incorporating SG7 into LDH is mainly Coulomb force and hydrogen bond. Thus, the energy level of SG7 solution should be higher than that of thermodynamic equilibrium SG7_{agg} and ordered SG7 in LDH. In addition, the conversion from SG7_{agg} ($\theta = 35^\circ$) to ordered SG7 in LDH ($\theta = 9^\circ$) needs to overcome an energy barrier of 70.55 kJ mol⁻¹ at $\theta = 18^\circ$ (the maximum in potential energy surface from 35° to 9°). Thus, ordered SG7 in LDH ($\theta = 9^\circ$) is kept in an energy well to be a metastable state.

With the increased number of SG7 molecules, calculating the energy level of following transformation needs unaffordable computational cost, which is beyond the state of art. Thus, the schematic illustration for energy level of metastable LSM, metastable LSP, elongated LSP and SSP is presented according to the optical information (Fig. 4k). During the transformation, metastable LSP ($\lambda_{em} = 525$ nm) has lower energy than metastable LSM ($\lambda_{em} = 470$ nm), confirmed by the red shift in Supplementary Fig. 35. After then, the energy of elongated LSP and SSP are lower than that of metastable LSP due to the energetically favored elongation step.

3. Definition of supramolecular polymerization

In my previous comment, I pointed out this is just crystallization rather than supramolecular polymerization. However, I could not understand what the authors wanted to insist. In my opinion, supramolecular polymerization is either one-dimensional or two-dimensional assembling processes. If the authors wanted to claim that their work can be regarded as supramolecular polymerization rather than crystallization, the authors simply explain the differences of these two in their definitions with references. The unrelated discussion together with data in the answer made me strongly hesitate to continue to review this manuscript.

Response:

Thank you very much for your comments. As you suggested, we have systematically studied and compared the difference between crystallization, supramolecular polymerization and crystallization-driven self-assembly in terms of their definition and the product. After that, we analyzed in detail our experimental process and results. By comparison, it proved that our system belongs to supramolecular polymerization.

3.1 Basic concept

3.1.1 Crystallization

The definition. Crystallization is the process by which a long-range ordered spatial arrangement is formed (*Nat. Rev. Chem.* **4**, 38–53 (2020)). Traditionally, crystallization has been widely described by the classical monomer-by-monomer addition to an isolated cluster. Such a growth mechanism relies on the Ostwald ripening process, that is, the dissolution of small particles in favor of large particles over time, driven by the decrease of surface energy. During this process, crystal growth is initiated by the formation of highly ordered nuclei in a

supersaturated medium (*Acc. Chem. Res.* **42**, 621–629 (2009)). Based on the Gibbs-Thompson law, an energy difference exists between the large and small particles, which results in the further growth of larger particles and the dissolution of the small particles (*J. Stat. Phys.* **38**, 231 (1985); *Science* **349**, aaa6760 (2015)).

The products. In molecular crystals, it is difficult to define a dominant direction of the interactions (crystals are fundamentally 3-dimensional) and even when interactions are stronger in one direction than in others, all specific aggregation is lost when these materials are heated or dissolved. Crystals can be confirmed by XRD results with clear diffraction peak.

3.1.2 Supramolecular polymerization

In a broad sense, both crystallization and supramolecular polymerization refer to a way to build a long-range ordered spatial arrangement for small molecules (only in the sense of organic field). Therefore, the two have many similarities or overlaps (*Nat. Rev. Chem.* **4**, 38–53 (2020); *Angew. Chem. Int. Ed.* **44**, 5071–5074 (2005); *J. Am. Chem. Soc.* **123**, 409–416 (2001); *J. Am. Chem. Soc.* **123**, 409–416 (2001); *Nature* **481**, 492–496 (2012)).

The definition. When the interaction between the monomers is generated by moderately strong, reversible noncovalent, but highly directional, forces that result in high molecular weight linear polymers under dilute conditions, the self-assembly is classified as a supramolecular polymerization. Such a process does not involve the dissolution of primary assemblies, which is distinct from the Ostwald ripening process. Supramolecular polymerizations can be classified on the basis of three different principles: (1) the physical

nature of the noncovalent force that lies at the origin of the reversible interaction (physical origin classification), (2) the type of monomer(s) used (structural monomer classification), and (3) the evolution of the Gibbs free energy of the polymer as a function of conversion (thermodynamical classification). There are three major mechanisms for supramolecular polymerization, namely, isodesmic, ring-chain, and cooperative growth (*Chem. Rev.* **109**, 5687–5754 (2009), this article is cited by 1607 publications).

The products. Although both supramolecular polymer and crystal have the ordered structure, the crystals have clear lattice parameter while supramolecular polymers do not have. The XRD pattern of supramolecular polymers only show broad diffraction peaks, confirming an amorphous structure.

Supramolecular polymers are defined as polymeric arrays of monomeric units that are brought together by reversible and highly directional secondary interactions, resulting in polymeric properties in dilute and concentrated solution as well as in the bulk. The directionality and strength of the supramolecular bonding are important features of systems that can be regarded as polymers and that behave according to well-established theories of polymer physics. Supramolecular polymers are 1-dimensional in nature, and in melts or (dilute) solutions of these materials distinguishable polymeric entities continue to exist (*Chem. Rev.* **101**, 4071–4098 (2001), this article is cited by 2261 publications).

3.1.3 Crystallization-driven self-assembly

The definition. The crystallization-driven self-assembly (CDSA) has two meanings proposed by Winnik and Manners (*Science* **317**, 644–647 (2007); *Macromolecules* **35**,

8258–8260 (2002)). One is that the topological structure of the aggregate formed is different from Eisenberg's classical system due to the crystallization of the crystallizable block; the other is refers to living crystal-driven self-assembly.

The products. The crystalline structure of the crystallizable core block can be confirmed by XRD results with the clear diffraction peak.

3.2 Our experiment results

From the formation process point of view: 1) our system is reversible (**see detailed analysis 1**); 2) the supersaturated solution is not the prerequisite (**see detailed analysis 2**). From the obtained product point of view: our assembly not a crystal (**see detailed analysis 3**).

Detailed analysis 1: Reversibility

Compared with crystals, the significant feature of supramolecular polymer is reversibility. For our system, the reversibility can be proved by disassembly (in CH₃OH/TFA (1:2 v/v)) and re-assembly (in CH₃OH/TFA (1:3 v/v)) of LSP (Supplementary Fig. 20). FL spectra are used to monitor this process. By increasing the ratio of CH₃OH to CH₃OH/TFA (1:2 v/v), metastable LSP ($\lambda_{em} = 525$ nm) disassembled to metastable LSM ($\lambda_{em} = 490$ nm), as confirmed by FL wavelength shift from 525 to 490 nm; by decreasing the ratio of CH₃OH to CH₃OH/TFA (1:3 v/v), metastable LSM could quickly re-assemble to metastable LSP, as confirmed by FL wavelength shift from 490 to 525 nm ($\lambda_{em} = 370$ nm, Supplementary Fig. 20).

Detailed analysis 2: Non-supersaturated condition

In our system, metastable LSP could be prepared with low concentrations (0.15, 0.19, 0.22 and 0.25 mM), which is far below supersaturation ($SG7_{agg}$: $[SG7] > 1$ mM and supersaturation: $[SG7] \gg 1$ mM, detailed in Supplementary Fig. 17). The appearance of metastable LSP can be confirmed by the nonlinear sigmoidal increase in temperature-dependent absorption, while supersaturated SG7 solution (5 mM) shows no increase (Supplementary Fig. 10). Therefore, in our system, supersaturation is not the prerequisite for the formation of LSP.

Moreover, the dilution of metastable LSM had no effect on the transformation from metastable LSM to metastable LSP, which can be confirmed by FL spectra. During dilution, the constant wavelength indicated no disassembly. It should be noted that the decreased FL intensity resulted from dilution (Supplementary Fig. 23). For diluted LSM, it can finally transform to LSP, confirmed by red shift to 525 nm (Supplementary Fig. 24). In contrast, the $SG7_{agg}$ disassembled to SG7 monomer after dilution and no LSP was formed, confirmed by blue shift of $SG7_{agg}$ and absence the emission at 525 nm (Supplementary Fig. 21–22).

Detailed analysis 3: Products

None of products in our system is crystal. This can be confirmed by comparing the XRD results (Supplementary Fig. 25), including metastable LSM, LSP, SSP, untreated SG7 powder, treated SG7 powder in CH_3OH/TFA (1:3 v/v) ($SG7_{agg}$), $Cl-LDH$ dissolved in TFA and SG7 crystal (JCPDS data base card No: 42-1955). Among all samples, only the XRD pattern of $SG7_{agg}$ shows two peaks in line with SG7 crystal. The XRD pattern of LSM shows several weak peaks, which are attributed to the metal salts of the LDH dissolved in TFA. Thus, the SG7 crystal did not appear.

Supplementary Figure 10. Temperature-dependent absorption of metastable LSP with a set of low concentration and supersaturated SG7 solution in CH₃OH/TFA (1:3 v/v) (5 mM) at 475 nm in UV-vis spectra.

Supplementary Figure 17. FL spectra of SG7 powder dispersed in CH₃OH/TFA (1:3 v/v). **(a)** 10–100 μM SG7 with increased intensity. **(b)** 0.1–5 mM SG7 with decreased intensity and red-shifted wavelength. **(c–d)** Normalized FL spectra of a–b, respectively.

Supplementary Figure 20. FL spectra of the reversible disassembly ($\lambda_{em} = 480\text{--}490\text{ nm}$) and re-assembly ($\lambda_{em} = 525\text{ nm}$) of LSP₂₀.

Supplementary Figure 21. (a) FL spectra and (b) normalized FL of SG7 solution (0.715 mM SG7 powder dispersed in CH₃OH/TFA (1:2.5 v/v)).

Supplementary Figure 22. (a) FL spectra and (b) time course of maximum emission wavelength of SG7 solution (0.715 mM SG7 powder dispersed in CH₃OH/TFA (1:2.5 v/v)).

Supplementary Figure 23. (a) FL spectra and (b) normalized FL of diluted LSM in CH₃OH/TFA (1:2.5 v/v).

Supplementary Figure 24. (a) FL spectra and (b) normalized FL of the transformation from metastable LSM₂₀ to metastable LSP₂₀ in CH₃OH/TFA (1:2.5 v/v) ([SG7] = 0.715 mM).

Supplementary Figure 25. XRD patterns of SG7 SSP (orange), SG7 LSP (red), SG7 LSM (blue), destroyed Cl-LDH in TFA (light gray), SG7_{agg} (gray), SG7 powder (dark gray) and SG7 crystal (black).

p. 7 Line 134 in manuscript:

In addition, this system is reversible and can be prepared in low concentration (detailed in Supplementary Fig. 20–24). None of products is crystal (Supplementary Fig. 25). The above results fully prove that this assembly process belongs to supramolecular polymerization.

4. Confinement effect

In my previous comment, I mentioned that the confinement effect never induced supramolecular polymerization. Of course, the confinement effect itself is necessary for the observed results. I don't deny it. Simply, the original title sounded mis-leading. However, here, the authors' answer something unrelated to my comment.

Response:

Thank you for your patience to point out the confusing part of our manuscript. As you suggested, we revised our title as “Oriented Arrangement of Simple Monomers Enabled by Confinement: Towards Living Supramolecular Polymerization”.

The detailed explanation is as follows:

In our system, LDH provided a confinement space to guide interlayer SG7 molecules to form ordered array (the predecessor of LSM) that otherwise would not be formed without confinement. The formed metastable LSM can be preserved in energy well (Fig. 4k and Supplementary Fig. 46) to maintain its structure after removal of LDH. After further addition of bad solvent, metastable LSM can transform to another metastable LSP state by autocatalysis process.

Moreover, LDH with different sizes provided different confinement space, indicating that the formation of LSP undergone different pathway (detailed in **response 1.1 Example 1**, Supplementary Fig. 35–36). The different confinement space endows metastable LSM with different sizes in primary nucleation step of metastable LSP, leading to that the transformation from metastable LSM_{20} – LSM_{3000} to metastable LSP_{20} – LSP_{3000} went through different pathway and energy diagram (Supplementary Fig. 36). During the process, the kinetics of transformation and the intermolecular interaction in structure of LSP_{20} – LSP_{3000} are different. As seen in the time-dependent FL spectra of LSP_{20} – LSP_{3000} (Supplementary Fig. 35), the rate of FL intensity decline and red shift are different from each other, indicating that the kinetics of transformation depend on confinement space due to pathway complexity. Therefore, among all experimental parameters, confinement is the most important factor for the pathway complexity and the controlled LSP.

Figure 4. Kinetics behavior and Living supramolecular polymerization of SSP. (a–c) The size of SSP as a function of (a) the concentrations of SG7_{agg} , (b) the volume ratio of added SG7_{agg} to the LSP_{seed} , (c) the concentrations of LSP_{seed} . (d) Log–log plot of the rate of increased absorbance at 460 nm as a function of the LSP_{seed} concentration. (e) Time scan of absorbance of SSP_{20} prepared by LSP_{seed} with different concentrations. (f–g) FL spectra of SSP_{20} obtained in Cycle 1–3. (h) The size of SSP_{20} as a function of cycle number x ($x = 1, 2$ and 3) (insets: corresponding SEM images of SSP_{20} obtained in Cycle 1–3, respectively). Schematic illustration for the energy diagrams of (j) SG7 monomer and SG7_{agg}, (k) SG7 monomer and SG7-LDH, together with (l) the products during the formation of LSP and SSP. All error bars are calculated using the standard error formalism for the data of three replicate experiments.

Supplementary Figure 9. FL spectra of metastable LSM₂₀ in CH₃OH/TFA (5:3 v/v) within (a) 0–10 min and (b) 10–30 min. FL spectra of the transformation of metastable LSM₂₀ in CH₃OH/TFA (1:3 v/v), after keeping metastable LSM₂₀ in CH₃OH/TFA (5:3 v/v) for (c) 30 min and (d) 24 h.

Supplementary Figure 35. FL spectra of transformation from metastable (a) LSM₂₀; (b) LSM₅₀; (c) LSM₁₀₀ and (d) LSM₃₀₀₀ to corresponding LSP in CH₃OH/TFA (1:3 v/v). (e–h) Normalized FL spectra in a–d, respectively. (i–l) Time-dependent FL intensity and wavelength in a–d, respectively.

Supplementary Figure 36. Schematic illustration of pathway for the formation of metastable LSP₂₀–LSP₃₀₀₀.

Supplementary Figure 46. (a) Schematic illustration for the definition of orientation angle, θ . (b) Energy diagram of SG7 LSM with different orientation. Optimized geometries of (c) SG7-LDH, (d) inactive SG7 aggregates ($\theta = 35^\circ$), (e) SG7 LSM+SG7 monomer ($\theta = 35^\circ$) and

(f) SG7 LSM+SG7 monomer ($\theta = 9^\circ$), together with the highlighted binding areas in e and f:

(g) $\theta = 35^\circ$ and (h) $\theta = 9^\circ$. The color of each element is labeled in c.

p. 10 Line 188 in manuscript:

During their formation, the decline rate of FL intensity and red shift are different from each other, indicating that the kinetics of transformation depends on confinement space due to pathway complexity (Supplementary Fig. 35). The pathway not only influenced the rate of primary nucleation of metastable LSP but also in energy, confirmed by their different emission wavelength at 525, 510, 516 and 523 nm, respectively) ($\lambda_{\text{ex}} = 370$ nm, Supplementary Fig. 35–36). In addition, metastable LSP₂₀–LSP₃₀₀₀ had different activity in elongation during the relaxation to the energetically favored state, confirmed by the different area of elongated LSP₂₀–LSP₃₀₀₀ (1.50, 0.32, 2.0 and 4.0 μm^2 in Fig. 2 e–h).

5. AIMD

MD simulation becomes an important tool to discuss how molecules behave. However, it is still not conclusive one. In particular, the molecular structures of LSM and metastable LSP are not yet understood by experiments. It is very risky to discuss the energy levels with these calculations.

Response:

Thanks for your suggestion. It should be noted that the method applied in this work is *ab initio* molecular dynamics (AIMD) simulations based on quantum mechanics, instead of molecular dynamics (MD) based on Newton mechanics. Unlike traditional molecular

dynamics simulations needing experimental parameters, the AIMD simulations are independent of experimental parameters. Therefore, AIMD simulations can be regarded as conclusive.

It should be clarified that AIMD simulations are performed to investigate the arrangement of SG7 in LDH interlayer, instead of the energy levels of LSM or metastable LSP. The interlayer arrangement of guest anion in LDH interlayer gallery has been widely studied using such method (*J. Am. Chem. Soc.* **130**, 12485-12495 (2008), *Chem. Sci.* **8**, 590-599 (2017)).

The energy levels of SG7 monomer, SG7_{agg}, and ordered SG7 in LDH are calculated with density functional theory (DFT). To guarantee the accuracy of calculated energy levels, two important factors of molecular structure are considered. Firstly, the orientation angle θ of SG7 is screened from 0 to 90° with the step of 1° to guarantee the accurate locations of global minima (thermodynamic assembly) and local minima (kinetic assembly). Secondly, the effect of the number of SG7 molecules on the energy diagram is investigated. The potential energy surface of ordered SG7 composed of four and eight molecules are searched, as displayed in Supplementary Fig. 46b. Although, the energy barrier overcome by LDH confinement space is influenced by the number of SG7 molecules, the locations of global minima and local minima are independent of the number of SG7 molecules (detailed in response 2). Therefore, the molecular structures have been carefully investigated to ensure the accuracy of calculated energy levels.

Supplementary Figure 46. (a) Schematic illustration for the definition of orientation angle, θ . (b) Energy diagram of SG7 LSM with different orientation. Optimized geometries of (c) SG7-LDH, (d) inactive SG7 aggregates ($\theta = 35^\circ$), (e) SG7 LSM+SG7 monomer ($\theta = 35^\circ$) and (f) SG7 LSM+SG7 monomer ($\theta = 9^\circ$), together with the highlighted binding areas in e and f: (g) $\theta = 35^\circ$ and (h) $\theta = 9^\circ$. The color of each element is labeled in c.

p. 16 Line 298 in manuscript:

The above experiments have fully proved the mechanism of the transformation from metastable LSM to LSP to be energetically favored. This beneficial elongation behavior derives from the huge energy barrier overcome by LDH, confirmed by theoretical calculation and optical information. As proved by theoretical calculation (Supplementary Fig. 46,

detailed information for model construction is listed in the methods section), the locations of global minima and local minima are independent of the number of SG7 molecules, the energies of SG7 monomer, SG7_{agg} and ordered SG7 in LDH can be calculated and compared by using the representative models composed of eight SG7 molecules, as displayed in Fig. 4i–j. The energy of SG7 monomer is calculated to be 167.02 kJ mol⁻¹ higher than that of SG7_{agg} and 126.92 kJ mol⁻¹ higher than that of ordered SG7 in LDH (Supplementary Fig. 46). The driving force of incorporating SG7 into LDH is mainly Coulomb force and hydrogen bond. Thus, the energy level of SG7 solution should be higher than that of thermodynamic equilibrium SG7_{agg} and ordered SG7 in LDH. In addition, the conversion from SG7_{agg} ($\theta = 35^\circ$) to ordered SG7 in LDH ($\theta = 9^\circ$) needs to overcome an energy barrier of 70.55 kJ mol⁻¹ at $\theta = 18^\circ$ (the maximum in potential energy surface from 35° to 9°). Thus, ordered SG7 in LDH ($\theta = 9^\circ$) is kept in an energy well to be a metastable state.

REVIEWERS' COMMENTS

Reviewer #2 (Remarks to the Author):

This is a fine revision. Authors have thoroughly revised the manuscript as per the reviewer's comments. The revised manuscript is recommended for publication in its current form.

Reviewer #3 (Remarks to the Author):

I appreciated the great efforts of the authors in order to answer to the questions raised by me. The most of my concerns were finally solved by these answers. Hence, I don't oppose to this manuscript being accepted in Nature Communications.

Responses to Reviewer 2

Overview: This is a fine revision. Authors have thoroughly revised the manuscript as per the reviewer's comments. The revised manuscript is recommended for publication in its current form.

Response:

Thank you for your recommending for publication without any revision.

Responses to Reviewer 3

Overview: I appreciated the great efforts of the authors in order to answer to the questions raised by me. The most of my concerns were finally solved by these answers. Hence, I don't oppose to this manuscript being accepted in Nature Communications.

Response:

Thank you for agreeing with the acceptance of this manuscript in its current form.